# Bridging Temporal and Semantic Gaps: Prompt Learning on Temporal Interaction Graphs

## Abstract

Temporal Interaction Graphs (TIGs) are widely utilized to represent real-world systems like e-commerce and social networks. While various TIG models have been proposed for representation learning, they face two critical gaps in their "pre-train, predict" training paradigm: a temporal gap limiting timely predictions and a semantic gap reducing adaptability to diverse downstream tasks. A potential solution is applying the "pre-train, prompt" paradigm, yet existing static graph prompting methods fail to address the time-sensitive dynamics of TIGs and have a deficiency in expressive power. To tackle these issues, we propose **Temporal Interaction Graph Prompting (TIGPrompt)**, a versatile framework that bridges the temporal and semantic gaps by integrating with existing TIG models. Specifically, we propose a "pre-train, prompt" training paradigm for TIGs, with a temporal prompt generator to offer temporally-aware prompts for different tasks. To cater to varying computational resource demands, we propose an extended "pre-train, prompt-based fine-tune" paradigm, offering greater flexibility. Through extensive experiments involving multiple benchmarks, representative TIG models, and downstream tasks, our TIGPrompt demonstrates the SOTA performance and remarkable efficiency advantages. The codes are available at an Anonymous Repository.

## 1 Introduction

In real-world scenarios, interaction data is often accompanied by temporal information, i.e., timestamps, necessitating its modeling as Temporal Interaction Graphs (TIGs) (Dai et al., 2016; Zhang et al., 2017). In this context, static graphs can hardly model such TIGs since they lack the necessary expressiveness to capture temporal dependencies. Specifically, in TIGs, objects are depicted as nodes, while timestamped interactions between these objects are represented as edges. Consequently, significant research efforts have been dedicated to TIG representation learning models (TIG models) (Trivedi et al., 2019; Xu et al., 2020; Rossi et al., 2020; Zhang et al., 2023c). These works aim to capture the dynamic nature of TIGs and learn temporal node representations, which can be applied to various downstream tasks (Kumar et al., 2019; Rossi et al., 2020; Zhang et al., 2023c).

**The "pre-train, predict" paradigm of existing TIG models.** Recently, researchers have tried to explore the design of TIG models, leading to various effective TIG model structures (Zhang et al., 2023c;b;a). For example, TGN (Rossi et al., 2020) employs a memory module to store historical information of nodes and a message module to store current node embeddings, each with an associated update function that updates the memory and node representations. Although powerful, as illustrated in Fig. 2 (a), we observe that nearly all of these models adopt a "*pre-train, predict*" learning framework, where a TIG model is pre-trained on a specific task (e.g., link prediction) and its learned knowledge is then transferred to various downstream tasks by tuning a corresponding predictor (e.g., MLP (Bishop & Nasrabadi, 2006)).

**Limitations of the "pre-train, predict" paradigm.** In this paper, we analyze the prevailing "pre-train–predict" paradigm in TIG models and identify two critical limitations: the **temporal gap** and the **semantic gap**. First, as temporal interactions evolve, pre-trained models quickly become outdated, leading to degraded performance on distant-future data (i.e., the **temporal gap**) (Zhou et al., 2022; Chen et al., 2023b). As shown in Fig. 1 (a), our preliminary experiments simulate this scenario and reveal a clear performance disparity between temporally proximal and temporally distant inference data, providing evidence of the existence of the temporal gap. However, mitigating this gap under

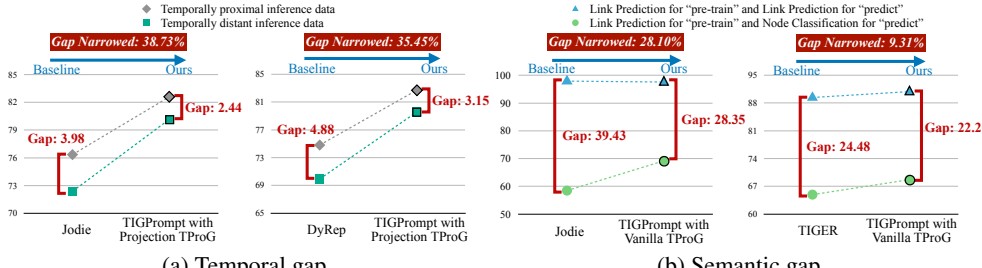

(a) Temporal gap.                    (b) Semantic gap.

Figure 1: Empirical analysis of the temporal gap and semantic gap on real-world TIG data. Our proposed TIG-Prompt can effectively narrow these two gaps for better TIG representation learning. For more implementation details, please refer to Appendix A.

the "pre-train, predict" paradigm typically requires exhaustive re-training to incorporate new data recursively into model updating, resulting in a significant consumption of computational resources (Devlin et al., 2018). Second, misalignment between pretext tasks and downstream objectives significantly limits transferability across tasks (i.e., the **semantic gap**). For instance, while most TIG models are pre-trained on edge-level prediction, downstream tasks may involve node-level objectives, which can even cause negative transfer (Sun et al., 2023). Fig. 1 (b) further validates the existence of this semantic gap. Such misalignment reduces the adaptability of TIG models, thereby constraining their effectiveness in handling various downstream tasks. The detailed definitions, illustrative examples, and quantification of the two gaps are provided in the Appendix A.

**Prompt learning paradigm on static graphs.** The aforementioned two gaps caused by the "pre-train, predict" paradigm call for a more flexible training paradigm for TIG models. Graph prompt learning offers such a potential solution by enabling efficient adaptation of pre-trained models through the design and training of lightweight prompts, while keeping the backbone model unchanged (Liu et al., 2023b; Fang et al., 2023). As demonstrated in static graph settings, prompt learning can not only reduce the cost of adapting models to evolving data compared with full re-training (Liu et al., 2023a), but also explicitly incorporate task-specific knowledge through prompt vectors (Sun et al., 2023), thereby providing greater flexibility than traditional learning frameworks.

**Limitations of existing graph prompt learning pradigm.** Existing studies on prompt learning for graphs have predominantly focused on static settings (Sun et al., 2022), providing limited insights into the more complex scenario of TIGs. Most of these methods overlook the temporal nature of TIGs, failing to incorporate temporal information into prompts to capture their evolving characteristics (Dai et al., 2016). In addition, current approaches typically employ over-simplified prompt vectors shared across all nodes (Liu et al., 2023b). While such designs may suffice for static graphs, they are inadequate for TIGs, where node representations evolve continuously and demand personalized updates over time. These limitations give rise to two technical challenges that hinder the direct application of traditional static graph prompt learning to TIGs. The **first challenge** is how to learn expressive prompts with the minimal cost to overcome the temporal gap caused by emerging data. The **second challenge** is how to design flexible and temporal-aware prompts that can support various TIG models and break down the semantic gap within diverse downstream application scenarios.

**Present work.** In this paper, we propose a new training architecture for TIG models, namely Temporal Interaction Graph Prompting (**TIGPrompt**), as shown in Fig. 2 (b). TIGPrompt instantiates a *"pre-train, prompt"* paradigm through a **T**emporal **Pro**mpt **G**enerator (**TProG**), which intelligently generates personalized temporal prompts for each node. By explicitly incorporating temporal information, the prompts adapt to timestamp-specific variability, thereby bridging the temporal gap and overcoming the limitations of static graph prompting methods. Furthermore, to mitigate the semantic gap between pretext and downstream tasks, the TProG is jointly tuned with the specific downstream task, facilitating adaptability to concrete down-

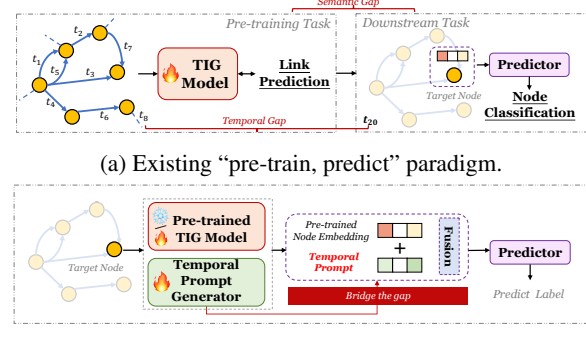

(a) Existing "pre-train, predict" paradigm.

(b) Our introduced prompting mechanism.

Figure 2: (a): The "pre-train, predict" paradigm adopted by existing TIG models, which exhibits both temporal and semantic gaps when applied on the downstream task. (b): Our introduced prompting mechanism, with an innovative TProG, designed to mitigate both gaps.

stream scenarios. Notably, TIGPrompt is lightweight, as it involves only tuning the TIGPrompt while keeping the TIG model frozen. It is also tolerant to weak supervision, requiring only a small portion of data for pre-training and prompt tuning. Furthermore, we extend the "pre-train, prompt" paradigm to cater to varying computational resource demands by introducing a *"pre-train, prompt-based fine-tune"* solution. We summarize our contributions as follows:

- We identify two critical gaps in the prevailing TIG training paradigm and study the prompting mechanism on TIG models. This is the first attempt that explores prompting on TIGs.

- We propose a "pre-train, prompt" paradigm specifically tailored for TIGs, bridging both the temporal and semantic gaps in the traditional training process. Meanwhile, our framework is compatible with various prompt generators and enables dynamic, personalized prompting.

- To enhance the flexibility and accommodate diverse computational resources, we extend the paradigm to a "pre-train, prompt-based fine-tune" solution. Both paradigms can be seamlessly integrated with existing TIG models.

- Extensive experiments on four datasets with seven representative TIG models across two downstream tasks demonstrate that our framework achieves SOTA performance with remarkable efficiency.

## 2 Preliminaries

**Definition of TIG.** Given a node set $\mathcal{V} = \{1, \ldots, |\mathcal{V}|\}$ and a sequence of time-stamped edges $\mathcal{E} = \{(u, v, t_{uv}) \mid u, v \in \mathcal{V},\ t_{uv} > 0\}$, where each edge $(u, v, t_{uv})$ denotes an interaction between nodes $u$ and $v$ at time $t_{uv}$, a TIG is defined as $\mathcal{G} = (\mathcal{V}, \mathcal{E})$. Each interaction may be associated with a feature vector $\mathbf{e}_{uv}(t)$, which encodes event-specific attributes such as interaction type or contextual information. For any interaction $(u, v, t_{uv}) \in \mathcal{E}$, the model has access only to historical events occurring before time $t_{uv}$, i.e., $(i, j, \tau) \in \mathcal{E} \mid \tau < t_{uv}$.

**TIG Models.** Given an interaction event $(u, v, t_{uv}) \in \mathcal{E}$ and its corresponding historical interaction records $(u, v, \tau) \in \mathcal{E} \mid \tau < t_{uv}$, TIG models aim to learn a mapping $f_{\Theta} : (u, v, t_{uv}) \mapsto \mathbf{z}_u(t_{uv}), \mathbf{z}_v(t_{uv})$, where $\mathbf{z}_u(t_{uv}), \mathbf{z}_v(t_{uv}) \in \mathbb{R}^d$ represent the dynamic embeddings of nodes $u$ and $v$ at time $t_{uv}$, and $d$ denotes the dimensionality of the embedding space. At the whole-graph level, the model's output can be equivalently expressed as $\mathbf{Z} = f_{\Theta}(\mathcal{V}, \mathcal{E})$, which yields the time-evolving representations for all nodes in the graph.

**Downstream Tasks.** After optimizing the backbone TIG model, the node representations produced by an arbitrary TIG encoder $f_{\Theta}(\cdot)$ can be retrieved for downstream tasks, formulated as $\hat{\mathbf{Y}} = p_{\Phi}(\mathbf{Z})$, where $p_{\Phi}(\cdot)$ denotes the task-specific projection head (i.e. predictor).

For the link prediction task, the model estimates whether an interaction between two nodes will occur at a future time, typically expressed as $p_{\Phi}(\mathbf{z}_u(t), \mathbf{z}_v(t)) \rightarrow \hat{y}_{uv}(t)$. This objective also serves as the pretext task adopted by most TIG models. Since the supervision signal (future interactions) is inherently available in the TIG, this training paradigm is self-supervised.

For the node classification task, the model predicts node-level labels (e.g., user categories or item types) using the learned dynamic node embeddings: $p_{\Phi}(\mathbf{z}_u(t)) \rightarrow \hat{y}_u$. Here, $p_{\Phi}$ is an additional trainable projection head that is optimized separately from the TIG encoder, and its training requires labeled node instances. As a result, node classification introduces an explicit supervised phase on top of the self-supervised TIG pre-training, where link prediction serves as the pretext task.

## 3 Proposed Method

In this section, we elaborate on the detailed designs within the TIGPrompt framework. We first provide an overview of the "pre-train, prompt" paradigm. Then, we show the implementation and optimization of our Temporal Prompt Generator (TProG) component, which enables the adaptability of pre-trained models across diverse downstream tasks. Finally, we extend this paradigm to the "pre-train, prompt-based fine-tune" mode, specifically devised to accommodate varying computing resource constraints. An overview of our method is illustrated in Fig. 3.

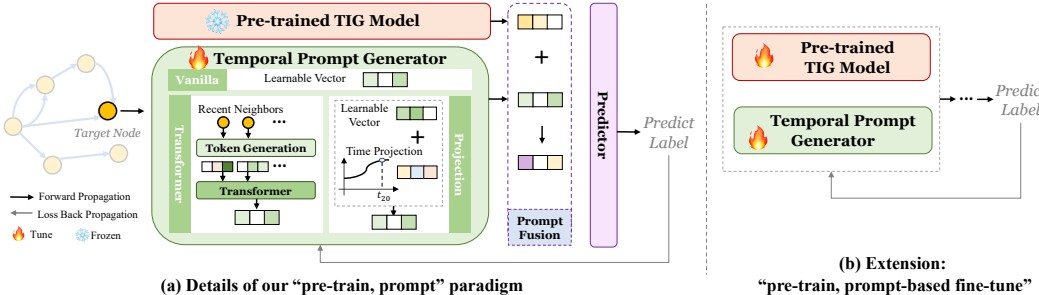

**(a) Details of our "pre-train, prompt" paradigm**

**(b) Extension: "pre-train, prompt-based fine-tune"**

Figure 3: Overview of TIGPrompt: (a) During the prompt tuning stage, the node embedding, calculated by the pre-trained TIG model, is combined with the personalized prompt embedding for downstream tasks. The TProG is optimized during this stage. (b) The key distinction between the two modes lies in whether the parameters of the TIG model are tuned.

### 3.1 "PRE-TRAIN, PROMPT" PARADIGM OVERVIEW

Existing TIG models such as JODIE (Kumar et al., 2019), DyRep (Trivedi et al., 2019), TGN (Rossi et al., 2020), and TIGER (Zhang et al., 2023c) primarily employ link prediction as the pre-training objective, with differences in their concrete model implementation. For instance, TGN (Rossi et al., 2020) introduces a memory-based approach and integrates previous works into a cohesive framework, while TIGER (Zhang et al., 2023c) puts forward a model that incorporates a dual-memory module for effective information aggregation. Once a TIG model is well-trained, node embeddings can be retrieved for task-specific predictions, such as node classification. The predictions are made as: $\hat{\mathbf{Y}} = p_\Phi(\mathbf{Z})$, where $\mathbf{Z} = f_\Theta(\mathcal{V}, \mathcal{E})$. Here, $p_\Phi(\cdot)$ denotes the projection head of the downstream task, $\mathbf{Z}$ denotes the learned node representations obtained from an arbitrary TIG model $f_\Theta(\cdot)$, which takes a TIG, $G(\mathcal{V}, \mathcal{E})$ as input. However, it is important to note that directly utilizing pre-trained node embeddings for downstream tasks is unfeasible as it overlooks two critical gaps: the temporal gap (i.e., the evolving nature of TIGs may render pre-trained node embeddings less expressiveness to the timely TIG data), and the semantic gap (i.e., the distinctions between link-level pretext task and node-level downstream task).

To bridge these gaps and enable the adaptability of a pre-trained TIG model across various scenarios, we propose to utilize personalized and temporal-aware *prompt* for each node. Combined with pre-trained node embeddings, these prompts can carry task-specific semantics to get adapted to different downstream tasks as:

$$\hat{\mathbf{Y}} = p_\Phi(\widetilde{\mathbf{Z}}), \ \widetilde{\mathbf{Z}} = f_\rho(\mathbf{Z}, \mathbf{P}), \tag{1}$$

where $\mathbf{P}$ denotes the prompt matrix produced by the TProG, $f_\rho(\cdot)$ represents the fusion function, and $\widetilde{\mathbf{Z}}$ denotes the final prompted node representations. The prompt generator is tuned with task-specific supervision, enabling the final synthesized node representations contain task-specific and temporal-aware knowledge. Notably, during this process, the pre-trained TIG model $f_\Theta(\cdot)$ remains frozen, making TIGPrompt lightweight to get adapted to concrete downstream scenarios. Then, we move to the description of how these prompts are generated and tuned.

### 3.2 TPROG: TEMPORAL PROMPT GENERATOR

In this subsection, we provide a detailed explanation of our implementation of TProG, which produces a prompt matrix $\mathbf{P} \in \mathbb{R}^{|\mathcal{V}| \times d}$. We initially introduce a *Vanilla* TProG, where a learnable vector is assigned to each node, enabling personalized prompts tailored for specific downstream scenarios. Note that Vanilla TProG can be considered an intermediate bridge between static and temporal interaction graph prompt learning, since it generates personalized prompts but does not inject temporal information. To enhance the temporal awareness of produced prompts, we extend the TProG by introducing two additional approaches: the *Transformer* TProG and the *Projection* TProG.

**Vanilla TProG.** We first introduce the simplest version of TProG, which aims to provide personalized expressiveness for each node. In this approach, the prompt for node $v \in \mathcal{V}$ is implemented as a learnable vector $\mathbf{p}_v \in \mathbb{R}^d$, which is initialized as zero vector. Current methods normally utilize the

link prediction task as the pretext task. In downstream tasks such as node classification, a projection head—commonly an MLP—is used to classify node embeddings derived from the pre-trained model. We enhance these node embeddings with learnable prompt vectors, i.e., Vanilla TProG, which are concurrently optimized with the downstream task's projection head. This strategy effectively embeds task-specific knowledge into the prompt vectors during the prompt tuning phase. This implementation bears a resemblance to traditional prompting techniques utilized in static graphs (Fang et al., 2023; Liu et al., 2023b), and serves as a conceptual bridge between traditional graph prompting and TIG prompt methods. Despite its simplicity, this method offers an intuitive design, easy implementation, and low parameterization, requiring only $\mathcal{O}(|\mathcal{V}|)$ parameters, scaling linearly with the size of the temporal interaction graph.

**Transformer TProG.** To generate temporal-aware prompt, we consider encoding the most relevant temporal information for each node. For a target node $v$, its most recent interactions provide valuable insights into its temporal information, which can be leveraged to generate the temporal prompt $\mathbf{p}_v$.

Therefore, at any timestamp $t$, we first retrieve the node's most recent neighbor set $\mathcal{N}_v^t = \{u | u \in \mathcal{V}, (u, v, t_{uv}) \in \mathcal{E}$ and $t_{uv} \leq t\}$. To avoid an excessively large neighbor set, we impose a restriction on the size of $\mathcal{N}_v^t$, returning only the most recent $K$ interactions. Then, for each neighboring node $u \in \mathcal{N}_v^t$, we first create a temporal neighbor token as: $\mathbf{t}_u = \mathbf{z}_v \,||\, \mathbf{z}_u \,||\, \mathbf{pos}_u \,||\, \mathbf{e}_{uv} \,||\, f_\omega(t - t_{uv})$, where $\mathbf{z}_u, \mathbf{z}_v$ are pre-trained node embeddings, $\mathbf{pos}_u$ corresponds to the position index of node $u$ within the neighbor set, $\mathbf{e}_{uv}$ denotes the edge feature of historical interaction $(u, v, t_{uv})$, $||$ denotes the concatenation operation, and $f_\omega(\cdot)$ denotes a time encoding function (we apply the same time encoding method used in (Xu et al., 2020; Rossi et al., 2020; Zhang et al., 2023c)). In this way, the neighboring token $\mathbf{t}_u$ incorporates both interactive and temporal knowledge, and we further leverage a Transformer (Vaswani et al., 2017) to encode those temporal neighboring tokens to generate temporal prompt $\mathbf{p}_v$ as:

$$\mathbf{p}_v = \text{Transformer}(\{\mathbf{t}_u | u \in \mathcal{N}_v^t\}). \tag{2}$$

This approach ensures that the generated prompt $\mathbf{p}_v$ captures expressive temporal and recent interactive knowledge, promising to enhance downstream predictions. The implementation of Transformer TProG is extremely lightweight, as the number of tunable parameters within this component is $\mathcal{O}(d)$, scaling linearly with the embedding dimension.

**Projection TProG.** In addition to encoding recent neighboring information, we can also generate a temporal-aware prompt by integrating personalized vectors and time encoding. Recall that in the Vanilla TProG, we introduce a learnable vector $\mathbf{p}_v^{\text{Personal}} \in \mathbb{R}^d$ for each node to represent the prompt. To incorporate the temporal knowledge, we fuse this personalized vector with time encoding. Specifically, at timestamp $t$, the temporal information can be encoded as $\mathbf{p}_v^{\text{Temporal}} = f_\omega(t - t_{v'})$, where $t_{v'}$ represents the most recent interaction timestamp of node $v$, and $f_\omega(\cdot)$ is a time encoding function. Finally, the temporal prompt $\mathbf{p}_v$ is generated via integrating both sides of information as:

$$\mathbf{p}_v = \text{MLP}(\mathbf{p}_v^{\text{Personal}} \,||\, \mathbf{p}_v^{\text{Temporal}}), \tag{3}$$

where $\text{MLP}(\cdot)$ (Bishop & Nasrabadi, 2006) is introduced to combine two types of information. The Projection TProG can be seen as a middle ground between the Vanilla TProG and the Transformer TProG, as it utilizes a learnable prompt vector to represent interactive information and a temporal vector to mimic the temporal evolution. Like the Vanilla TProG, the number of tunable parameters required for the Projection TProG is $\mathcal{O}(|\mathcal{V}|)$, scaling linearly with the size of the graph.

### 3.3 PROMPT TUNING AND INFERENCE

Recall in Equ. 1, a fusion function is introduced to combine pre-trained node embeddings $\mathbf{Z}$ and prompt matrix $\mathbf{P}$ to yield prompted node representations. Specifically, we implement $f_\rho(\cdot)$ via a MLP parameterized by $\rho$ as:

$$\widetilde{\mathbf{Z}} = f_\rho(\mathbf{Z}, \mathbf{P}) = \text{MLP}_\rho(\mathbf{Z} \,||\, \mathbf{P}), \tag{4}$$

where $\widetilde{\mathbf{Z}}$ can be regarded as prompted embeddings, incorporating temporal knowledge to adapt to specific downstream tasks.

Take the downstream link prediction task as an example, suppose a TIG has edge set $\mathcal{E}$, which can be split into three disjoint sets as $\mathcal{E} = \mathcal{E}^{\text{pre-train}} \cup \mathcal{E}^{\text{prompt}} \cup \mathcal{E}^{\text{val/test}}$. Here, $\mathcal{E}^{\text{pre-train}}$ denotes the

set of edges used for pre-training the TIG model $f_\Theta(\cdot)$, $\mathcal{E}^{\text{prompt}}$ represents the set used to tune the prompt generator, and $\mathcal{E}^{\text{val/test}}$ denotes the edges for validation or testing. Specifically, given $\mathcal{E}^{\text{prompt}}$, the TProG is optimized using predictions and ground-truth labels: $\mathcal{L}_{\text{prompt-tune}}(\Phi, \rho, \mathbf{P}) = \text{Cross-Entropy}(p_\Phi(f_\rho(\mathbf{Z}, \mathbf{P})), \mathbf{Y}^{\text{prompt}})$, where $\mathbf{Y}^{\text{prompt}}$ denotes the ground-truth labels provided by $\mathcal{E}^{\text{prompt}}$, $p_\Phi(\cdot)$ denotes the projection head of the link prediction task. Notably, during the prompt tuning stage, the TIG model remains frozen, avoiding exhaustive re-training processes. The tuning data only constitutes a small portion, meaning that even a small number of samples can help improve the adaptation of the pre-trained TIG model to downstream predictions. Similarly, the downstream node classification task can provide a small number of samples to tune TProG and generate meaningful $\mathbf{P}$. Once TProG is well-tuned, downstream predictions can be made as $\hat{\mathbf{Y}} = p_\Phi(f_\rho(\mathbf{Z}, \mathbf{P}))$. By leveraging task-specific supervision to tune TProG, the prompts can incorporate task-specific semantics. This tuning process helps bridge both semantic and temporal gaps, resulting in improved downstream predictions.

**Extension: "Pre-train, Prompt-based Fine-tune" Paradigm.** To accommodate to diverse computational resource requirements, we extend the proposed "pre-train, prompt" paradigm to the "pre-train, prompt-based fine-tune" paradigm. The main difference between these two modes lies in whether the parameters of TIG model $f_\Theta(\cdot)$ is tuned during the prompt tuning stage. Therefore, for this paradigm, given prompt samples, both the prompts and the TIG model are optimized concurrently as: $\mathcal{L}_{\text{fine-tune}}(\Phi, \rho, \mathbf{P}, \Theta) = \text{Cross-Entropy}(p_\Phi(f_{\rho, \Theta}(\mathbf{Z}, \mathbf{P})), \mathbf{Y}^{\text{prompt}})$. By jointly optimizing the TIG model and the prompts, these two components reinforce each other, leading to improved adaptability in various scenarios.

### 3.4 CONNECTION TO EXISTING GRAPH PROMPTING APPROACHES

Various prompting methods have been developed for static graphs (Please refer to Appendix B Related Work for more details). Most of these methods are specifically designed for a range of downstream tasks unique to static graph contexts. Among these methods, GraphPrompt (Liu et al., 2023b) and GPF (Fang et al., 2023) stand out as representatives and amenable to adaptation for the TIG model. GraphPrompt (Liu et al., 2023b) utilizes a prompt vector on the outputted embeddings of GNN models, whereas GPF (Fang et al., 2023) employs a similar prompt vector on the input data features. Therefore, in Sec. 4.5 we transfer these ideas to the TIG model and conduct experiments to see the comparable performance with our temporal graph prompting approach.

## 4 EXPERIMENTS

### 4.1 DATASETS AND BASELINES

We apply the proposed TIGPrompt on four public datasets, Wikipedia, Reddit, MOOC and LastFM (Kumar et al., 2019). Detailed statistics of these datasets are presented in Appendix C (Tab. 5). Only Wikipedia, Reddit and MOOC are with dynamic labels indicating state changes of users. For datasets missing node or edge features, we adopt the approach used in prior works (Rossi et al., 2020; Zhang et al., 2023c), representing them with zero vectors.

For baseline comparisons, we select representative TGN-based methods[1], including Jodie (Kumar et al., 2019), DyRep (Trivedi et al., 2019), TGN (Rossi et al., 2020) and TIGE (Zhang et al., 2023c). Additionally, we include TIGER-T (Zhang et al., 2023c) as a baseline, considering it is a variant of TIGE and potentially offers improved performance over the TIGE model. We also compare our method with GraphMixer (Cong et al., 2023) and DyGFormer (Yu et al., 2023), which employ different model architectures, with a detailed discussion provided in Appendix E.2.

### 4.2 EXPERIMENTAL SETTINGS

Our implementation and hyper-parameter settings are consistent with those in previous works (Rossi et al., 2020; Zhang et al., 2023c). More information is discussed in Appendix J. Typically, the chosen baseline models split interaction edges chronologically into 70% for training, 15% for validation, and 15% for testing. However, as discussed in Sec. 3, our aim is to demonstrate our method's adeptness

---

[1]These methods can be integrated into a unified framework based on TGN (Rossi et al., 2020).

Table 1: Under the "pre-train, prompt" paradigm, results for the link prediction task — encompassing both transductive and inductive settings — are presented using Average Precision (%). For the dynamic node classification task, results are measured in terms of AUROC (%). The best performance is highlighted in **bold**.

| | TProG | Transductive Link Prediction | | | | Inductive Link Prediction | | | | Node Classification | | |
|---|---|---|---|---|---|---|---|---|---|---|---|---|
| | | Wikipedia | Reddit | MOOC | LastFM | Wikipedia | Reddit | MOOC | LastFM | Wikipedia | Reddit | MOOC |
| Jodie | Baseline | $94.62_{\pm0.5}$ | $97.11_{\pm0.3}$ | $76.50_{\pm1.8}$ | $68.77_{\pm3.0}$ | $93.11_{\pm0.4}$ | $94.36_{\pm1.1}$ | $77.83_{\pm2.1}$ | $82.55_{\pm1.9}$ | $86.27_{\pm2.2}$ | $58.48_{\pm2.6}$ | $65.39_{\pm1.1}$ |
| | Vanilla | $94.10_{\pm0.4}$ | $97.65_{\pm0.0}$ | $74.47_{\pm0.9}$ | $74.15_{\pm1.0}$ | $91.43_{\pm0.3}$ | $93.07_{\pm0.4}$ | $72.23_{\pm1.4}$ | $79.42_{\pm1.1}$ | $86.79_{\pm2.1}$ | $\mathbf{69.22_{\pm0.4}}$ | $69.21_{\pm0.4}$ |
| | Transformer | $\mathbf{96.50_{\pm0.1}}$ | $98.28_{\pm0.0}$ | $\mathbf{82.90_{\pm1.1}}$ | $77.98_{\pm2.1}$ | $\mathbf{95.08_{\pm0.2}}$ | $95.68_{\pm0.1}$ | $79.81_{\pm1.2}$ | $85.72_{\pm0.9}$ | $80.91_{\pm6.7}$ | $63.80_{\pm2.2}$ | $70.67_{\pm1.1}$ |
| | Projection | $96.44_{\pm0.3}$ | $\mathbf{98.99_{\pm0.0}}$ | $82.47_{\pm0.9}$ | $\mathbf{89.39_{\pm0.7}}$ | $94.75_{\pm0.5}$ | $\mathbf{97.43_{\pm0.1}}$ | $\mathbf{79.89_{\pm1.2}}$ | $\mathbf{92.72_{\pm0.4}}$ | $\mathbf{87.08_{\pm1.1}}$ | $68.26_{\pm0.9}$ | $\mathbf{76.45_{\pm0.6}}$ |
| DyRep | Baseline | $94.59_{\pm0.2}$ | $97.98_{\pm0.1}$ | $75.37_{\pm1.7}$ | $68.77_{\pm2.1}$ | $92.05_{\pm0.3}$ | $95.68_{\pm0.2}$ | $78.55_{\pm1.1}$ | $81.33_{\pm2.1}$ | $85.11_{\pm1.4}$ | $62.77_{\pm2.1}$ | $66.68_{\pm3.4}$ |
| | Vanilla | $89.64_{\pm1.0}$ | $97.63_{\pm0.0}$ | $71.57_{\pm2.7}$ | $72.62_{\pm1.1}$ | $85.45_{\pm1.2}$ | $92.92_{\pm0.3}$ | $71.34_{\pm0.5}$ | $77.48_{\pm1.7}$ | $84.88_{\pm1.4}$ | $\mathbf{65.67_{\pm2.4}}$ | $68.38_{\pm0.9}$ |
| | Transformer | $94.51_{\pm0.4}$ | $98.27_{\pm0.0}$ | $\mathbf{80.59_{\pm1.9}}$ | $76.89_{\pm1.6}$ | $92.44_{\pm0.4}$ | $95.73_{\pm0.1}$ | $\mathbf{78.89_{\pm0.2}}$ | $84.81_{\pm3.0}$ | $60.87_{\pm3.8}$ | $58.20_{\pm2.3}$ | $70.80_{\pm0.9}$ |
| | Projection | $\mathbf{96.87_{\pm0.2}}$ | $\mathbf{99.06_{\pm0.0}}$ | $79.76_{\pm1.9}$ | $\mathbf{89.04_{\pm0.6}}$ | $\mathbf{95.37_{\pm0.3}}$ | $\mathbf{97.48_{\pm0.0}}$ | $78.56_{\pm0.7}$ | $\mathbf{92.58_{\pm0.4}}$ | $\mathbf{85.25_{\pm1.3}}$ | $64.50_{\pm1.5}$ | $\mathbf{76.06_{\pm0.9}}$ |
| TGN | Baseline | $\mathbf{98.46_{\pm0.1}}$ | $98.70_{\pm0.1}$ | $85.88_{\pm3.0}$ | $71.76_{\pm5.3}$ | $\mathbf{97.81_{\pm0.1}}$ | $97.55_{\pm0.1}$ | $\mathbf{85.55_{\pm2.9}}$ | $80.42_{\pm4.9}$ | $84.93_{\pm1.1}$ | $65.99_{\pm3.8}$ | $69.80_{\pm1.8}$ |
| | Vanilla | $96.40_{\pm0.2}$ | $98.36_{\pm0.0}$ | $86.71_{\pm1.0}$ | $79.67_{\pm1.7}$ | $95.02_{\pm0.2}$ | $95.54_{\pm0.2}$ | $81.99_{\pm1.2}$ | $83.76_{\pm1.3}$ | $85.79_{\pm1.1}$ | $\mathbf{66.13_{\pm1.3}}$ | $70.16_{\pm1.9}$ |
| | Transformer | $97.36_{\pm0.3}$ | $98.67_{\pm0.0}$ | $89.21_{\pm0.7}$ | $81.63_{\pm0.6}$ | $96.19_{\pm0.4}$ | $96.68_{\pm0.2}$ | $83.35_{\pm0.9}$ | $84.82_{\pm1.2}$ | $86.39_{\pm1.8}$ | $64.89_{\pm1.1}$ | $71.13_{\pm1.4}$ |
| | Projection | $97.83_{\pm0.1}$ | $\mathbf{99.29_{\pm0.0}}$ | $\mathbf{89.28_{\pm0.8}}$ | $\mathbf{91.85_{\pm0.3}}$ | $96.79_{\pm0.2}$ | $\mathbf{98.14_{\pm0.1}}$ | $84.49_{\pm1.0}$ | $\mathbf{93.17_{\pm0.7}}$ | $\mathbf{87.09_{\pm0.4}}$ | $66.07_{\pm1.5}$ | $\mathbf{73.44_{\pm1.4}}$ |
| TIGE | Baseline | $98.83_{\pm0.1}$ | $99.04_{\pm0.0}$ | $89.64_{\pm0.9}$ | $87.85_{\pm0.9}$ | $98.45_{\pm0.1}$ | $98.39_{\pm0.1}$ | $89.51_{\pm0.7}$ | $90.14_{\pm1.0}$ | $83.98_{\pm3.4}$ | $\mathbf{65.36_{\pm2.9}}$ | $69.61_{\pm2.5}$ |
| | Vanilla | $98.75_{\pm0.0}$ | $98.88_{\pm0.0}$ | $88.91_{\pm0.4}$ | $89.54_{\pm0.3}$ | $98.22_{\pm0.0}$ | $97.73_{\pm0.0}$ | $88.22_{\pm0.3}$ | $90.78_{\pm0.0}$ | $86.18_{\pm0.5}$ | $62.13_{\pm2.0}$ | $70.57_{\pm1.1}$ |
| | Transformer | $98.95_{\pm0.0}$ | $99.25_{\pm0.0}$ | $\mathbf{91.10_{\pm0.4}}$ | $90.65_{\pm0.3}$ | $98.52_{\pm0.1}$ | $98.68_{\pm0.0}$ | $88.82_{\pm0.9}$ | $91.71_{\pm0.2}$ | $82.02_{\pm7.0}$ | $61.41_{\pm2.6}$ | $71.44_{\pm0.6}$ |
| | Projection | $\mathbf{99.10_{\pm0.1}}$ | $\mathbf{99.47_{\pm0.0}}$ | $90.94_{\pm0.2}$ | $\mathbf{95.21_{\pm0.2}}$ | $\mathbf{98.75_{\pm0.1}}$ | $\mathbf{99.07_{\pm0.0}}$ | $\mathbf{89.61_{\pm0.4}}$ | $\mathbf{95.81_{\pm0.1}}$ | $\mathbf{86.65_{\pm0.9}}$ | $60.75_{\pm1.3}$ | $\mathbf{75.18_{\pm2.1}}$ |
| TIGER | Baseline | $98.90_{\pm0.0}$ | $99.02_{\pm0.0}$ | $86.99_{\pm1.6}$ | $85.17_{\pm0.2}$ | $98.58_{\pm0.0}$ | $98.59_{\pm0.0}$ | $86.42_{\pm1.7}$ | $89.11_{\pm0.3}$ | $80.84_{\pm4.6}$ | $62.58_{\pm1.3}$ | $64.91_{\pm5.2}$ |
| | Vanilla | $98.89_{\pm0.0}$ | $98.90_{\pm0.0}$ | $87.43_{\pm0.4}$ | $86.13_{\pm0.4}$ | $98.50_{\pm0.0}$ | $98.33_{\pm0.0}$ | $87.28_{\pm1.5}$ | $88.18_{\pm0.5}$ | $85.12_{\pm0.3}$ | $\mathbf{63.16_{\pm1.4}}$ | $68.68_{\pm1.9}$ |
| | Transformer | $98.98_{\pm0.0}$ | $99.22_{\pm0.0}$ | $\mathbf{90.31_{\pm0.4}}$ | $88.22_{\pm0.4}$ | $98.59_{\pm0.0}$ | $98.88_{\pm0.0}$ | $89.05_{\pm1.0}$ | $90.69_{\pm0.4}$ | $77.15_{\pm8.9}$ | $61.94_{\pm2.1}$ | $71.26_{\pm1.2}$ |
| | Projection | $\mathbf{99.16_{\pm0.0}}$ | $\mathbf{99.49_{\pm0.0}}$ | $89.74_{\pm0.5}$ | $\mathbf{93.73_{\pm0.2}}$ | $\mathbf{98.89_{\pm0.0}}$ | $\mathbf{99.26_{\pm0.0}}$ | $\mathbf{89.42_{\pm1.5}}$ | $\mathbf{95.07_{\pm0.3}}$ | $\mathbf{86.30_{\pm0.8}}$ | $62.75_{\pm1.5}$ | $\mathbf{74.07_{\pm0.5}}$ |
| GraphMixer | Baseline | $97.25_{\pm0.0}$ | $97.31_{\pm0.0}$ | $82.78_{\pm0.2}$ | $75.61_{\pm0.2}$ | $96.65_{\pm0.0}$ | $95.26_{\pm0.0}$ | $81.41_{\pm0.2}$ | $82.11_{\pm0.4}$ | $86.80_{\pm0.8}$ | $64.22_{\pm3.3}$ | $69.42_{\pm0.8}$ |
| | Vanilla | $96.12_{\pm0.0}$ | $92.95_{\pm0.3}$ | $80.86_{\pm0.6}$ | $76.57_{\pm1.5}$ | $95.56_{\pm0.0}$ | $94.33_{\pm0.2}$ | $78.28_{\pm1.1}$ | $75.73_{\pm2.5}$ | $\mathbf{89.00_{\pm0.0}}$ | $\mathbf{69.77_{\pm0.8}}$ | $70.91_{\pm0.5}$ |
| | Transformer | $97.39_{\pm0.0}$ | $98.28_{\pm0.0}$ | $84.44_{\pm0.4}$ | $79.28_{\pm0.1}$ | $96.98_{\pm0.1}$ | $\mathbf{96.67_{\pm0.0}}$ | $82.14_{\pm0.8}$ | $84.39_{\pm0.2}$ | $88.24_{\pm0.1}$ | $68.82_{\pm2.9}$ | $\mathbf{71.69_{\pm0.8}}$ |
| | Projection | $\mathbf{99.33_{\pm0.0}}$ | $\mathbf{99.18_{\pm0.1}}$ | $\mathbf{87.42_{\pm0.4}}$ | $\mathbf{89.53_{\pm3.1}}$ | $\mathbf{97.89_{\pm0.4}}$ | $94.64_{\pm0.3}$ | $\mathbf{84.00_{\pm0.5}}$ | $\mathbf{87.87_{\pm1.3}}$ | $87.92_{\pm0.8}$ | $67.73_{\pm0.7}$ | $71.60_{\pm0.5}$ |
| DyGFormer | Baseline | $99.03_{\pm0.0}$ | $99.22_{\pm0.0}$ | $87.52_{\pm0.5}$ | $93.00_{\pm0.1}$ | $98.59_{\pm0.0}$ | $98.84_{\pm0.0}$ | $86.96_{\pm0.4}$ | $94.23_{\pm0.1}$ | $87.44_{\pm1.1}$ | $\mathbf{68.00_{\pm1.7}}$ | $78.37_{\pm0.6}$ |
| | Vanilla | $98.98_{\pm0.1}$ | $99.20_{\pm0.0}$ | $84.96_{\pm0.8}$ | $92.85_{\pm0.1}$ | $98.63_{\pm0.1}$ | $98.85_{\pm0.1}$ | $84.50_{\pm0.4}$ | $94.19_{\pm0.0}$ | $\mathbf{88.92_{\pm1.5}}$ | $62.95_{\pm1.6}$ | $75.91_{\pm0.6}$ |
| | Transformer | $99.11_{\pm0.1}$ | $99.53_{\pm0.0}$ | $88.05_{\pm0.3}$ | $94.14_{\pm0.1}$ | $98.88_{\pm0.2}$ | $\mathbf{99.17_{\pm0.0}}$ | $87.05_{\pm0.6}$ | $95.08_{\pm0.1}$ | $86.93_{\pm1.0}$ | $66.57_{\pm2.6}$ | $\mathbf{78.56_{\pm0.5}}$ |
| | Projection | $\mathbf{99.80_{\pm0.0}}$ | $\mathbf{99.87_{\pm0.0}}$ | $\mathbf{90.60_{\pm0.4}}$ | $\mathbf{95.04_{\pm0.7}}$ | $\mathbf{99.32_{\pm0.1}}$ | $98.82_{\pm0.2}$ | $\mathbf{87.91_{\pm0.6}}$ | $\mathbf{95.49_{\pm0.4}}$ | $88.14_{\pm0.7}$ | $65.03_{\pm2.8}$ | $76.06_{\pm0.4}$ |

in adapting to emerging data. For a fair comparison, we utilize the same data portion for training and inference. Therefore, we use only 50% of the data for pre-training and 20% for prompt tuning or fine-tuning, with the remaining 30% equally divided for validation and testing. In essence, we train our model with less data and leverage a smaller portion for prompt tuning or fine-tuning to achieve enhanced performance on downstream tasks compared to the baselines. The data amount used for experiments is detailed summarized in the Appendix G.

### 4.3 "Pre-train, Prompt"

**Link Prediction.** In the initial set of experiments, we keep the established protocols (Rossi et al., 2020; Zhang et al., 2023c) to assess model performance in both transductive and inductive temporal link prediction tasks. In the transductive setting, we focus on those edges linked to nodes previously encountered in the training dataset. Conversely, in the inductive setting, the predictions center on temporal links between nodes that are unseen during the training phase. The evaluation metric is the average precision (AP) score. We discuss the TGN-based methods in this section, while the results of GraphMixer and DyGFormer will be discussed in the Appendix E.2.

Adhering to the proposed "pre-train, prompt" training paradigm, we keep the pre-trained model's parameters frozen during prompt tuning phrase. As illustrated in Tab. 1, the experiments utilize three distinct proposed TProGs, respectively. Notably, the integration of prompts generated by the proposed TProGs with the original node representations results in significant improvements in downstream tasks. This approach yields SOTA results across nearly all datasets and baselines. This effectiveness stems from the fact that the prompts generated by the proposed TProG comprehensively incorporate temporal information, thereby bridging the temporal gap between pre-training and downstream task data. Particularly for the LastFM dataset, where performance is previously sub-optimal, our method enhances performance by 29% compared to prior approaches, as evidenced on two baselines. The fact that only a small portion of the data is used for prompt tuning underscores the efficacy of our methods, particularly the Transformer TProG and Projection TProG, in facilitating model adaptation to evolving timely TIG data. However, in certain specific dataset/model combinations, such as Wikipedia/TGN, our model does not surpass the baseline. This limitation arises because

Table 2: The results for the link prediction task under the "pre-train, prompt" paradigm, note that **only 20% of data is used** in total (10% for pre-train, 10% for fine-tune). Results colored in blue indicate that they **even surpass** the baseline achieved with **70% of the data used** for training.

| Only 20% of data used | | Transductive Link Prediction | | | | Inductive Link Prediction | | | |
|---|---|---|---|---|---|---|---|---|---|
| | TProG | Wikipedia | Reddit | MOOC | LastFM | Wikipedia | Reddit | MOOC | LastFM |
| Jodie | Baseline | $79.28_{\pm4.2}$ | $92.39_{\pm1.4}$ | $55.73_{\pm2.2}$ | $68.00_{\pm0.7}$ | $79.30_{\pm4.8}$ | $80.58_{\pm2.8}$ | $58.51_{\pm2.6}$ | $80.96_{\pm1.3}$ |
| | Vanilla | $89.17_{\pm0.4}$ | $96.39_{\pm0.1}$ | $63.10_{\pm0.2}$ | $72.57_{\pm1.0}$ | $88.00_{\pm0.6}$ | $94.33_{\pm0.1}$ | $63.52_{\pm0.3}$ | $77.13_{\pm0.9}$ |
| | Transformer | $92.11_{\pm0.9}$ | $97.54_{\pm0.0}$ | $72.98_{\pm0.3}$ | $77.99_{\pm0.6}$ | $92.34_{\pm0.7}$ | $96.43_{\pm0.0}$ | $73.25_{\pm0.3}$ | $81.63_{\pm0.8}$ |
| | Projection | $\mathbf{95.64}_{\pm0.3}$ | $\mathbf{98.54}_{\pm0.1}$ | $\mathbf{76.23}_{\pm0.3}$ | $\mathbf{89.21}_{\pm0.1}$ | $\mathbf{95.04}_{\pm0.2}$ | $\mathbf{97.71}_{\pm0.1}$ | $\mathbf{76.31}_{\pm0.3}$ | $\mathbf{90.94}_{\pm0.2}$ |
| DyRep | Baseline | $88.19_{\pm1.0}$ | $96.82_{\pm0.3}$ | $73.13_{\pm1.7}$ | $67.38_{\pm1.1}$ | $85.99_{\pm0.9}$ | $92.01_{\pm0.8}$ | $71.91_{\pm2.1}$ | $79.67_{\pm1.8}$ |
| | Vanilla | $84.27_{\pm1.2}$ | $96.35_{\pm0.1}$ | $61.19_{\pm1.3}$ | $69.85_{\pm0.5}$ | $83.93_{\pm0.9}$ | $93.82_{\pm0.3}$ | $61.42_{\pm1.5}$ | $75.50_{\pm0.2}$ |
| | Transformer | $91.68_{\pm0.4}$ | $97.40_{\pm0.1}$ | $72.44_{\pm1.0}$ | $74.78_{\pm0.4}$ | $91.23_{\pm0.5}$ | $96.28_{\pm0.2}$ | $72.75_{\pm1.0}$ | $80.07_{\pm0.2}$ |
| | Projection | $\mathbf{95.74}_{\pm0.2}$ | $\mathbf{98.63}_{\pm0.0}$ | $\mathbf{76.40}_{\pm0.2}$ | $\mathbf{88.26}_{\pm0.2}$ | $\mathbf{95.40}_{\pm0.2}$ | $\mathbf{97.74}_{\pm0.1}$ | $\mathbf{76.40}_{\pm0.3}$ | $\mathbf{90.51}_{\pm0.1}$ |
| TGN | Baseline | $96.34_{\pm0.2}$ | $97.63_{\pm0.1}$ | $56.54_{\pm0.5}$ | $66.54_{\pm2.0}$ | $95.86_{\pm0.3}$ | $95.98_{\pm0.4}$ | $61.11_{\pm0.9}$ | $75.09_{\pm2.8}$ |
| | Vanilla | $95.59_{\pm0.1}$ | $97.63_{\pm0.1}$ | $74.30_{\pm1.2}$ | $64.36_{\pm2.0}$ | $95.27_{\pm0.2}$ | $96.32_{\pm0.2}$ | $74.58_{\pm1.1}$ | $67.92_{\pm1.5}$ |
| | Transformer | $96.23_{\pm0.1}$ | $98.09_{\pm0.0}$ | $75.15_{\pm0.8}$ | $67.65_{\pm3.0}$ | $95.79_{\pm0.1}$ | $97.35_{\pm0.1}$ | $75.25_{\pm0.7}$ | $70.43_{\pm3.4}$ |
| | Projection | $\mathbf{96.93}_{\pm0.2}$ | $\mathbf{98.95}_{\pm0.0}$ | $\mathbf{79.10}_{\pm0.6}$ | $\mathbf{87.42}_{\pm0.4}$ | $\mathbf{96.58}_{\pm0.3}$ | $\mathbf{98.38}_{\pm0.1}$ | $\mathbf{79.17}_{\pm0.5}$ | $\mathbf{88.65}_{\pm0.6}$ |
| TIGE | Baseline | $98.36_{\pm0.1}$ | $98.71_{\pm0.1}$ | $80.60_{\pm1.5}$ | $84.73_{\pm0.7}$ | $98.11_{\pm0.1}$ | $98.46_{\pm0.1}$ | $80.71_{\pm1.4}$ | $85.73_{\pm0.8}$ |
| | Vanilla | $98.50_{\pm0.0}$ | $98.58_{\pm0.0}$ | $80.58_{\pm0.4}$ | $85.24_{\pm0.3}$ | $98.20_{\pm0.0}$ | $98.16_{\pm0.0}$ | $80.88_{\pm0.3}$ | $86.45_{\pm0.0}$ |
| | Transformer | $98.92_{\pm0.0}$ | $99.08_{\pm0.0}$ | $80.32_{\pm1.2}$ | $87.77_{\pm0.4}$ | $98.69_{\pm0.0}$ | $98.90_{\pm0.0}$ | $80.56_{\pm1.1}$ | $88.81_{\pm0.3}$ |
| | Projection | $98.82_{\pm0.0}$ | $99.32_{\pm0.0}$ | $\mathbf{83.11}_{\pm0.1}$ | $93.40_{\pm0.2}$ | $98.63_{\pm0.0}$ | $99.16_{\pm0.0}$ | $83.30_{\pm0.1}$ | $93.94_{\pm0.1}$ |
| TIGER | Baseline | $98.32_{\pm0.1}$ | $98.67_{\pm0.1}$ | $80.31_{\pm0.6}$ | $84.53_{\pm0.4}$ | $98.10_{\pm0.1}$ | $98.12_{\pm0.2}$ | $78.07_{\pm0.5}$ | $88.54_{\pm0.5}$ |
| | Vanilla | $98.50_{\pm0.0}$ | $98.62_{\pm0.0}$ | $80.47_{\pm0.3}$ | $84.66_{\pm0.1}$ | $98.22_{\pm0.0}$ | $98.31_{\pm0.0}$ | $80.88_{\pm0.3}$ | $86.05_{\pm0.3}$ |
| | Transformer | $98.93_{\pm0.0}$ | $99.04_{\pm0.0}$ | $80.66_{\pm0.9}$ | $88.18_{\pm0.2}$ | $98.67_{\pm0.0}$ | $98.85_{\pm0.0}$ | $80.90_{\pm0.9}$ | $89.36_{\pm0.2}$ |
| | Projection | $98.77_{\pm0.0}$ | $99.35_{\pm0.0}$ | $\mathbf{82.96}_{\pm0.3}$ | $93.10_{\pm0.1}$ | $98.57_{\pm0.0}$ | $99.23_{\pm0.0}$ | $83.16_{\pm0.3}$ | $93.73_{\pm0.0}$ |
| GraphMixer | Baseline | $95.88_{\pm0.1}$ | $96.51_{\pm0.0}$ | $75.65_{\pm1.5}$ | $74.14_{\pm0.4}$ | $95.61_{\pm0.0}$ | $94.43_{\pm0.0}$ | $74.10_{\pm1.6}$ | $80.84_{\pm0.6}$ |
| | Vanilla | $94.41_{\pm0.1}$ | $96.32_{\pm0.1}$ | $73.34_{\pm2.2}$ | $77.30_{\pm0.2}$ | $93.80_{\pm0.1}$ | $94.77_{\pm0.1}$ | $73.28_{\pm2.2}$ | $80.48_{\pm0.4}$ |
| | Transformer | $95.55_{\pm0.2}$ | $97.48_{\pm0.1}$ | $82.71_{\pm0.8}$ | $78.39_{\pm0.1}$ | $95.30_{\pm0.1}$ | $96.71_{\pm0.1}$ | $82.67_{\pm0.8}$ | $81.02_{\pm0.1}$ |
| | Projection | $\mathbf{98.80}_{\pm0.2}$ | $\mathbf{98.91}_{\pm0.1}$ | $\mathbf{87.05}_{\pm2.0}$ | $\mathbf{83.67}_{\pm3.5}$ | $\mathbf{97.44}_{\pm0.5}$ | $96.13_{\pm0.3}$ | $\mathbf{86.68}_{\pm1.9}$ | $\mathbf{84.72}_{\pm2.6}$ |
| DyGFormer | Baseline | $98.84_{\pm0.0}$ | $98.91_{\pm0.0}$ | $77.52_{\pm1.3}$ | $92.02_{\pm0.0}$ | $98.40_{\pm0.0}$ | $98.46_{\pm0.1}$ | $74.45_{\pm1.2}$ | $93.48_{\pm0.0}$ |
| | Vanilla | $98.57_{\pm0.0}$ | $98.60_{\pm0.1}$ | $75.69_{\pm4.2}$ | $91.04_{\pm0.2}$ | $98.28_{\pm0.0}$ | $98.42_{\pm0.1}$ | $75.85_{\pm4.1}$ | $91.85_{\pm0.2}$ |
| | Transformer | $98.80_{\pm0.0}$ | $99.20_{\pm0.1}$ | $82.31_{\pm1.8}$ | $\mathbf{92.74}_{\pm0.2}$ | $98.62_{\pm0.0}$ | $99.04_{\pm0.1}$ | $82.46_{\pm1.9}$ | $\mathbf{93.56}_{\pm0.2}$ |
| | Projection | $\mathbf{99.75}_{\pm0.1}$ | $\mathbf{99.76\pm0.0}$ | $\mathbf{87.77}_{\pm1.9}$ | $92.47_{\pm0.2}$ | $\mathbf{99.47}_{\pm0.1}$ | $98.82_{\pm0.5}$ | $\mathbf{87.46}_{\pm1.9}$ | $92.85_{\pm0.3}$ |

breaking down the temporal gap in these contexts adversely affects the results. However, in the node classification experiments discussed subsequently, both temporal and semantic gaps exist between the pretext and downstream tasks. In these cases, our model achieves superior performance, indicating that in such scenarios, the semantic gap predominates as the primary limiting factor for the performance of the TIG model.

**Node Classification.** Dynamic node classification is conducted aiming to predict dynamic labels of nodes. It is utilized as a downstream task to validate the prompt's effectiveness and to demonstrate how the proposed method effectively bridges the temporal and semantic gap between pretext and downstream tasks. We conduct dynamic node classification on datasets with dynamic labels, i.e., Wikipedia, Reddit, and MOOC.

We use the same pre-trained models as in the link prediction task. The TProGs, however, are exclusively initialized and trained during the node classification process. Following the approach in (Xu et al., 2020; Rossi et al., 2020; Zhang et al., 2023c), we pass time-aware representations through a two-layer MLP to determine the probabilities of dynamic labels. However, these time-aware representations are substituted with prompted node representations, generated by the TProG. In the original methodology, validation, and testing phases are not distinct, with the last epoch's results under a fixed maximum number of epochs being directly used for testing. To incorporate TProG training into this process, we adapt the validation and testing phases to mirror the link prediction task approach, allocating 15% of the data for validation and another 15% for testing. Concurrently, the baseline settings align with those used here.

As the results shown in Tab. 1, our method significantly outperforms the baseline on almost all node classification tasks, achieving the SOTA performance. It is worth noting that on the Reddit dataset, the Vanilla TProG alone is sufficient to achieve superior results. At the same time, the Projection TProG not only surpasses the baseline on Reddit but also shows the best performance on the other two datasets. For the MOOC dataset, our method improves upon the DyRep baseline by 15%. These results demonstrate the substantial impact of the proposed training paradigm in bridging the gap between pretext and downstream tasks.

## 4.4 "PRE-TRAIN, PROMPT-BASED FINE-TUNE"

In the "pre-train, prompt-based fine-tune" paradigm, we follow a similar experimental setting as with "pre-train, prompt", with a key difference: instead of freezing the pre-trained model's parameters, we allow for their simultaneous optimization while using 20% of the data to train the TProG. This adjustment aims to enhance the model's adaptability to new data and downstream tasks. The full experimental results are shown in Appendix D. As shown in Tab. 6, this paradigm yields improved results compared to "pre-train, prompt" on link prediction task, attributable to the fine-tuning of the pre-trained model. However, this approach requires more training resources due to the optimization of the pre-trained model's parameters. Thus, this training paradigm is recommended when sufficient resources are available to achieve optimal results. More details of node classification task are discussed in Appendix D.2.

## 4.5 COMPARISON WITH EXISTING GRAPH PROMPTS

As discussed in Sec. 3.4, we conduct experiments using prompts from static graphs, i.e., GraphPrompt (Liu et al., 2023b) and GPF (Fang et al., 2023), where a single, learnable prompt vector is applied uniformly across all nodes, either on the input (Fang et al., 2023) or on the output (Liu et al., 2023b) embeddings. The comparative results of these experiments are depicted in Fig. 4. The results demonstrate that our method significantly outperforms the traditional prompt method used in static graphs, demonstrating our effectiveness once again.

## 4.6 EFFECTIVENESS OF VARIOUS TPROGS

As indicated in Tab. 1 and 6, the Projection TProG generally outperforms other types of TProG in link prediction tasks, with the Transformer TProG also excelling in certain scenarios. In contrast, the Vanilla TProG often shows weaker performance, likely due to its limited capacity to express temporal information. However, in node classification tasks, the Vanilla TProG demonstrates improved results on specific datasets. Meanwhile, the Projection TProG consistently surpasses the baseline, though the Transformer TProG shows slightly lower effectiveness.

The Transformer TProG captures recent behavior patterns, whereas the Projection TProG emphasizes the global historical state. The scenarios where the Transformer TProG demonstrates superior performance are predominantly observed on the MOOC dataset. This suggests that the recent behavioral characteristics inherent to this dataset are particularly effective in bridging the existing gaps. The robust performance of the Projection TProG across various tasks can be ascribed to its ability to model global historical information, which possesses significant expressive power for capturing temporal dynamics in TIGs. Additionally, its node-specific learnable embeddings play a pivotal role in effectively bridging the semantic gap between pretext and downstream tasks.

Although the Transformer and Projection TProG generally exhibit stronger temporal expressiveness, there remain cases, particularly in node classification task, where the simpler Vanilla TProG performs competitively or even slightly better. This phenomenon is consistent with the distinct nature of node classification, which typically depends more on semantic alignment than on detailed temporal dynamics. As analyzed earlier, the semantic gap arising from the mismatch between link-level pretext training and node-level downstream objectives often becomes the primary bottleneck for node classification. The Vanilla TProG introduces node-specific learnable embeddings that directly encode task-relevant semantic information without additional temporal modeling. In datasets such as Reddit, where interactions are dense and long-term temporal dependencies are relatively weak, this lightweight semantic adaptation proves particularly effective, leading to performance that rivals or occasionally surpasses more expressive variants.

**Source of the Performance Improving.** As shown in Tab. 1, the Vanilla TProG, without using the temporal information, generally exhibits inferior performance in link prediction tasks compared to the Transformer and Projection TProG, both of which incorporate time-related prompts. This demonstrates that adding time-related information contributes to performance enhancement. Furthermore, our comparison with static graph methods in Sec. 4.5, indirectly corroborates that the observed improvements are attributable to the proposed TProG.

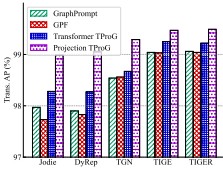 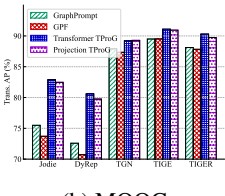 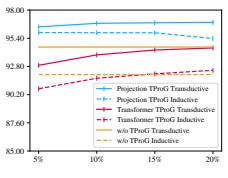 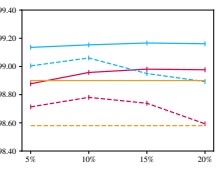

(a) Reddit (b) MOOC (a) Wikipedia/DyRep (b) Wikipedia/TIGER

Figure 4: Comparison between traditional prompt on static graphs (Liu et al., 2023b; Fang et al., 2023) and our methods ("pre-train, prompt" paradigm, transductive link prediction on Reddit and MOOC).

Figure 5: Performance w.r.t the Proportion of Prompting Data. This figure is continued in Appendix E.1, Fig. 7.

### 4.7 PERFORMANCE WITH LIMITED DATA

#### 4.7.1 PERFORMANCE WITH LIMITED TRAINING DATA

To validate the effectiveness of the proposed prompt method and demonstrate that it requires only a small dataset to achieve superior results, we strategically design an experiment using merely 10% of the data for pre-training, followed by another 10% for prompt tuning ("pre-train, prompt"). As a baseline for comparison, we utilized the results reported in TIGE (Zhang et al., 2023c), which is trained on only 20% of the data. The experimental outcomes, detailed in Tab. 2, clearly illustrate that our method, even with limited data for training and prompt tuning, can attain the best results among all the baselines. Remarkably, on certain dataset/model combinations, our results even surpass the baseline achieved with 70% of the data used for training.

#### 4.7.2 PERFORMANCE WITH LIMITED PROMPT DATA

To further explore the efficiency of our method, we investigate the minimum amount of data required for prompt tuning to surpass baseline performances. We utilize 50% of the data for pre-training, and 5% to 20% data for prompt tuning. We select DyRep (Trivedi et al., 2019) and TIGER (Zhang et al., 2023c) to conduct experiments under the "pre-train, prompt" paradigm for this analysis. The results, as depicted in Fig. 5 and Fig. 7, reveal that as little as 10%, and in some cases only 5%, of the data is needed for our approach to prompt tuning to achieve improved results. Furthermore, we observe that increasing the amount of data used for prompt tuning correspondingly enhances the performances in the transductive setting. This finding reaffirms the efficacy of our approach.

## 5 CONCLUSION

In this paper, we introduce two novel training paradigms for TIGs, which are grounded in pre-training, prompting, and fine-tuning techniques. Additionally, we present and compare three distinct temporal prompt generators, designed to ensure the resulting prompt vectors encapsulate a significant amount of temporal information. Employing the proposed paradigms can bridge both temporal and semantic gaps in the traditional training paradigm. Moreover, through extensive experimentation, we demonstrate that our methods significantly improve the performance of TIG models over baselines across various downstream tasks, thus achieving SOTA performance.

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

## A  TEMPORAL GAP AND SEMANTIC GAP

### A.1  DEFINITION AND EXAMPLES OF THE GAPS

**Temporal gap**: The gap caused by the time difference between training and inference data. For example, in TIG models, data (interaction edges) is input into the model chronologically, with training data occurring earlier than the data encountered during inference phase. During inference, the model trained on the training data is used to generate node representations. Previous TIG models usually rely on a memory module to store historical information. Specifically, they predict nodes' future behaviors based on the stored memory, which is continuously updated. However, although the updating branch for temporal embedding modules generates new representations, the branch for memory updating often neglects this new information, leading to stale memory (Zhang et al., 2023c; Chen et al., 2023b). As a result, when there is a significant time gap between the training and inference data, the memory generated during inference cannot provide expressive historical information. Consequently, the training process becomes outdated with temporal interactions, resulting in ineffective predictions for future events (Zhang et al., 2023c).

**Semantic gap**: The gap between edge-level pretext task and node-level downstream task. For example, in the pre-training phase, the pretext task is typically link prediction, which usually brings connected nodes closer in the latent representation space. However, for node-level downstream tasks, such as node classification, using the node representations generated by the pre-trained model requires training an additional classification predictor. Since this process cannot access the pre-trained model, the output representations from the edge-level pre-trained model may lead to negative transfer when connected nodes have different labels, potentially resulting in misclassification of node labels (Sun et al., 2023). Intuitively, this is because edge-level pre-training strategy tends to enforce smoothness of node representations along observed edges, but there are many cases that two connected nodes have totally different labels, thereby exacerbating negative transfer (Sun et al., 2023).

### A.2  QUANTIFICATION OF THE GAPS

Since these gaps are often implicitly embedded in node embeddings or representations, our idea is to assess them or identify the gaps through performance on downstream tasks. For example, using prompts that incorporate temporal information (Transformer or Projection TProG) reduces the temporal gap (i.e., in link prediction tasks, the models with these two TProGs outperform the baseline), while using only the Vanilla TProG without temporal information directly narrows the semantic gap (i.e., in node classification tasks, the models with Vanilla TProG successfully outperform the baseline). We propose a set of intuitive experiments to illustrate our claims.

**Temporal gap**: Building on the previous main experiments, we further split the inference (test) data into two parts, where the edge timestamps are increasing—i.e., interactions in the first part (1st Part, corrsponds to "temporally proximal inference data" in Fig. 1 (a)) occur earlier and are closer to the training data than those in the second part (2nd Part, corrsponds to "temporally distant inference data" in Fig. 1 (a)). We then apply them and conduct inference separately. If our hypothesis about the temporal gap holds true, the performance on the first part should be better than on the second part when using the baseline methods. When applying our proposed Transformer or Projection TProGs (we use Projection TProG and take MOOC dataset as example here for illustration), the performance should be improved, and the difference between the two parts should narrow. In line with main experiments, we use AP as the evaluation metric. As shown in the Tab. 3, the results align with our hypothesis. This validates the existence of the temporal gap and demonstrates that our method helps reduce it.

**Semantic gap**: Since the link prediction and node classification tasks both use the node embeddings generated by the pre-trained models for downstream tasks, a simple way to locate the semantic gap is to compare the same metric on both link prediction task and node classification task. For a fair comparison, we use AUROC as the evaluation metric for both tasks and conduct experiments on different dataset and backbone model combinations. By comparing the difference in AUROC between the two tasks before and after applying our proposed Vanilla TProG, it can be seen (from the Tab. 4) that the differences are narrowed after applying our "pre-train, prompt" training paradigm and TProG. This proves that the semantic gap indeed exists and that our method helps to narrow it.

Table 3: Quantification of Temporal Gap: Evaluated by AP (%). The $1^{st}$ Part and the $2^{nd}$ Part corrspond to "temporally proximal inference data" and "temporally distant inference data" in Fig. 2 (c), respectively.

| Models | | Baseline | Projection TProG | Gap Narrowed |
|---|---|---|---|---|
| Jodie | $1^{st}$ Part | 76.35 | 82.60 | |
| | $2^{nd}$ Part | 72.38 | 80.17 | 38.73% |
| | GAP | 3.98 | 2.44 | |
| DyRep | $1^{st}$ Part | 74.81 | 82.71 | |
| | $2^{nd}$ Part | 69.93 | 79.56 | 35.45% |
| | GAP | 4.88 | 3.15 | |
| TGN | $1^{st}$ Part | 88.34 | 88.91 | |
| | $2^{nd}$ Part | 86.85 | 88.06 | 42.62% |
| | GAP | 1.49 | 0.86 | |
| TIGE | $1^{st}$ Part | 89.37 | 89.94 | |
| | $2^{nd}$ Part | 88.01 | 89.10 | 38.24% |
| | GAP | 1.36 | 0.84 | |
| TIGER | $1^{st}$ Part | 87.16 | 89.56 | |
| | $2^{nd}$ Part | 86.08 | 88.79 | 28.70% |
| | GAP | 1.08 | 0.77 | |

Table 4: Quantification of Semantic Gap: Evaluated by AUROC (%). (Wiki. refers to Wikipedia dataset)

| Dataset/Models | | Baseline | Vanilla TProG | Gap Narrowed |
|---|---|---|---|---|
| Wiki. / TGN | Link Prediction | 98.11 | 96.25 | |
| | Node Classification | 84.93 | 85.79 | 20.63% |
| | GAP | 13.18 | 10.46 | |
| Reddit/ Jodie | Link Prediction | 97.91 | 97.57 | |
| | Node Classification | 58.48 | 69.22 | 28.10% |
| | GAP | 39.43 | 28.35 | |
| MOOC/ TIGER | Link Prediction | 89.39 | 90.88 | |
| | Node Classification | 64.91 | 68.68 | 9.31% |
| | GAP | 24.48 | 22.20 | |

## A.3 How TProGs Narrow the Gaps

We now provide a brief theoretical analysis of how each TProG variant contributes to narrowing the semantic and temporal gaps.

**Vanilla TProG** introduces node-specific prompt vectors that are directly optimized via node-level supervision signals. This establishes a task-conditioned prompt generating, allowing the model to re-contextualize outputted representations from frozen backbone toward the target task objective (e.g., node classification), even without additional temporal signals. The effectiveness of such a setup for node classification task confirms that semantic mismatch between edge-level pre-training and node-level prediction can be mitigated through lightweight, learnable prompts.

**Projection TProG** builds upon Vanilla TProG by introducing explicit time conditioning, effectively providing a soft temporal hint to the node representation. By projecting node-specific prompt vectors into a temporal latent space using most recent interaction, it encourages the model to align the node embedding from frozen backbone models with its current or recent temporal context. Intuitively, this allows the prompt to act as a "reminder" or "hint" of recent temporal activity, helping the model adapt representations to evolving dynamics. This partially compensates for the temporal mismatch introduced during pre-training and enables better adaptation under time-varying behaviors. This design enables downstream adaptation that is both semantically aligned and temporally consistent, effectively narrowing the temporal gap and semantic gap that arise from stale backbone parameters.

**Transformer TProG** further generalizes this mechanism by conditioning prompt generation on a sequence of recent interactions through self-attention. The prompt depends on the temporal distribution and relational dynamics of recent neighbors. This captures higher-order temporal

dependencies and behavioral recency, which are crucial in interaction-dense data. As a result, the prompt embedding space adapts in a temporally fine-grained manner.

In sum, the three variants form a progressive design spectrum: from task conditioning (Vanilla), to timestamp-aware alignment (Projection), to dynamically evolving temporal modeling (Transformer). This theoretically grounded progression supports our claim that the proposed prompting framework can systematically mitigate both semantic and temporal gaps in TIG models.

# B RELATED WORK

**Temporal Interaction Graph Models.** Temporal Interaction Graph representation learning models (TIG models) are specifically designed to learn dynamic representations of the nodes in TIGs. These models employ node representations to execute downstream tasks, including link prediction (by computing node similarity) and node classification (through additional training of a classifier, i.e., projection head). The development of contemporary TIG models began with Jodie (Kumar et al., 2019). Jodie utilizes two RNNs to dynamically update node representations and employs a projection operator to estimate the embeddings of nodes that have not interacted for an extended period. DyRep (Trivedi et al., 2019) introduces a deep temporal point process model, employing a dual-time scale approach to effectively capture both association and communication dynamics. TGAT (Xu et al., 2020) revolutionizes TIG models by incorporating an attention mechanism, wherein it substitutes the original position coding with time coding to effectively aggregate information from a node's neighbors. Building on this, TGN (Rossi et al., 2020) introduces a memory module to store nodes' historical interaction information, and integrating these developments into a cohesive framework. TIGER (Zhang et al., 2023c) presents a model equipped with a dual-memory module, specifically designed for enhanced aggregation of neighbor information. TIGER also introduces a restarter module, responsible for generating surrogate representations, which serve as a warm initialization for node representations. Additionally, several works are devoted to addressing challenges and resolving specific complexities inherent in TIG models, including large-scale training (Zhou et al., 2022; Chen et al., 2023b), noise dynamics (Zhang et al., 2023a), and node-wise long-term modeling (Zhang et al., 2023b) issues. However, two critical issues persist: the limited adaptability of these models to new data, and the semantic gap between pretext tasks and downstream tasks.

**Graph Prompt Learning.** Prompt-tuning methods, originating from the NLP domain (Devlin et al., 2018; Liu et al., 2023a), have gained widespread use in adapting pre-trained language models to a variety of downstream tasks. More recently, prompt learning has emerged in the graph domain (Qin & Eisner, 2021; Tsimpoukelli et al., 2021; Sun et al., 2022; Zhu et al., 2023; Liu et al., 2023b; Sun et al., 2023; Tan et al., 2023; Fang et al., 2023; Huang et al., 2023a; Shirkavand & Huang, 2023; Gong et al., 2023; Chen et al., 2023a; Ma et al., 2024; Ge et al., 2023; Yu et al., 2024a) as a promising approach for directing downstream tasks. Pioneering works like GPPT (Sun et al., 2022) focus on the node classification task, incorporating learnable prompts directly into graphs. Similarly, GraphPrompt (Liu et al., 2023b) introduces a uniform prompt design, specifically tailored to address both node- and graph-level downstream tasks. All-in-One (Sun et al., 2023) expands graph prompt learning further by encompassing prompt tokens, structures, and insertion patterns, introducing a comprehensive, albeit complex, prompting framework. Recent advancements in prompt learning for static graphs have explored more fine-grained aspects of representation learning. GraphPrompt+ (Yu et al., 2024b) incorporates subgraph similarity and fixed structural patterns into the prompt learning framework, enabling more structured guidance. ProNoG (Yu et al., 2024c) addresses the challenges of non-homophilic graphs by focusing on structural irregularities and designing node-specific prompting strategies. STGP (Hu et al., 2024) extends prompt learning to spatio-temporal graphs in urban computing, highlighting cross-domain and multi-task transfer through a two-stage prompting mechanism. Nevertheless, there is a noticeable absence of prompt tuning methods specifically designed for the temporal interaction graphs, as existing static graph prompting works lack a temporal consideration and exhibit weak expressiveness.

**Comparison with Contemporaneous Work.** We identify a contemporaneous work, DyGPrompt (Yu et al., 2025), and provide a conceptual comparison as follows. While both TIGPrompt and DyGPrompt aim to bridge the gap between pre-training and downstream tasks in dynamic graph learning through prompt-based adaptation, the two methods differ in design goals and technical implementation. DyGPrompt introduces a dual-prompt and dual-conditioning framework, utilizing both feature and

temporal prompts along with a node-time co-conditioning mechanism. This design enables fine-grained joint modeling of node features and timestamps through a sophisticated co-conditioning process. In contrast, our work identifies two fundamental gaps—temporal and semantic—in traditional TIG training paradigms, and proposes a novel prompt-based training approach to bridge them. Specifically, we propose to use "pre-train, prompt" paradigm or "pre-train, prompt-based fine-tune" paradigm, bridging the temporal and semantic gaps and introduce TProGs to construct prompts that incorporate temporal information, aligning with the inherent characteristics of TIGs. Our approach emphasizes a new training paradigm for TIGs and lightweight, time-aware prompt generation through variants of TProGs. We thus consider DyGPrompt a complementary contemporaneous work. While DyGPrompt emphasizes fine-grained adaptivity in node-time modeling, TIGPrompt offers a simple, efficient, and broadly applicable solution. Due to the unavailability of DyGPrompt's source code, we do not include a direct empirical comparison in our paper.

## C DATASETS

In alignment with previous studies (Kumar et al., 2019; Trivedi et al., 2019; Rossi et al., 2020; Zhang et al., 2023c), we utilize four public datasets made available by the authors of Jodie (Kumar et al., 2019). Detailed statistics of these datasets can be found in Tab. 5.

Table 5: Dataset Statistics. $d_n$ and $d_e$ indicate the dim of nodes and edges, respectively.

|  | # Nodes | # Edges | $d_n$ | $d_e$ | Classes |
|---|---|---|---|---|---|
| Wikipedia | 9,227 | 157,474 | 172 | 172 | 2 |
| Reddit | 10,984 | 672,447 | 172 | 172 | 2 |
| MOOC | 7,144 | 411,749 | 172 | 172 | 2 |
| LastFM | 1,980 | 1,293,103 | 172 | 172 | - |

## D EXPERIMENTS UNDER "PRE-TRAIN, PROMPT-BASED FINE-TUNE"

### D.1 LINK PREDICTION

We provide the complete experiment results for the "pre-train, prompt-based fine-tune" paradigm link prediction task in both transductive and inductive settings in Tab. 6.

### D.2 NODE CLASSIFICATION

#### D.2.1 TRAINING STRATEGIES

Under the "pre-train, prompt-based fine-tune" paradigm for the node classification task, three different strategies can be applied: (1) directly employing the TProG trained in the link prediction task to generate prompts; (2) using the link prediction-trained TProG to initialize a TProG and then further optimizing it during node classification; and (3) discarding the previously TProG and re-initializing a new one for optimization alongside the node classification task.

We choose the first strategy for our experiments, with the outcomes detailed in Tab. 7. Notably, a part of these results exceed those achieved under the "pre-train, prompt" paradigm. However, similar to the link prediction task, this approach demands additional training resources. A comparison of three training strategies is presented in Appendix D.2.2. This comparison demonstrates that applying the other two strategies has the potential to improve the performance of node classification tasks.

#### D.2.2 COMPARISON BETWEEN THREE STRATEGIES OF NODE CLASSIFICATION TRAINING

Beyond the initial experiments conducted under "pre-train, prompt-based fine-tune" for the node classification task, we extend our investigation to include various training strategies outlined in Appendix D.2.1. A series of experiments was conducted using the Wikipedia dataset, employing the Projection TProG. The outcomes of these experiments are illustrated in Fig. 6. The results indicate that our method outperforms the baseline models when different strategies are applied, thereby demonstrating the effectiveness of our approach.

Table 6: Full results of Average Precision (%) for the link prediction tasks under the "pre-train, prompt-based fine-tune" paradigm in both Transductive and Inductive settings.

| | TProG | Transductive Link Prediction | | | | Inductive Link Prediction | | | |
|---|---|---|---|---|---|---|---|---|---|
| | | Wikipedia | Reddit | MOOC | LastFM | Wikipedia | Reddit | MOOC | LastFM |
| Jodie | Baseline | $94.62_{\pm0.5}$ | $97.11_{\pm0.3}$ | $76.50_{\pm1.8}$ | $68.77_{\pm3.0}$ | $93.11_{\pm0.4}$ | $94.36_{\pm1.1}$ | $77.83_{\pm2.1}$ | $82.55_{\pm1.9}$ |
| | Vanilla | $94.22_{\pm0.9}$ | $97.17_{\pm0.3}$ | $76.32_{\pm1.6}$ | $74.45_{\pm1.3}$ | $92.66_{\pm1.0}$ | $93.91_{\pm0.9}$ | $74.58_{\pm2.5}$ | $81.27_{\pm1.0}$ |
| | Transformer | $\mathbf{97.01_{\pm0.4}}$ | $98.25_{\pm0.1}$ | $\mathbf{85.52_{\pm0.6}}$ | $76.48_{\pm1.5}$ | $\mathbf{96.13_{\pm0.5}}$ | $96.71_{\pm0.3}$ | $\mathbf{84.33_{\pm0.6}}$ | $84.63_{\pm1.3}$ |
| | Projection | $96.72_{\pm0.6}$ | $\mathbf{98.84_{\pm0.1}}$ | $83.03_{\pm0.3}$ | $\mathbf{88.82_{\pm0.5}}$ | $95.36_{\pm0.6}$ | $\mathbf{97.79_{\pm0.2}}$ | $81.72_{\pm1.3}$ | $\mathbf{92.51_{\pm0.4}}$ |
| DyRep | Baseline | $94.59_{\pm0.2}$ | $97.98_{\pm0.1}$ | $75.37_{\pm1.7}$ | $68.77_{\pm2.1}$ | $92.05_{\pm0.3}$ | $95.68_{\pm0.2}$ | $78.55_{\pm1.1}$ | $81.33_{\pm2.1}$ |
| | Vanilla | $90.48_{\pm1.1}$ | $97.15_{\pm0.2}$ | $74.88_{\pm2.5}$ | $72.96_{\pm0.5}$ | $88.50_{\pm1.3}$ | $93.31_{\pm0.7}$ | $73.42_{\pm2.7}$ | $80.79_{\pm1.8}$ |
| | Transformer | $95.62_{\pm0.4}$ | $98.17_{\pm0.1}$ | $\mathbf{84.81_{\pm1.1}}$ | $74.22_{\pm1.8}$ | $94.52_{\pm0.6}$ | $96.61_{\pm0.2}$ | $\mathbf{83.38_{\pm0.7}}$ | $83.74_{\pm2.5}$ |
| | Projection | $\mathbf{97.19_{\pm0.2}}$ | $\mathbf{98.96_{\pm0.1}}$ | $82.53_{\pm1.7}$ | $\mathbf{88.83_{\pm0.4}}$ | $\mathbf{96.11_{\pm0.3}}$ | $\mathbf{97.78_{\pm0.2}}$ | $81.51_{\pm1.0}$ | $\mathbf{92.59_{\pm0.4}}$ |
| TGN | Baseline | $\mathbf{98.46_{\pm0.1}}$ | $98.70_{\pm0.1}$ | $85.88_{\pm3.0}$ | $71.76_{\pm5.3}$ | $\mathbf{97.81_{\pm0.1}}$ | $97.55_{\pm0.1}$ | $85.55_{\pm2.9}$ | $80.42_{\pm4.9}$ |
| | Vanilla | $97.72_{\pm0.2}$ | $98.32_{\pm0.1}$ | $88.58_{\pm1.1}$ | $72.69_{\pm5.0}$ | $96.94_{\pm0.1}$ | $96.51_{\pm0.3}$ | $87.89_{\pm0.9}$ | $78.97_{\pm3.9}$ |
| | Transformer | $98.25_{\pm0.1}$ | $98.68_{\pm0.1}$ | $89.95_{\pm1.7}$ | $77.79_{\pm3.2}$ | $97.59_{\pm0.2}$ | $97.62_{\pm0.1}$ | $89.11_{\pm1.2}$ | $83.48_{\pm2.4}$ |
| | Projection | $98.38_{\pm0.1}$ | $\mathbf{99.29_{\pm0.0}}$ | $\mathbf{90.00_{\pm1.4}}$ | $\mathbf{90.08_{\pm0.9}}$ | $\mathbf{97.81_{\pm0.1}}$ | $\mathbf{98.61_{\pm0.1}}$ | $\mathbf{89.15_{\pm1.6}}$ | $\mathbf{92.64_{\pm0.9}}$ |
| TIGE | Baseline | $98.83_{\pm0.1}$ | $99.04_{\pm0.0}$ | $89.64_{\pm0.9}$ | $87.85_{\pm0.9}$ | $98.45_{\pm0.1}$ | $98.39_{\pm0.1}$ | $89.51_{\pm0.7}$ | $90.14_{\pm1.0}$ |
| | Vanilla | $98.84_{\pm0.0}$ | $98.87_{\pm0.0}$ | $90.18_{\pm0.7}$ | $89.06_{\pm0.5}$ | $98.37_{\pm0.0}$ | $97.82_{\pm0.2}$ | $89.59_{\pm0.5}$ | $91.06_{\pm0.4}$ |
| | Transformer | $98.99_{\pm0.0}$ | $99.20_{\pm0.0}$ | $\mathbf{92.14_{\pm0.9}}$ | $91.22_{\pm0.3}$ | $98.58_{\pm0.0}$ | $98.70_{\pm0.1}$ | $\mathbf{91.22_{\pm0.8}}$ | $92.81_{\pm0.3}$ |
| | Projection | $\mathbf{99.12_{\pm0.0}}$ | $\mathbf{99.48_{\pm0.0}}$ | $91.68_{\pm0.4}$ | $\mathbf{95.30_{\pm0.1}}$ | $\mathbf{98.84_{\pm0.0}}$ | $\mathbf{99.16_{\pm0.0}}$ | $91.16_{\pm0.4}$ | $\mathbf{96.20_{\pm0.1}}$ |
| TIGER | Baseline | $98.90_{\pm0.0}$ | $99.02_{\pm0.0}$ | $86.99_{\pm1.6}$ | $85.17_{\pm0.2}$ | $98.58_{\pm0.0}$ | $98.59_{\pm0.0}$ | $86.42_{\pm1.7}$ | $89.11_{\pm0.3}$ |
| | Vanilla | $98.90_{\pm0.0}$ | $98.84_{\pm0.0}$ | $85.12_{\pm1.1}$ | $85.59_{\pm0.5}$ | $98.49_{\pm0.1}$ | $98.13_{\pm0.1}$ | $84.37_{\pm0.8}$ | $88.43_{\pm0.6}$ |
| | Transformer | $99.05_{\pm0.0}$ | $99.18_{\pm0.0}$ | $87.00_{\pm0.9}$ | $87.84_{\pm0.2}$ | $98.68_{\pm0.0}$ | $98.78_{\pm0.0}$ | $86.07_{\pm1.0}$ | $90.50_{\pm0.3}$ |
| | Projection | $\mathbf{99.17_{\pm0.0}}$ | $\mathbf{99.49_{\pm0.0}}$ | $\mathbf{87.83_{\pm0.6}}$ | $\mathbf{93.50_{\pm0.2}}$ | $\mathbf{98.88_{\pm0.0}}$ | $\mathbf{99.28_{\pm0.0}}$ | $\mathbf{87.38_{\pm0.9}}$ | $\mathbf{94.90_{\pm0.3}}$ |
| GraphMixer | Baseline | $97.25_{\pm0.0}$ | $97.31_{\pm0.0}$ | $82.78_{\pm0.2}$ | $75.61_{\pm0.2}$ | $96.65_{\pm0.0}$ | $95.26_{\pm0.0}$ | $81.41_{\pm0.2}$ | $82.11_{\pm0.4}$ |
| | Vanilla | $96.24_{\pm0.1}$ | $97.52_{\pm0.0}$ | $81.27_{\pm0.3}$ | $76.91_{\pm0.3}$ | $95.65_{\pm0.1}$ | $94.25_{\pm0.2}$ | $79.27_{\pm0.9}$ | $81.86_{\pm0.4}$ |
| | Transformer | $97.45_{\pm0.0}$ | $98.12_{\pm0.0}$ | $84.09_{\pm0.9}$ | $78.19_{\pm0.3}$ | $97.02_{\pm0.0}$ | $\mathbf{96.40_{\pm0.0}}$ | $81.61_{\pm1.2}$ | $83.81_{\pm0.3}$ |
| | Projection | $\mathbf{98.99_{\pm0.2}}$ | $\mathbf{99.23_{\pm0.0}}$ | $\mathbf{87.48_{\pm0.2}}$ | $\mathbf{88.84_{\pm3.1}}$ | $\mathbf{97.78_{\pm0.5}}$ | $94.43_{\pm0.9}$ | $\mathbf{84.76_{\pm0.1}}$ | $\mathbf{86.92_{\pm2.3}}$ |
| DyGFormer | Baseline | $99.03_{\pm0.0}$ | $99.22_{\pm0.0}$ | $87.52_{\pm0.5}$ | $93.00_{\pm0.1}$ | $98.59_{\pm0.0}$ | $98.84_{\pm0.0}$ | $86.96_{\pm0.4}$ | $94.23_{\pm0.1}$ |
| | Vanilla | $98.97_{\pm0.0}$ | $99.16_{\pm0.0}$ | $86.42_{\pm0.4}$ | $92.78_{\pm0.1}$ | $98.55_{\pm0.0}$ | $98.78_{\pm0.0}$ | $85.67_{\pm0.5}$ | $94.14_{\pm0.0}$ |
| | Transformer | $99.07_{\pm0.1}$ | $99.50_{\pm0.1}$ | $87.92_{\pm0.3}$ | $93.76_{\pm0.1}$ | $98.76_{\pm0.1}$ | $\mathbf{99.12_{\pm0.1}}$ | $87.17_{\pm0.3}$ | $94.69_{\pm0.3}$ |
| | Projection | $\mathbf{99.84_{\pm0.0}}$ | $\mathbf{99.87_{\pm0.0}}$ | $\mathbf{91.06_{\pm0.3}}$ | $\mathbf{95.12_{\pm0.2}}$ | $\mathbf{99.44_{\pm0.0}}$ | $98.79_{\pm0.2}$ | $\mathbf{89.08_{\pm0.2}}$ | $\mathbf{94.99_{\pm0.4}}$ |

Table 7: AUROC (%) for dynamic node classification task under "pre-train, prompt-based fine-tune".

| | TProG | Node Classification | | |
|---|---|---|---|---|
| | | Wikipedia | Reddit | MOOC |
| Jodie | Baseline | $86.27_{\pm2.2}$ | $58.48_{\pm2.6}$ | $65.39_{\pm1.1}$ |
| | Vanilla | $84.82_{\pm0.3}$ | $63.87_{\pm1.4}$ | $66.32_{\pm1.8}$ |
| | Transformer | $\mathbf{86.42_{\pm2.4}}$ | $\mathbf{67.19_{\pm1.0}}$ | $71.36_{\pm0.8}$ |
| | Projection | $84.41_{\pm3.0}$ | $62.27_{\pm3.8}$ | $\mathbf{75.89_{\pm1.5}}$ |
| DyRep | Baseline | $85.11_{\pm1.4}$ | $62.77_{\pm2.1}$ | $66.68_{\pm3.4}$ |
| | Vanilla | $\mathbf{88.64_{\pm1.8}}$ | $58.64_{\pm2.7}$ | $65.00_{\pm2.2}$ |
| | Transformer | $83.73_{\pm0.3}$ | $\mathbf{64.58_{\pm2.2}}$ | $71.98_{\pm2.8}$ |
| | Projection | $85.35_{\pm0.5}$ | $58.84_{\pm2.1}$ | $\mathbf{75.09_{\pm1.3}}$ |
| TGN | Baseline | $\mathbf{84.93_{\pm1.1}}$ | $\mathbf{65.99_{\pm3.8}}$ | $69.80_{\pm1.8}$ |
| | Vanilla | $82.49_{\pm2.7}$ | $62.93_{\pm3.8}$ | $64.66_{\pm3.9}$ |
| | Transformer | $82.43_{\pm1.1}$ | $64.67_{\pm3.5}$ | $70.03_{\pm2.9}$ |
| | Projection | $83.86_{\pm1.4}$ | $60.28_{\pm4.8}$ | $\mathbf{77.15_{\pm3.1}}$ |
| TIGE | Baseline | $83.98_{\pm3.4}$ | $\mathbf{65.36_{\pm2.9}}$ | $69.61_{\pm2.5}$ |
| | Vanilla | $81.43_{\pm6.8}$ | $62.46_{\pm2.5}$ | $70.35_{\pm0.8}$ |
| | Transformer | $85.87_{\pm2.0}$ | $64.14_{\pm1.6}$ | $67.61_{\pm5.9}$ |
| | Projection | $\mathbf{88.51_{\pm0.8}}$ | $59.08_{\pm3.9}$ | $\mathbf{78.04_{\pm3.2}}$ |
| TIGER | Baseline | $80.84_{\pm4.6}$ | $62.58_{\pm1.3}$ | $64.91_{\pm5.2}$ |
| | Vanilla | $84.93_{\pm2.5}$ | $\mathbf{64.22_{\pm1.8}}$ | $68.16_{\pm2.9}$ |
| | Transformer | $83.95_{\pm4.4}$ | $60.75_{\pm1.3}$ | $68.26_{\pm1.8}$ |
| | Projection | $\mathbf{85.13_{\pm1.4}}$ | $61.20_{\pm2.2}$ | $\mathbf{81.58_{\pm1.2}}$ |

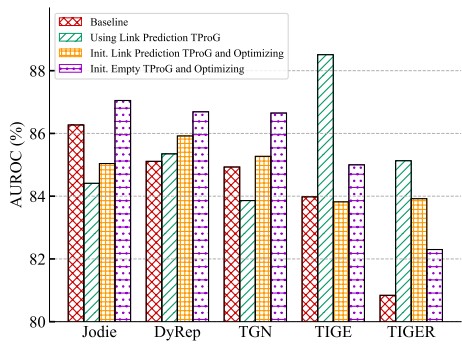

Figure 6: Comparison between three different "pre-train, prompt-based fine-tune" node classification training strategies (Wikipedia dataset, employing the Projection TProG).

# E CONTINUED EXPERIMENT RESULTS

## E.1 RESULTS FOR LIMITED PROMPT DATA EXPERIMENTS

We provide the complete experiment results for limited prompt data analysis in Fig. 7).

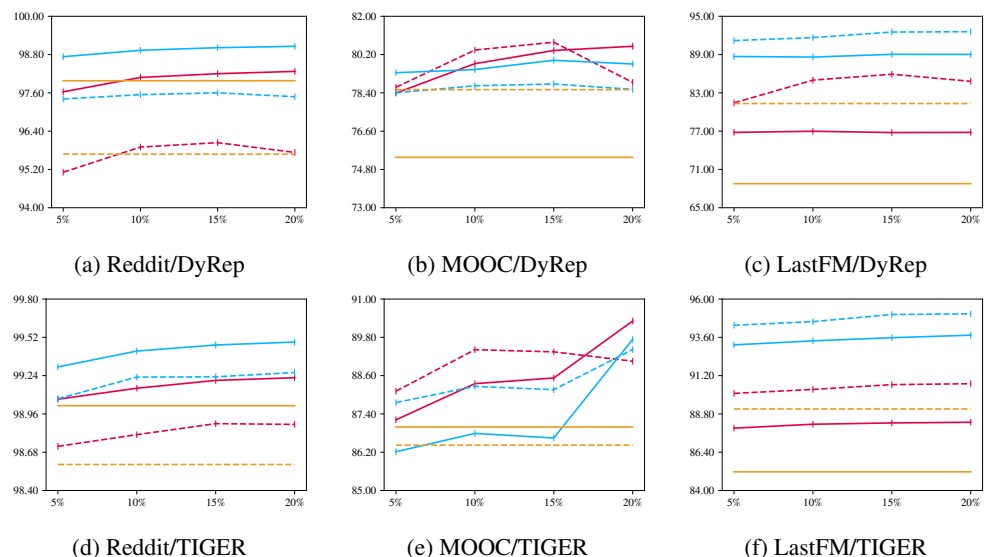

Figure 7: Performance w.r.t the Proportion of Prompting Data. This is a continued figure of Fig. 5.

### E.2 APPLYING TO NON-MEMORY-BASED TIG METHODS

The basic baseline models utilized in this paper are based on the TGN architecture (Rossi et al., 2020), which employ a memory module to store historical interaction information for nodes. Recently, various model architectures have been proposed by researchers, incorporating different backbone models. GraphMixer (Cong et al., 2023) and DyGFormer (Yu et al., 2023) are two representative works based on MLP and Transformer architectures, respectively. Although GraphMixer and DyGFormer do not share a similar architecture with the memory-based TIG methods (i.e., methods based on TGN architecture or TGN-based methods), they similarly utilize representations for downstream tasks. Our proposed TIGPrompt, wherein the prompt is fused with node representations for use in downstream tasks, is thus thought to potentially combine effectively with GraphMixer and DyGFormer. To explore this possibility, we conduct a set of experiments based on these two models. As demonstrated in Tab. 1, 2 and 6, our proposed TIGPrompt can effectively enhance the performance of non-memory-based TIG models on both link prediction and node classification tasks.[2]

Although we only implement experiments on GraphMixer and DyGFormer, the underlying mechanism is similar for other methods that build upon them, such as DyGMamba (Ding et al., 2024) and FreeDyG (Tian et al., 2024). Our proposed method is not a new backbone model, but rather a general training paradigm designed to adapt existing TIG models to downstream tasks in a more flexible and efficient manner. While the motivation is inspired by TGN-based architectures, our empirical evaluation covers models beyond memory-based designs, i.e., GraphMixer (MLP-based) and DyGFormer (Transformer-based). These results demonstrate that TIGPrompt is broadly compatible with different backbone types, as long as they follow the triditional "pre-train, predict" training paradigm.

## F TPROG VARIANT SELECTION

We provide guidance on selecting among the three TProG variants according to dataset characteristics and computational–performance considerations.

Vanilla TProG, with its lightweight $\mathcal{O}(|\mathcal{V}|)$ node-dependent parameters, focuses primarily on mitigating the semantic gap and offers the fastest inference among all variants. It is well suited for datasets with relatively few nodes, scenarios requiring low-latency inference, and node classification tasks where semantic alignment dominates over temporal dynamics.

---

[2]Experiments are conducted based on the open-source repository DyGLib (Yu et al., 2023). We employ the best model configurations as provided by DyGLib for the pre-training process with default settings.

Projection TProG also scales with $\mathcal{O}(|\mathcal{V}|)$ parameters but incorporates temporal cues, enabling it to address both the semantic and temporal gaps while maintaining high computational efficiency. This variant is particularly appropriate for small- to medium-scale datasets or applications that require a balanced trade-off between temporal expressiveness and inference cost.

In contrast, Transformer TProG employs a lightweight Transformer encoder with parameters scaling as $\mathcal{O}(d)$, making it more scalable for large graphs and especially effective when modeling complex or irregular temporal patterns. It typically achieves the strongest performance in settings where temporal gap mitigation is crucial and accuracy is prioritized over inference speed.

## G  DATA AMOUNT FOR TRAINING

In this section, we analyze the amount of data used in our experiment for training and the reasons behind the resulting experimental outcomes. Note that all experiments use 15% of the data for validation and a different 15% for testing. The data amount used for training is summarized in Tab. 8.

Firstly, we use 50% of the data for pre-training, followed by 20% of the data for prompt tuning, making a total of 70% of the data used for training (Sec. 4.3 and 4.4). This setup is to align with the 70% data for training of baseline models. Additionally, we adjust the amount of 20% prompt tuning data through comparative experiments to explore the effects of different tuning data volumes (Sec. 4.7.2). Then, we compare the situation where only a small amount of training data is available, i.e., the baseline uses only 20% of the data for training, whereas our method uses only 10% of the data for pre-training and 10% for prompt tuning, making a total of 20% of the data for overall training (Sec. 4.7.1).

It is natural that some experimental results may show degradation when only 10% of the data is allocated for pre-training, compared to the baseline results achieved with 70% of the data used for training. This can be attributed to the substantial decrease in the amount of overall training data. However, as can be seen from Tab. 2, almost all results of our method surpass the baseline of using only 20% of the data for training, with part of results (marked in blue in the Tab. 2) surpass the baseline models training with 70% of data. This demonstrates the effectiveness of our proposed method.

Table 8: Training Data Amount for different experiments.

| Experiments | Methods | Pre-train/ Training | Prompt tuning | Total for Training |
|---|---|---|---|---|
| Main (Sec. 4.3 and 4.4) | Baseline | 70% | / | 70% |
| | TIGPrompt | 50% | 20% | 70% |
| Limited Training Data (Sec. 4.7.1) | Baseline | 20% | / | 20% |
| | TIGPrompt | 10% | 10% | 20% |
| Limited Prompt Data (Sec. 4.7.2) | Baseline | 70% | / | 70% |
| | TIGPrompt | 50% | 5%-20% | 55%-70% |

**Discussions on "Weak Supervision".** In the original prompt learning literature from NLP (Devlin et al., 2018; Liu et al., 2023a), the concept of few-shot learning is well-established. However, this notion is difficult to directly translate into the context of TIGs. In TIGs, a few-shot setting can only be simulated by restricting the amount of data used during either the fine-tuning phases. Notably, temporal link prediction—the core task for both pretext and downstream objectives in many TIG models—does not lend itself easily to a few-shot formulation. This is because the supervision signal arises from future interactions rather than class labels, making it hard to define a fixed number of "support" instances typical of few-shot learning. For node classification, existing few-shot methods designed for NLP (Devlin et al., 2018; Liu et al., 2023a) or static graphs (Liu et al., 2023b; Sun et al., 2023) are also not directly applicable. In TIGs, the task typically involves dynamic node classification, where the label of a node may evolve over time. Additionally, training a classification head in this setting still requires a minimum amount of data, further complicating the establishment of a rigorous few-shot regime. As such, we argue that constructing an effective few-shot setting for TIG representation learning remains an open and under-explored challenge.

To address this, we introduce the concept of weak supervision in TIG prompt learning. Here, weak supervision refers to training under limited data availability—not only during prompt tuning but also throughout the entire training pipeline, including pre-training.

Specifically, we explore scenarios where only 5%–20% of the data is used for prompt tuning (with a total training budget of 55%–70% data, please refer to Sec.4.7.1), or even more extreme cases where 10% is allocated for pre-training and another 10% for prompt tuning—resulting in a total training budget of just 20% data (please refer to Sec.4.7.2). These settings demonstrate the strong data efficiency and weak-supervision tolerant of our method, particularly when compared to traditional baselines trained on the full 70% of the data, which still underperformed.

## H  PARAMETER ANALYSIS

In these experiments, we explore the impacts of the dimension of the prompt vector. Additionally, we examine whether increasing the dimensions could yield even better results. As shown in Fig. 8, the results indicate that a 64-dimensional prompt vector suffices to surpass the baseline performance in most cases. While higher dimensions do improve outcomes, they also increase the model's complexity. Researchers, therefore, should weigh the trade-off between experimental effectiveness and resource efficiency when selecting the optimal prompt vector dimension.

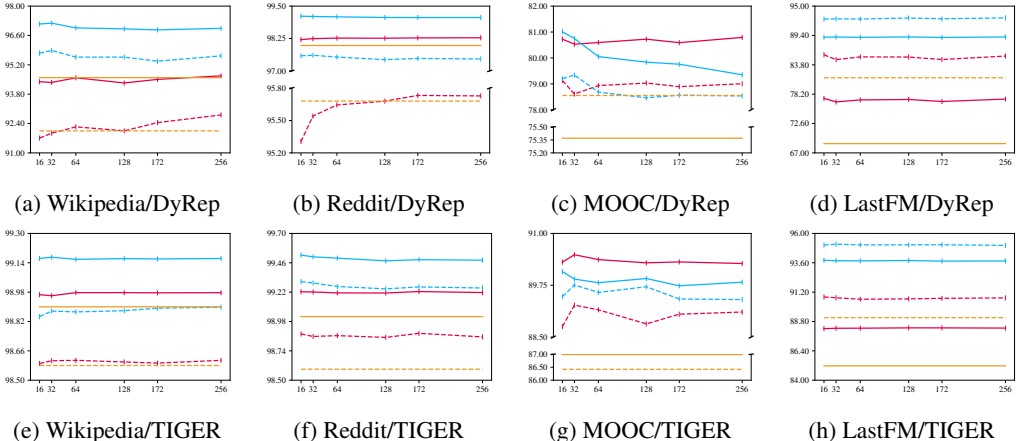

Figure 8: Performance w.r.t the Prompts Dimension. This figure shares the same legend with Fig. 5.

## I  EFFICIENCY ANALYSIS

We first record the training time on the Nvidia V100 GPU of the most commonly used baseline model, TGN (Rossi et al., 2020), on two datasets. As shown in Tab. 9, the Transformer TProG exhibits modest time efficiency due to the inherent computational slowness of transformers. However, the other two TProGs both register substantial efficiency enhancements. The results demonstrate that the proposed method is indeed lightweight.

We further provide a theoretical comparison between TProGs and TGN (other backbones exhibit similar complexity).

For TGN, assuming the node embeddings, including the memory, and prompts use the same dimension $d_n$ as the input node features, and edge features dimension is $d_e$.

The complexity for the time encoding is $\mathcal{O}(d_{te})$, where $d_{te}$ is the dimension of time encoding. The memory module's complexity is $\mathcal{O}(|\mathcal{V}| \cdot d_n)$, where $|\mathcal{V}|$ is the total number of nodes. TGN employs a GRU as the memory updater, which has a complexity of $\mathcal{O}((d_{te} + 2d_n + d_e) \cdot d_n + d_n^2)$. TGN uses multi-head attention to compute node embeddings, with the complexity of $\mathcal{O}(L \cdot ((d_n + d_e + d_{te}) \cdot h + (n + 1)^2 + h \cdot d_n))$, where $h$ is hidden layer dimension, $L$ is number of layers, and $n$ is the number of neighbors. Thus, the overall space complexity of TGN can be expressed as adding

these four terms together. As observed, the memory module contributes significantly to TGN's space complexity, especially for large graphs.

In contrast, Vanilla TProG introduces only a learnable prompt vector for each node. Its overall complexity is $\mathcal{O}(|\mathcal{V}| \cdot d_n)$. This results in a lower computational complexity compared to TGN.

Transformer TProG employs a 1-layer Transformer to generate prompts, with a complexity of $\mathcal{O}((2d_n + 1 + d_e + d_{te}) \cdot h + K^2 + h \cdot d_n)$, as derived in Equ. 2, where $K$ is the sampled historical interactions used to compute the prompts. Notably, this complexity is independent of the number of nodes, i.e., $|\mathcal{V}|$, making it more efficient for larger TIGs with many nodes.

Projection TProG shares a similar structure with Vanilla TProG in maintaining a node-specific prompt vector, but further incorporates a lightweight MLP to model temporal dependencies. Its complexity can be expressed as $\mathcal{O}(d_n \cdot d_{te} + d_n^2 + |\mathcal{V}| \cdot d_n)$, which remains lower than that of TGN, while offering improved modeling capability.

As the results in Tab. 9 and the complexity analysis show, our method boosts efficiency and lowers training resources versus the baselines. Despite its efficiency, our method still yields favorable outcomes in downstream tasks.

Table 9: Training time for one epoch (in seconds) comparison.

| | TProG | Training Time |
|---|---|---|
| Wikipedia | Baseline | 15.1 |
| | Vanilla | 4.4(-70.9%) |
| | Transformer | 14.4(-4.6%) |
| | Projection | 4.2(-72.2%) |
| MOOC | Baseline | 36.9 |
| | Vanilla | 12.6(-65.9%) |
| | Transformer | 23.2(-37.1%) |
| | Projection | 12.4(-66.4%) |

## J    IMPLEMENTATION DETAILS

We implement our methods in PyTorch, building on the official implementations of TGN (Rossi et al., 2020), TIGER (Zhang et al., 2023c) and DyGFormer (Yu et al., 2023). Unless specified otherwise, we adhere to the default hyper-parameters listed in Tab. 10 and maintain the same data pre-processing and hyper-parameter settings as in the original implementations. Since we strictly follow the settings in the original implementations, we reuse the baseline results reported in (Zhang et al., 2023c) as baselines. To fairly assess the effect of our proposed training framework, we deliberately refrain from adjusting hyper-parameters, and treat negative sampling strategies (Yu et al., 2023; Huang et al., 2023b) as intrinsic, hyper-parameter-level choices specific to each backbone model (e.g., DyGFormer (Yu et al., 2023) adopts different strategies across datasets and model variants). Consequently, we keep all default configurations unchanged and integrate TIGPrompt on top of the original implementations. This ensures that the observed performance gains stem from the prompting paradigm rather than backbone-specific heuristics.

All experiments are conducted on a single server with 72 cores, 32GB memory, and single Nvidia Tesla V100 GPU.

## K    LIMITATIONS AND FUTURE WORK

We provide a novel training paradigm for TIGs, while we may need to conduct a certain amount of additional experiments to test which TProG is more suitable for the current dataset/baseline combination for performance consideration. However, for the improvement in performance, we believe this extra effort is worthwhile. We also provide practical guidance for selecting among different TProG variants in Appendix F. Our paper has demonstrated that all three TProGs are effective through extensive experiments. The current work only focuses on and considers a series of

Table 10: Default values of hyper-parameters.

| Hyper-parameter | Value |
|---|---|
| Batch size (Pre-training) | 200 |
| Batch size (Prompt tuning) | 100 |
| Learning rate | 0.0001 |
| Optimizer | Adam |
| Prompt dimension | 172 |
| Memory dimension | 172 |
| Negative sampling | Same as backbone models |

baseline models based on TGN. The current method only considers individual datasets and does not account for integrating multiple datasets to construct a large dataset for pre-training.

In light of our study's scope and findings, we identify several potential directions for future work:

- Designing TProG variants to better match various baseline models and datasets.
- Utilizing larger datasets to complete comprehensive pre-training processes, followed by fine-tuning or prompt tuning for diverse datasets.
- Extending our methodologies to additional downstream tasks, including graph-level tasks.

## L  THE USE OF LARGE LANGUAGE MODELS

The Large Language Models are only used for editing and formatting purposes.

