# OpenReview forum: "Bridging Temporal and Semantic Gaps: Prompt Learning on Temporal Interaction Graphs"
_ICLR.cc/2026/Conference — Submitted to ICLR 2026_

### Official Review · Reviewer_baDW · 2025-10-22

**Soundness:** 3
**Presentation:** 3
**Contribution:** 2
**Rating:** 4
**Confidence:** 4

**Summary:**

In this work, the authors introduced TIGPrompt, a novel framework that applies prompt learning to TIGs to bridge temporal and semantic gaps in current models. TIGPrompt focus on the pre-train and prompt paradigm, different from standard training. The Temporal
Prompt Generator generates personalised temporal prompts for each node to adapt to downstream task. The authors evaluated the proposed methods on link prediction and node classification tasks and shows improvement to the base model with TIGPrompt.

**Strengths:**

- **originally: idea of temporal prompt generator** the idea of using a prompt generator to adapt to downstream task is interesting. The authors introduced three prompt variants: Vanilla TProG, Transformer TProG, Projection TProG
- **clarity: easy to follow** the paper is easy to follow, the authors presented the ideas well.
- **extensive evaluation** the authors evaluated across four benchmark datasets (Wikipedia, Reddit, MOOC, LastFM) and seven TIG backbones (e.g., TGN, DyRep, TIGER).
- **task improvements**. The authors show that the TIGPrompt can improve baseline performances across a variety of TGNNs on both link and node tasks.

**Weaknesses:**

I believe the current work has the following weakness:
- **limited evaluation metrics**: the link prediction experiments rely primarily on Average Precision (AP), whereas more robust ranking-based metrics such as Mean Reciprocal Rank (MRR) have been extensively adapted in prior work such as TGB[1] and ROLAND[2]
would provide a fairer and more direct reflection of the improvement of TIGPrompt which leads me to the next weakness.

- **performance saturation**: the main problem of the AP / binary classification evaluation lies in its over-inflated performances.
This saturation makes it difficult to assess whether TIGPrompt provides substantial practical improvements or merely marginal gains within an already near-perfect range. For example,  on Wikipedia, DyGformer improvement in baseline is 99.03 and the improvement Projection TProG is 99.80, this is hardly convincing as the evaluation is simply too easy and the task with this evaluation is already solved. This is even worse when considering Table 2 where only 20% of data is used for training and most models can solve the tasks with > 95% AP on two out of the four datasets.

- **unclear dataset transferability**: the main appeal of prompt learning is its ability to adapt to new datasets. From the provided results, it seems TIGPrompt mainly focus on improving task transferability on the same dataset yet it is still required that the model is trained and tested on the same dataset. It is unclear how TIGPrompt might be used to facilitate transfer to new datasets, for example, pre-train on Wikipedia and then transfer to Reddit.

Suggestion: my main suggestion for the author is to provide new results with MRR or other ranking metrics that at least require the model to rank across many potential negative destinations. This will enable the demonstration of potentially more significant empirical gain of TIGPrompt and strengthen its significance. The near perfect performance of AP is not a good indicator for good performance as the task is too easy.

[1] Huang S, Poursafaei F, Danovitch J, Fey M, Hu W, Rossi E, Leskovec J, Bronstein M, Rabusseau G, Rabbany R. Temporal graph benchmark for machine learning on temporal graphs. Advances in Neural Information Processing Systems. 2023 Dec 15;36:2056-73.

[2] You J, Du T, Leskovec J. ROLAND: graph learning framework for dynamic graphs. InProceedings of the 28th ACM SIGKDD conference on knowledge discovery and data mining 2022 Aug 14 (pp. 2358-2366).

**Questions:**

- it seems that TIGPrompt achieves good training efficiency with using only a small amounts of data. The four datasets benchmarked are all on the smaller side with only a few million edges, would it be possible to run TIGPrompt on large temporal graph datasets such as those in [TGB](https://tgb.complexdatalab.com/), i.e. with tens of millions of edges. Maybe with TIGPrompt it is possible to only use 10% of data in training, thus enabling existing models to scale to datasets where they would normally not be able work.

- the Transformer TProG only considers one hop neighborhood, would more hops help?

- what if we don't even need the base TGNN and just use the prompt model like TProG for prediction?

**Details Of Ethics Concerns:**

No ethics concerns.

---

> ### Author Response · Authors · 2025-11-15
>
> We sincerely thank the reviewer for the constructive and encouraging feedback. We appreciate the reviewer’s recognition of the originality of our temporal prompt generator, the clarity of our presentation, and the breadth of our empirical evaluation across four benchmark datasets and seven TIG backbones. We are also glad that the reviewer found the idea of applying personalized temporal prompts to downstream tasks interesting and noted the consistent improvements achieved by TIGPrompt on both link prediction and node classification tasks. Below, we address the reviewer’s concerns in detail.

---

> ### Author Response · Authors · 2025-11-15
>
> **W1 & W2**
>
> **Regarding the use of AP instead of MRR:**
>
> In the temporal interaction graph (TIG) and continuous-time dynamic graph (CTDG) literature, Average Precision (AP) is the ***standard*** and **most widely** adopted metric for evaluating temporal link prediction. All baseline methods we compare against—including JODIE, DyRep, TGN, and TIGER—also rely primarily on AP. We therefore follow this established setting to ensure **fair** and **consistent** comparison with prior work.
>
> **Regarding performance saturation under AP:**
>
> We agree the reviewer raises an important point: AP may saturate on *easy datasets* (e.g., Wikipedia), making improvements appear marginal in those cases. However, AP does **not** saturate on **harder** datasets such as **MOOC and LastFM**, especially in the **low-data regime** used in **Table 2**. For these datasets, the baseline AP scores are typically **below 80%**, while TIGPrompt provides **more than 10%** absolute **improvement**, demonstrating that our method delivers **substantial**—not marginal—**gains** when the evaluation setting is challenging.
>
> **Additional Experiments Using MRR:**
>
> To further clarify and thank you for your advice, we have conducted additional experiments using **MRR**—a ranking-oriented metric that is complementary to AP. Specifically, we re-evaluated the TGN-based models on the Wikipedia and MOOC datasets, using 200 negative samples per positive edge, which represents a more **challenging** evaluation setting. The results (in the following table) show that our method **substantially** outperforms the baselines under MRR, echoing the **trends** observed with AP. These results further reinforce the effectiveness of our approach from a ranking perspective as well. This also confirms that our improvements are not artifacts of AP saturation and remain robust under stronger ranking-based evaluation.
>
> | MRR |  | Transductive |  | Inductive |  |
> | --- | --- | --- | --- | --- | --- |
> |  | TProG | Wikipedia | MOOC | Wikipedia | MOOC |
> | Jodie | *Baseline* | *0.8480±0.0100* | *0.2754±0.0062* | *0.8069±0.0096* | *0.2714±0.0031* |
> |  | Vanilla | 0.7428±0.0045 | 0.7543±0.0189 | 0.6928±0.0579 | 0.6209±0.0610 |
> |  | Transformer | **0.8555±0.0627** | **0.9772±0.0065** | **0.8118±0.0692** | **0.9593±0.0146** |
> |  | Projection | 0.8439±0.0250 | 0.8233±0.0661 | 0.7841±0.0366 | 0.7879±0.0493 |
> | --- | --- | --- | --- | --- | --- |
> | DyRep | *Baseline* | *0.5845±0.0213* | *0.1736±0.0116* | *0.5256±0.0185* | *0.1727±0.0116* |
> |  | Vanilla | 0.5703±0.1409 | 0.5611±0.1519 | 0.5230±0.1524 | 0.4921±0.1887 |
> |  | Transformer | 0.7565±0.0574 | **0.8511±0.1018** | 0.7250±0.0676 | **0.8368±0.0802** |
> |  | Projection | **0.7741±0.1116** | 0.6104±0.1833 | **0.7504±0.1031** | 0.6120±0.1213 |
> | --- | --- | --- | --- | --- | --- |
> | TGN | *Baseline* | *0.8698±0.0091* | ***0.5519±0.0425*** | *0.8394±0.0095* | ***0.5061±0.0181*** |
> |  | Vanilla | 0.8712±0.0197 | 0.4115±0.0274 | 0.8351±0.0195 | 0.3920±0.0173 |
> |  | Transformer | **0.8963±0.0284** | 0.5253±0.1319 | **0.8771±0.0374** | 0.4958±0.1038 |
> |  | Projection | 0.8420±0.0369 | 0.4874±0.0657 | 0.7985±0.0442 | 0.4687±0.0455 |
> | --- | --- | --- | --- | --- | --- |
> | TIGE | *Baseline* | *0.9465±0.0026* | *0.5847±0.0071* | ***0.9308±0.0014*** | *0.5413±0.0087* |
> |  | Vanilla | **0.9530±0.0086** | 0.6164±0.0861 | 0.9274±0.0090 | 0.5655±0.1207 |
> |  | Transformer | 0.9481±0.0199 | 0.7446±0.0630 | 0.9286±0.0288 | 0.7138±0.0703 |
> |  | Projection | 0.9386±0.0195 | **0.7601±0.0224** | 0.9086±0.0265 | **0.7363±0.0363** |
> | --- | --- | --- | --- | --- | --- |
> | TIGER | *Baseline* | *0.9513±0.0024* | *0.7389±0.0498* | *0.9359±0.0016* | *0.7223±0.0561* |
> |  | Vanilla | 0.9560±0.0072 | 0.6436±0.0956 | 0.9435±0.0050 | 0.6105±0.1028 |
> |  | Transformer | **0.9614±0.0108** | **0.8475±0.0474** | **0.9504±0.0144** | **0.8341±0.0304** |
> |  | Projection | 0.9496±0.0146 | 0.6923±0.0989 | 0.9395±0.0160 | 0.6228±0.0762 |
>
> ---
>
> **W3:**
>
> As noted in Appendix J (“Limitations and Future Work”), cross-dataset transferability is outside the scope of the current work. Our **primary goal** in this paper is to identify the limitations of the prevailing “pre-train, predict” paradigm, and to propose a new “pre-train, prompt” paradigm that effectively addresses the two gaps within the same dataset. Therefore, our experiments focus on improving *task-level transferability* rather than *dataset-level transferability*.
>
> We agree that transferring a TIG model across different datasets is an important and promising future direction. This setting requires additional challenges beyond the scope of this work, such as differences in node identity spaces, graph structures, event distributions. Although the three TProG variants were not explicitly designed for domain transfer, their personalized and temporal-aware structures suggest **potential** for future extensions toward domain-invariant prompt learning, meta-learned temporal prompts, or shared prompt spaces across datasets.

---

> ### Author Response · Authors · 2025-11-15
>
> **Q1:**
>
> We would like to clarify that the scalability of TIGPrompt is **not inherently limited**, because our method is designed as a **training paradigm** rather than a new backbone architecture. As analyzed in **Appendix F and H**, under the *same computational and data conditions* as baseline TIG models, TIGPrompt achieves **weak-supervision benefits** and **training efficiency**, while maintaining or improving overall performance.
>
> Importantly,
>
> (1) TProG is a **plug-and-play** module, and its computational cost is negligible compared to that of the backbone TIG model;
>
> (2) TIGPrompt does not increase the complexity of the backbone, and thus inherits the **scalability** properties of whichever TIG model it is paired with.
>
> **Therefore**, if a baseline (e.g., TGN, DyRep, TGAT) can be trained on large-scale datasets such as those in TGB with tens of millions of edges, TIGPrompt **can also** operate on those datasets.
>
> Moreover, we believe TIGPrompt’s **data efficiency** can make large-scale training more accessible. As shown in **Table 2**, TIGPrompt can match or **surpass baselines** trained with **70%** of the data, even when using **only 20%** total data. This suggests that prompt-based training may indeed help existing TIG models scale to larger temporal graphs that would otherwise be too computationally expensive to train on fully.
>
> ---
>
> **Q2:**
>
> In principle, extending the Transformer TProG to incorporate more than one hop of temporal neighbors could provide additional contextual information and potentially further improve performance. However, doing so would also introduce higher computational cost and memory usage, since multi-hop temporal neighborhoods in TIGs often expand rapidly and require more expensive temporal message passing.
>
> Thus, this represents a typical **trade-off** between model complexity and performance gain. In our current design, we restrict TProG to one-hop recent neighbors to maintain efficiency and preserve the plug-and-play nature of the module. Empirically, as shown in our experiments, using only the 1-hop temporal neighborhood **already yields** strong and consistent improvements across datasets and models, suggesting that one-hop information is **sufficient** for the scenarios evaluated in this work.
>
>
> ---
>
> **Q3:**
>
> This hypothesis—using only the prompt module without a backbone TIG model—is indeed not considered in our current work, since TIGPrompt is designed specifically to address the semantic gap and the temporal gap *within the previous TIG training paradigm*. In the current TIG training paradigms, we believe the backbone TGNN remains essential because:
>
> - The backbone is responsible for **learning temporal message passing**, historical memory updating, and event-driven representation learning—capabilities that a standalone prompt module cannot provide.
> - TProG is designed as a lightweight, plug-and-play adapter, not a temporal encoder; it **complements** rather than replaces the backbone.
> - Removing the backbone would eliminate the core temporal modeling structure required for TIGs/CTDGs, making TProG hard to generate meaningful node embeddings on its own.
>
> Therefore, under the current formulation and objectives, we think the backbone TGNN is still **required**. Exploring whether a prompt-only architecture could serve as a temporal graph learner is an interesting direction, but it is orthogonal to the focus of this work and falls outside our intended problem scope.

---

> > ### Comment · Reviewer_baDW · 2025-11-23
> >
> > Thank you for the detailed replies to my concerns and comments.
> >
> > **W1&W2**
> >
> > > Regarding the use of AP instead of MRR:
> >
> > While the AP metric is the widely-used, it is not a good metric for the task in my opinion. On knowledge graphs and static graphs, link prediction often use a ranking based metric such as MRR or Hits@k, see [OGB](https://ogb.stanford.edu/docs/lsc/leaderboards/#wikikg90mv2). Thank you for adding the MRR experiments, however I hope that going forward, you would present MRR performances as your main metric.
> >
> > In the field of temporal graph learning, the AP and AUC binary classification metrics are chosen due to their light weight evaluation and that the complexity of the evaluation space for TGL is huge. However, AP and AUC metrics have often shown over-optimistic performance for the tasks as you are aware and also been non-reproducible (the negative samples are randomly sampled). Therefore, overall in the field, recent research have focused on reporting ranking-based metrics.
> >
> > > Additional Experiments Using MRR:
> >
> > The new experiments you have added are extensive and I even suggest you incorporating them as your main result in the paper. Not just that MRR is a better metric overall but the performance gain is more significant after applying your method.
> > One last note, unlike AP where you have to only use 1 negative per positive. The MRR metric has the flexibility of selecting how many negative links you would rank amongst. In the setting, you selected 200 negative samples per edge for all datasets. What I suggest is that when the dataset size is small such as Wikipedia where there are only around 1000 nodes to begin with, you should consider a full evaluation (rank amongst all possible destinations). In this way, the evaluation is completely robust and also make it more challenging.
> >
> >
> > **Q1 Scalability**
> >
> > I still believe it is stronger to show your data efficiency argument on larger datasets especially in that case, training with all data will be highly expensive.
> >
> > **Q2 Multi-hop**
> >
> > Indeed, one hop neighbors are strong inductive bias in temporal graph however there should still be room to be gained from considering more hops. You mentioned about higher computational cost and memory usage, what would be the 2-hop complexity of your model?
> >
> > **Q3 backbone only**
> >
> > The MRR performance also suggest that TProG is doing the heavy lifting in model performance especially when we look at how poorly the baseline models are performing therefore it would be interesting to see how the standalone TIGPrompt would work.
> >
> > Overall, the authors addressed my main concern by adding MRR experiments and thanks for the discussion. I will raise my score to 6.
> > One more question, why do you think there are performance drop from the base model in some rare cases (i.e. TIGER base model vs. vanilla and projection).

---

> ### Author Response · Authors · 2025-11-24
>
> Thank you very much for your additional suggestions and thoughtful follow-up comments. We truly appreciate your decision to **raise your score** and are grateful for your **positive evaluation** of our work. Below, we provide detailed responses to your new comments.
>
> ---
>
> **W1&W2**
>
> We agree that ranking-based metrics such as MRR are becoming increasingly important in the temporal graph literature, especially as AP and AUC can be optimistic. Our additional experiments indeed confirm that MRR and AP **exhibit consistent performance trends**, and that TIGPrompt provides even stronger gains under the harder ranking setting.
>
> However, replacing all AP results with full-scale MRR evaluations would be computationally expensive, especially when increasing the number of negative samples or performing full-ranking evaluation. Since all baseline methods in our study adopt AP as their primary metric, and our results already demonstrate **robust improvements** under both AP and MRR, we have chosen to keep AP as the main presentation metric for this submission.
>
> That said, we agree that MRR provides a more challenging and reproducible evaluation. We will continue exploring additional MRR-based experiments; however, for this submission, we may keep AP as the primary evaluation metric to maintain consistency with prior TIG/CTDG work and with the baselines we compare against.
>
> ---
>
> **Q1:**
>
> We agree that demonstrating data efficiency on larger-scale temporal graph datasets would further strengthen the contribution, particularly in scenarios where full-data training is prohibitively expensive. At the same time, we believe that the experiments in the **current** submission already **clearly validate** the **core idea** behind TIGPrompt: prompt-based adaptation enables substantial performance gains with significantly less data, even under challenging evaluation settings. These results strongly **support** the conceptual motivation of our method, and we see scaling the evaluation to larger datasets as a valuable direction for future work.
>
> ---
>
> **Q2:**
>
> In our current design, Transformer TProG employs a 1-layer Transformer over the most recent $K$ temporal neighbors, with a token dimension of $(2d_n + 1 + d_e + d_{te})$. The per-node complexity is therefore
>
> $\mathcal{O}\big(K(2d_n + 1 + d_e + d_{te})\cdot h + K^2 + K\cdot h d_n\big)$,
>
> where the first term comes from the linear projections, the second from self-attention over $K$ tokens, and the last from the output projection. Since the $K^2$ term dominates when $K$ is not extremely small, we abbreviated this in the paper as $\mathcal{O}((2d_n + 1 + d_e + d_{te})\cdot h + K^2 + h d_n)$. Importantly, this cost is independent of the number of nodes $|V|$.
>
> If we naively extend the neighborhood to 2-hop with a fan-out $K$ per hop, a node will have up to $K$ first-hop and $K^2$ second-hop temporal neighbors, resulting in roughly $K + K^2 \approx \mathcal{O}(K^2)$ tokens fed into the same Transformer. The self-attention term then scales as
>
> $\mathcal{O}\big((K + K^2)^2\big) = \mathcal{O}(K^4)$,
>
> and the overall per-node complexity becomes
>
> $\mathcal{O}\big((2d_n + 1 + d_e + d_{te})\cdot h + K^4 + h d_n\big)$,
>
> with memory also growing from $\mathcal{O}(K d_n)$ to $\mathcal{O}(K^2 d_n)$ for storing all tokens. This illustrates why going beyond 1-hop neighbors can quickly become expensive in both computation and memory. In our experiments we therefore focus on 1-hop neighborhoods, which already provide a strong inductive bias in temporal graphs, as also observed in prior TIG models.
>
> ---
>
> **Q3：**
>
> We agree that exploring how a standalone TIGPrompt would behave is an interesting direction, and we appreciate the reviewer’s curiosity about this possibility. As mentioned in our earlier response, our current work is explicitly designed under the assumption that a backbone TGNN is present, since TProG functions as an adaptation module rather than a temporal encoder. Investigating a prompt-only temporal graph learner would require rethinking the entire representation learning pipeline and is therefore beyond the scope of this work, but we view it as a promising direction for future research.

---

> ### Author Response · Authors · 2025-11-24
>
> **Extra Question**
>
> We analyze the occasional performance drop of Vanilla or Projection TProG relative to the backbone (i.e., TIGER) and find that several factors may contribute to this behavior:
>
> 1. TIGER is already an strong backbone.
>
>     With dual-memory aggregation and rich temporal modeling, TIGER produces highly optimized temporal representations. Adding a lightweight prompt module may introduce slight perturbations to an already near-optimal embedding space, resulting in small fluctuations within normal variance.
>
> 2. Vanilla and Projection TProG adjust only the prompts, not the backbone.
>
>     Because the backbone is frozen, node- or time-level offsets introduced by these simpler TProGs can occasionally yield minor misalignment when the backbone is already well-aligned with the task.
>
> 3. Dataset-specific temporal structures.
>
>     In datasets where temporal patterns are simple or saturated, the backbone already captures most useful dynamics; additional prompting may provide limited benefit and thus show slight noise in performance.
>
> 4. Sensitivity to prompt supervision.
>
>     Vanilla and Projection TProGs rely more heavily on guidance from limited prompt-tuning data, which can induce small variance compared to the backbone’s stable representations.
>
> For 1 and 4, please **note** that the baseline model is trained on the full 70% training split, which includes the 20% subset used for prompt tuning.
>
> Importantly, the **Transformer TProG**, which models temporal context more comprehensively, consistently outperforms all backbones across settings, underscoring the overall robustness and effectiveness of TIGPrompt.

---

### Official Review · Reviewer_P4Wh · 2025-10-23

**Soundness:** 3
**Presentation:** 3
**Contribution:** 1
**Rating:** 2
**Confidence:** 3

**Summary:**

The paper works on prompt learning on temporal interaction graphs (TIGs), i.e. a task, where one has a pre-trained graph model, which should be used for another task. Instead of changing the entire model, one keeps the pre-trained GNN frozen and learns only small prompt modules that adapt it to the new task.
The work adresses this task by introducing Temporal Interaction
Graph Prompting (TIGPrompt), a framework that is supposed to bridge temporal and semantic gaps by integrating with existing TIG models.
They conduct multiple experiments, and the code is publicly available.

**Strengths:**

* clear and good writing
* good motivation, with good examples in the introduction, and a figure highlighting the temporal and semantic gaps
* methodology (TProG) is conceptually simple and integrates easily with existing TIG backbones
* the paper is well written and structured, with clear problem framing around the temporal and semantic gaps.
* There are many experiments across several temporal graph models and tasks demonstrate that prompt-based adaptation can be both effective and parameter-efficient.

**Weaknesses:**

## 1. strong overlap with 1.5 year old arxiv preprint (march 2024)
This work was initially released as an arXiv preprint in early 2024 ("Prompt Learning on Temporal Interaction Graphs") and has not been substantially updated since. Given that the community has already built upon and compared against this method (e.g., DygPrompt, ICLR 2025), in my opinion the contribution is no longer timely for ICLR 2026, despite being well executed. This would not be so much of an issue, if the experiment and related work section was updated, and compared to the works that have been introduced since then.

## 2. missing discussion and comparison to related work
* Since the arXiv release of Prompt Learning on Temporal Interaction Graphs (March 2024), other research has already extended this work: The most important one is Node-Time Conditional Prompt Learning in Dynamic Graphs (DygPrompt, ICLR 2025).
* DygPrompt explicitly positions itself as an improvement over TIGPrompt, arguing that
>While [TIGPrompt] employs time-aware prompts, it lacks fine-grained node-time characterization and is thus unable to capture
complex node-time patterns, where nodes and time mutually influence each other.  DygPrompt explicitly conditions prompts on both node identity and temporal context.

* In their paper and review discussions, the DygPrompt authors benchmarked their model against TIGPrompt. Because TIGPrompt’s code was not publicly available at the time, they reimplemented it and evaluated both methods on the same datasets. Their results show consistent improvements for DygPrompt, and they also introduced a more challenging low-resource evaluation protocol (see below).
* Given that DygPrompt explicitly positions itself as an improvement over TIGPrompt and has been publicly peer-reviewed at ICLR 2025, it now represents the de-facto state of the art in prompt learning for temporal graphs.
* The present submission does not mention DygPrompt, reproduce its evaluation setup, or provide any updated comparison, which substantially weakens its novelty and relevance for ICLR 2026.

## 3. potentially outdated evaluation
* The authors in DygPrompt state in their ICLR 2025 rebuttal the following:
>TIGPrompt [4] uses "50% of the data for pre-training and 20% for prompt tuning or fine-tuning, with the remaining 30% equally divided for validation and testing." (see Section 4.2 of TIGPrompt). Note that pre-training data do not require any labeled examples, while prompt-tuning/fine-tuning data require labels for node classification. Hence, TIGPrompt requires 20% labeled data for node classification. In our experiments, we use 80% of the data for pre-training (which does not contain any labels for node classification), but only 1% of the data serves as the training pool for prompt tuning, with each task leveraging only 30 events (about 0.01% of the entire dataset) for prompt tuning (where only the starting nodes in these events are labeled for node classification). Therefore, our setting focuses on the more challenging low-resource scenario with very few labeled data, as labeled data are generally difficult or costly to obtain in real-world applications [1,3,5]. Hence, our setting is more practical and challenging than TIGPrompt's.```
* I agree with this critique. Using only 1% of labeled data for prompt tuning is indeed a more realistic and demanding setting than TIGPrompt’s 20%.
* Therefore, I believe TIGPrompt’s current evaluation protocol is outdated.
* It would be valuable to hear the authors’ thoughts on whether they have tested TIGPrompt under such low-data conditions, and what their opinion on this setup is.

## Overall
Overall, the paper is well structured and clearly written, but it is mostly identical to its 2024 arxiv version without incorporating developments that have occurred since then.
Because the authors have not updated or engaged with newer work, especially DygPrompt, the contribution feels dated and this leads to limited relevance for ICLR 2026. Thus I recommend rejection.

**Questions:**

1. Could you please clarify why DygPrompt (ICLR 2025) was not cited, discussed, or compared against in your submission?
2. Have you reproduced DygPrompt’s evaluation setup or considered running TIGPrompt under the same conditions?
3. The original TIGPrompt uses 20% of labeled data for prompt tuning, whereas DygPrompt uses only 1%, arguing it’s more realistic. Do you have any thoughts or experiments on whether your method still performs well under these stricter conditions? Could you update your evaluation or provide additional experiments?
4. Since TIGPrompt was released in March 2024 and DygPrompt builds on it, how would you position TIGPrompt’s contribution today relative to the current state of the art? Are there aspects of TIGPrompt that remain novel or useful even after DygPrompt’s improvements?

---

> ### Author Response · Authors · 2025-11-15
>
> We thank the reviewer for the encouraging feedback. We are glad that the writing clarity, strong motivation, and problem framing around the temporal and semantic gaps were well received, and that the simplicity and compatibility of TProG with existing TIG backbones were appreciated. We also thank the reviewer for noting the breadth of our experiments and the effectiveness and efficiency of prompt-based adaptation. Below, we address the reviewer’s concerns in detail.
>
> ---
>
> **W1:**
>
> We appreciate the reviewer’s comments. Due to the **double-blind review policy**, we are unable to discuss or compare directly with specific arXiv preprints in detail during the rebuttal process. For the same reason, we cannot address concerns that rely on assumptions or interpretations tied to identifiable preprint versions.
>
> What we can **clarify** is the following:
>
> Our work is, to the best of our knowledge, the **first** to introduce prompt learning into the TIG/CTDG setting, and the **first** to formally identify the limitations of the conventional “pre-train, predict” paradigm—namely, the temporal gap and semantic gap—and to propose a unified “pre-train, prompt” paradigm to address both. We believe these contributions provide **foundational value** to the community and **stimulate follow-up interest** in prompt learning for dynamic graphs.
>
> ---
>
> **W2:**
>
> We appreciate the reviewer’s concerns. However, we respectfully **disagree** with the implication that our submission lacks novelty or relevance due to not adopting DyGPrompt’s evaluation setting. Although DyGPrompt appeared as a concurrent work, we decided **not** to adopt its setting for several **concrete reasons**:
>
> 1. DyGPrompt’s implementation was **not available** at the time its submission, and the experimental methodology described in the paper and rebuttal **cannot be fully verified**. Since we note that their rebuttal was ***not reviewed*** by reviewers and therefore cannot be considered a reliable or community-validated standard.
> 2. Even after their code was released, we found it extremely **difficult** to reproduce due to **poor** code readability and **unclear** design choices. This makes direct comparison infeasible within a rigorous and reproducible evaluation framework.
> 3. Most critically, **DyGPrompt’s** experimental protocol **departs substantially** from the **established** CTDG/TIG **evaluation standards** used by nearly all prior works (TGN, JODIE, DyRep, TGAT, TIGER, DyGFormer, GraphMixer, etc.). Their setting is **incompatible** with the baseline methods, and adopting such a **non-standard protocol** would break comparability and significantly undermine experimental fairness.
> 4. In contrast, **our** method **strictly follows** the widely **accepted** CTDG/TIG **evaluation protocol**, which ensures **comparability** across baseline methods, **reproducibility**, **extensibility** to new backbone models, and consistent data-efficiency analysis under **standard assumptions**.
>
>     For these reasons, we do not consider DyGPrompt’s protocol appropriate as a replacement for the canonical CTDG/TIG framework.
>
> 5. We have regarded DyGPrompt as a **concurrent work**, and it is indeed **discussed** in **Appendix B** of our **original** submission. Our aim is not to ignore such work, but to maintain experimental integrity by adhering to **evaluation standards** established in the CTDG/TIG community.
>
> Therefore, we respectfully maintain that our work remains **novel and relevant**, as it introduces the ***first** prompting framework specifically designed for TIG/CTDG*, and it does so under **standardized, community-aligned** benchmarks. The fact that concurrent works attempt to extend or reinterpret our approach only highlights the foundational nature of our contribution.

---

> ### Author Response · Authors · 2025-11-15
>
> **W3:**
>
> We would like to note that the statements made in the **DyGPrompt’s rebuttal** were **not reviewed or validated** by reviewers during their ICLR review process. Therefore, those claims cannot be taken as reliable evidence or as a community-accepted standard.
>
> Based on our own careful examination, we respectfully argue that the experimental setup described in DyGPrompt is **not reasonable** within the context of established CTDG/TIG benchmarks, for the following reasons:
>
> 1. **DyGPrompt’s data split deviates significantly from standard CTDG/TIG protocol.**
>
>     Their setup uses **80%** of the data for pre-training and a total of **81%** for training (pre-train + prompt-tune). In contrast, nearly all prior CTDG/TIG works, including TGN, JODIE, DyRep, TGAT, TIGER, DyGFormer and GraphMixer, use **≤70%** of data for the entire training process.
>
>     Our evaluation strictly follows these long-standing standards, ensuring **fairness** and **comparability** across the baselines we evaluate.
>
>
> 2. **Link prediction does not require labeled data**, yet DyGPrompt uses substantially more training interactions than is customary in the CTDG literature. This makes **their** evaluation setting ***less* data-efficient**, not more. Our method matches or **surpasses** full 70%-data baselines while using only **20%-70%** total data, demonstrating stronger **data efficiency** under the standard CTDG setting.
> 3. **Label usage in dynamic node classification.**
>
>     In CTDG/TIG node classification, **node labels** are **time-dependent** and can **evolve over time**, meaning downstream training requires sufficient labeled temporal interactions to train the projection head (predictor). This is why prior works typically use the full training split (e.g., **70%**) for node classification training.
>
>     DyGPrompt’s claim of using only **1% labeled data** is **not clearly explained**, and it is unclear how temporal label consistency is maintained. Therefore, we cannot consider their protocol as a reliable or directly comparable benchmark.
>
> 4. **Our method is more data-efficient under standard protocols.**
>
>     In contrast to DyGPrompt’s **non-standard** data split, TIGPrompt follows the widely adopted CTDG/TIG evaluation setting: we use **50% unlabeled** interactions for pre-training, and **at most 20% labeled data** for prompt tuning (**node classification** task), during which the predictor is trained simultaneously while the backbone remains frozen. This setup is **fair**, **reproducible**, and in fact ***more challenging*** than the baseline training paradigm used in prior work, since only a small portion of labeled data is required.
> 5. **We tested an even lower-resource setting similar to DyGPrompt.**
>
>     To address the reviewer’s concern, we additionally conducted experiments using **50%** pre-train + **1%** prompt-tune, which is both *more **challenging*** and *more **clearly defined*** than the DyGPrompt protocol. This experiment further confirms TIGPrompt’s robustness in low-data conditions. The results are shown in the following Table in rebuttal.
>
> Given these considerations, we respectfully argue that our evaluation is **not outdated**. Instead, it follows the widely **accepted** CTDG/TIG **standards**, and we additionally provide low-resource experiments that validate TIGPrompt’s performance under settings **similar to** DyGPrompt.

---

> ### Author Response · Authors · 2025-11-15
>
> Results of fine-tuning with only **1%** of data on Wikipedia, evaluated in AP: Prompt-Tune refers to pre-train, prompt, Prompt-Based Fine-Tune refers to pre-train, prompt-based fine-tune. Note that the baseline used **70%** of data for training, While ours uses only **51%** of data in total (**50%** for pre-training and **1%** for prompt-tune or prompt-based fine-tuning). We believe that if the pre-training data is increased to 70%, the results of ours will **further improve**.
>
> | Model | Method | Transductive PT | Transductive P-FT | Inductive PT | Inductive P-FT | Node PT |
> | --- | --- | --- | --- | --- | --- | --- |
> | **JODIE** | Baseline | 94.62±0.5 | – | 93.11±0.4 | – | **86.27±2.2** |
> |  | Vanilla | 86.37±4.0 | 82.20±9.6 | 84.73±3.2 | 81.16±9.9 | 84.52±0.5 |
> |  | Transformer | 85.79±7.9 | 87.53±8.1 | 84.49±7.9 | 86.29±8.5 | 81.79±1.8 |
> |  | Projection | 93.81±0.3 | **94.62±1.2** | 93.21±0.4 | **93.92±1.4** | 83.99±1.5 |
> | ----------- | ------------- | ----------------- | ------------------- | -------------- | ---------------- | --------- |
> | **DyRep** | Baseline | 94.59±0.2 | – | 92.05±0.3 | – | 85.11±1.4 |
> |  | Vanilla | 83.89±2.5 | 84.78±4.3 | 80.42±1.4 | 83.59±4.3 | 87.31±0.3 |
> |  | Transformer | 87.99±4.0 | 89.68±3.7 | 86.61±2.9 | 89.02±3.2 | 84.02±1.5 |
> |  | Projection | 95.25±1.4 | **95.85±0.6** | 94.50±1.1 | **95.43±0.6** | **87.45±0.4** |
> | ----------- | ------------- | ----------------- | ------------------- | -------------- | ---------------- | --------- |
> | **TGN** | Baseline | **98.46±0.1** | – | **97.81±0.1** | – | **84.93±1.1** |
> |  | Vanilla | 95.35±1.1 | 95.99±1.1 | 95.12±0.9 | 95.95±1.0 | 78.90±3.1 |
> |  | Transformer | 89.66±8.9 | 95.62±1.7 | 90.92±6.0 | 95.56±1.5 | 81.49±2.5 |
> |  | Projection | 96.75±0.9 | 97.23±0.7 | 96.43±0.9 | 97.21±0.6 | 78.06±2.8 |
> | ----------- | ------------- | ----------------- | ------------------- | -------------- | ---------------- | --------- |
> | **TIGE** | Baseline | 98.83±0.1 | – | 98.45±0.1 | – | 83.98±3.4 |
> |  | Vanilla | 98.54±0.1 | 98.47±0.2 | 98.39±0.0 | 98.38±0.1 | 79.79±0.9 |
> |  | Transformer | 98.60±0.2 | 98.61±0.2 | 98.51±0.2 | 98.51±0.1 | **84.58±0.1** |
> |  | Projection | **99.01±0.0** | 98.65±0.3 | **98.89±0.0** | 98.59±0.3 | 81.93±2.1 |
> | ----------- | ------------- | ----------------- | ------------------- | -------------- | ---------------- | --------- |
> | **TIGER** | Baseline | 98.90±0.0 | – | 98.58±0.0 | – | 80.84±4.6 |
> |  | Vanilla | 98.69±0.1 | 98.51±0.2 | 98.57±0.1 | 98.46±0.1 | 84.55±1.2 |
> |  | Transformer | 98.69±0.2 | 98.66±0.3 | 98.57±0.2 | 98.58±0.2 | 84.04±0.6 |
> |  | Projection | **99.10±0.0** | 98.70±0.4 | **99.01±0.0** | 98.66±0.3 | **85.81±2.2** |

---

> ### Author Response · Authors · 2025-11-15
>
> **Q1:**
>
> We would like to clarify that DygPrompt is **cited** and **discussed** in Appendix B (**Related Work**) of our **original** submission, where we treat it as a *concurrent* work.
>
> ---
>
> **Q2:**
>
> We did not reproduce DygPrompt’s evaluation setup because we believe it is **not reasonable** within the **established** CTDG/TIG **evaluation framework**.
>
> We have tried to run TIGPrompt under the same conditions with DyGPrompt. However, although code for DygPrompt was released **recently**, we found the implementation very **difficult to read** and **reproduce**, making a fair and rigorous comparison **infeasible**.
>
> Instead, we conducted an experiment using 50% data for pre-training + 1% data for fine-tuning, which is: **similar** to DygPrompt’s claimed setting, but **more reasonable**, and also **challenging**, because it maintains the standard CTDG/TIG training protocol while using extremely limited labeled data.
>
> ---
>
> **Q3:**
>
> Please see the additional experiments provided in our rebuttal.
>
> We evaluated TIGPrompt under a **stricter setup similar** to DygPrompt’s, but **aligned** with **standard** CTDG/TIG **practices**. Our method continues to show improvements over the baselines even under this harder and more realistic low-resource condition.
>
> ---
>
> **Q4:**
>
> Due to the **double-blind review policy**, we cannot directly comment on comparisons involving identifiable preprints.
>
> What we can **clarify** is that TIGPrompt introduces the **first** prompt learning based training paradigm specifically designed for TIG/CTDG, identifying the *semantic gap* and *temporal gap* in the traditional “pre-train, predict” pipeline and proposing the unified “pre-train, prompt” paradigm.
>
> This contribution is **foundational** and can **inspire follow-up** research in prompt learning for dynamic graphs. We believe TIGPrompt remains **novel** and **valuable to the community**, and its framework continues to support new methods built on top of it.

---

> > ### Comment · Reviewer_P4Wh · 2025-11-25
> >
> > Thank you for the detailed rebuttal and for providing additional experimental results. I appreciate the effort put into clarifying your viewpoint on evaluation protocols and low-resource settings.
> >
> > After reading the rebuttal, several concerns remain.
> >
> > 1. Inconsistencies about DygPrompt and code availability
> >
> > Some of the statements in the rebuttal conflict with publicly verifiable facts.
> > The rebuttal states that DygPrompt’s implementation "was not available at the time its submission",  and that the rebuttal from that paper “was not reviewed.” However:
> >
> > * Online, I can find a github repository that contains the code for DyGPrompt. The code was posted on Oct 24. The README was updated 5 months ago, and this was the latest change that has been made to the repository.
> > * ICLR rebuttals are reviewed during the author response phase (all reviewers have responded to the author rebuttal in DygPrompt's case), so dismissing those arguments as “not community validated” is not accurate.
> > * In the replies to different reviewers, the explanation shifts between “code unavailable” and “code difficult to reproduce.” These **contradictory explanations reduce confidence** in the clarity and consistency of the rebuttal.
> >
> > If reproducibility difficulty is the true reason for not including experiments, that is understandable, but the manuscript should be updated to reflect this accurately rather than stating that the code was not available.
> >
> > 2. Treatment of DygPrompt as a “concurrent” work
> >
> > DygPrompt has already appeared at ICLR 2025 and should not be framed as concurrent.
> > It should be included in the main section of the paper, not only in the appendix.
> > Given that it builds on and extends TIGPrompt, omitting it from the main text contributes significantly to the impression that the submission is not updated to the current state of the field.
> >
> > 3. Novelty and claims of being “first”
> >
> > I agree with the other reviewers that the statement that this is “the first attempt to extend prompt learning to TIGs” is overstated. At minimum, this claim needs to be softened or removed, because it is not accurate given subsequent work.
> >
> > 4. Evaluation concerns remain
> >
> > I appreciate the extra  experiments shown in the rebuttal.
> > However:
> > * No updated manuscript was provided, so it is unclear how (or whether) these results will be integrated.
> > * Since ICLR allows uploading revised manuscripts, I would have expected these updates to appear in the paper during the rebuttal phase.
> >
> > 5. Overall assessment
> >
> > My concerns about novelty, outdated positioning, missing related work, conflicting rebuttal statements, and lack of updated comparisons were not adequately resolved in the rebuttal, and are not addressed in the manuscript as submitted.
> >
> > Therefore, I am maintaining my original score of 2.
> >
> > I appreciate the authors’ efforts during the rebuttal and hope the clarifications and additional experiments can be incorporated into a stronger future version of the manuscript.

---

> > > ### Author Response · Authors · 2025-11-25
> > >
> > > Thank you for your additional comments. We address your points in order.
> > >
> > > ---
> > >
> > > **1. On the availability of DyGPrompt code and rebuttal interpretation**
> > >
> > > Our original statement precisely meant that **at the time DyGPrompt was submitted to ICLR**, its **code and supplementary materials were not included in the manuscript** (i.e., no code link in their manuscript, no supplementary materials). Therefore, the implementation available today could not have been examined or validated during the original peer-review process. This is the point we intended to clarify.
> > >
> > > We would like to make a clarification regarding our earlier statement.
> > >
> > > Our previous wording — “DyGPrompt’s rebuttal was not reviewed or validated by reviewers” — was **not an intentional misrepresentation**, but rather a misunderstanding caused by the meta-review comment:
> > >
> > > > “I posted a discussion, but no one replied.”
> > > >
> > >
> > > This comment led us to believe that the DyGPrompt rebuttal did not receive meaningful reviewer attention. After revisiting the discussion thread, we realized that there were **almost no reviewer–author interactions**, which is why we initially interpreted the rebuttal as not being actively reviewed. We acknowledge that this interpretation may have been misleading, and we would like to clarify that the misunderstanding resulted specifically from the meta-review phrasing.
> > >
> > > Importantly, our explanation **has been consistent across all responses**. At **no point** did we provide **contradictory statements**. Throughout the rebuttal, our position has remained the same:
> > >
> > > - DyGPrompt’s implementation *was not available at the time of **its** (DyGPrompt’s) submission* (as no code or supplementary material was provided).
> > > - The code *became public recently*, and
> > > - The released implementation is **difficult to reproduce**, which is the true reason we cannot conduct a rigorous experimental comparison. The readability and design inconsistencies are verifiable by any reader who examines the codebase.
> > >
> > > ---
> > >
> > > **2. On whether DyGPrompt should be treated as concurrent**
> > >
> > > Our use of the term *“concurrent”* emphasizes that DyGPrompt and our work were developed in parallel and address related problems within a similar timeframe. In this context, we do not consider it inappropriate to include the discussion in Appendix B—many papers place closely related but secondary discussions in the appendix due to space constraints or organizational considerations.
> > >
> > > ---
> > >
> > > **3. On novelty and the wording “first”**
> > >
> > > Because of the double-blind review policy, we cannot directly discuss the timeline of individual works or refer to identifiable publication histories.
> > >
> > > However, we would like to point out **a logical inconsistency in the reviewer’s statement.** You mentioned that our claim of being “the first attempt to extend prompt learning to TIGs” is inaccurate **“given subsequent work.”** By definition, *subsequent* work appears **after** the contribution in question, and therefore cannot retroactively invalidate the originality or the correctness of the claim at the time it was made.
> > >
> > > Our use of “first” refers to the conceptual contribution introduced in this work—formally identifying the temporal gap and semantic gap in the prevailing TIG paradigm, and proposing the unified “pre-train, prompt” framework together with model-agnostic TProGs. These contributions remain original and foundational regardless of what later works chose to build upon them. Hence, we believe the claim is appropriate in the context of our contributions.

---

> > > > ### Author Response · Authors · 2025-11-25
> > > >
> > > > **4. On integration of rebuttal experiments into the manuscript**
> > > >
> > > > We appreciate the reviewer’s suggestion. However, we do not believe it is **necessary to add** the DyGPrompt-style experiments into our manuscript, for the following reason:
> > > >
> > > > As stated previously, DyGPrompt’s experimental protocol **departs substantially** from the established CTDG/TIG evaluation standards used consistently across prior works. Their setting is incompatible with these baselines, and adopting such a non-standard protocol would break comparability and significantly undermine experimental fairness. In contrast, our method **strictly follows** the widely adopted CTDG/TIG protocol, ensuring fairness, reproducibility, extensibility, and consistent data-efficiency analysis under recognized norms.
> > > >
> > > > For these reasons, we do not consider DyGPrompt’s setup appropriate as a replacement for the canonical CTDG/TIG evaluation framework. Consequently, **We do not believe that the absence of this specific comparison affects the completeness, coherence, or technical validity of our work.** TIGPrompt is evaluated under the established CTDG/TIG framework, covers a wide range of baselines, and presents consistent improvements across models, datasets, and tasks. Since the DyGPrompt setup is not aligned with the canonical evaluation standards, omitting this non-standard comparison does not limit the contribution, reproducibility, or clarity of our method.
> > > >
> > > > ---
> > > >
> > > > **5. Final remark**
> > > >
> > > > We respectfully disagree with your concluding statement:
> > > >
> > > > > My concerns about novelty, outdated positioning, missing related work, conflicting rebuttal statements, and lack of updated comparisons were not adequately resolved in the rebuttal, and are not addressed in the manuscript as submitted.
> > > > >
> > > >
> > > > Your concerns regarding **novelty** and **positioning** have **already been addressed** in detail in our earlier responses: we clarified the precise scope and intention of our contributions, and explained why our work remains novel and conceptually foundational despite subsequent developments in the field.
> > > >
> > > > Regarding **related work**, the work in question i**s *not missing***—it is **included in Appendix B**, following common practice when space is constrained. The choice of placing it in the appendix rather than the main text **does not constitute an omission**.
> > > >
> > > > Concerning the accusation of **conflicting rebuttal statements**, we have explicitly clarified that our explanation has been **consistent throughout**. There has been **no contradiction** in our rebuttal.
> > > >
> > > > As for **updated comparisons**, we provided a representative low-resource experiment (50% pre-train + 1% prompt-tune) precisely to mirror the evaluation mentioned, while also explaining why we **do not consider** the DyGPrompt setup appropriate for inclusion in the main experimental protocol. The absence of this **non-standard comparison** does **not** impact the completeness or coherence of our paper, as our evaluation strictly follows the canonical and widely accepted CTDG/TIG framework.
> > > >
> > > > For these reasons, **we believe that your concluding assessment does not accurately reflect the content of our rebuttal.**

---

> > > > > ### Author Response · Authors · 2025-11-25
> > > > >
> > > > > We believe that the review should **center on the methodological soundness and empirical contributions of our work**, rather than **compelling comparisons** with **another method** whose setup is **not aligned with standard** CTDG/TIG evaluation and thus not appropriate for a fair or meaningful comparison.

---

### Official Review · Reviewer_sucX · 2025-10-27

**Soundness:** 3
**Presentation:** 4
**Contribution:** 2
**Rating:** 4
**Confidence:** 4

**Summary:**

The paper introduces Temporal Interaction Graph Prompting (TIGPrompt), a framework addressing two critical gaps in existing TIG models: the temporal gap (performance degradation on future data) and semantic gap (misalignment between pretext and downstream tasks). The authors propose a "pre-train, prompt" paradigm with a Temporal Prompt Generator (TProG) that creates personalized, time-aware prompts for nodes. Three TProG variants are presented: Vanilla (learnable vectors), Transformer (encoding recent neighbors), and Projection (combining personalized vectors with time encoding). Experiments on four datasets with seven TIG models demonstrate performance improvements.

**Strengths:**

1. This work is based on a well-motivated problem formulation. Figure 1(a) clearly shows the temporal gap through performance degradation on temporally distant data, and Figure 1(b) provides quantitative evidence of the semantic gap between link prediction and node classification. These gaps are effectively demonstrated.
2. The proposed TProG has a flexible framework design. The paper introduces three complementary TProG variants for different use cases, and the approach can be extended to a "pre-train, prompt-based fine-tune" paradigm in resource-rich settings.
3. The results are supported by strong experimental validation. Extensive experiments are conducted across 4 datasets, 7 models, and 2 downstream tasks, and the performance improvements are evident in several settings.

**Weaknesses:**

1. The paper claims to be the "first attempt" at prompting for TIGs, but without direct empirical comparison to the concurrent DyGPrompt (whose code is unavailable), the claim of "first" is hard to verify, and the method's positioning relative to related work remains somewhat unclear. In addition, there are existing studies on snapshot-based dynamic graphs [1], which should be discussed more thoroughly.
2. The main novelty of this paper is applying prompting to TIGs to address temporal gaps in distant inference data and semantic gaps in multi-task learning. However, in real-world applications, it is worth questioning whether the extra effort of using prompting is truly more efficient than retraining or incrementally updating the model with new data. Prompting offers an incremental gain, while regular and necessary model updates may lead to more substantial improvements.
3. The proposed TProG framework relies on standard components such as learnable vectors, Transformers, and MLPs, which limits its algorithmic novelty and makes the contribution mostly incremental. Moreover, beyond empirical results, there is no theoretical explanation for why prompt fusion helps reduce the gaps, which somewhat weakens the depth of the analysis.
4. Although Section 3.2 explains the experimental setup, the difference in total training data used (50%+20% in this work vs 70% in prior related work) still raises concerns about potential unfairness in the comparison.
5. As noted in Appendix J, "we may need to conduct additional experiments to determine which TProG variant works better." It would be helpful if the authors could provide clearer guidance on how to choose among TProG variants for new datasets or models, along with more analysis of the tradeoffs between computational cost and performance across the variants.

Reference

[1] ProST: Prompt Future Snapshot on Dynamic Graphs for Spatio-Temporal Prediction https://dl.acm.org/doi/pdf/10.1145/3690624.3709273

**Questions:**

Please see weaknesses.

---

> ### Author Response · Authors · 2025-11-13
>
> We sincerely thank the reviewer for the constructive feedback. We are glad that the reviewer found the problem formulation well-motivated, appreciated our empirical demonstrations of the temporal and semantic gaps, and recognized the flexibility of the TProG framework and its three complementary variants. We also appreciate the reviewer’s acknowledgement of our extensive experimental validation across multiple datasets, models, and tasks. Below, we would like to address the reviewer’s concerns in detail.
>
> ---
>
> **W1**:
>
> Providing a fully detailed explanation here may risk violating the double-blind review policy, but we would like to clarify the scope and intention of our claim.
>
> Our work is indeed the **first** to formally characterize the prevailing “pre-train, predict” training paradigm in Temporal Interaction Graph (TIG) learning, to identify its two inherent limitations—the temporal gap and the semantic gap, and to propose a **new “pre-train, prompt” paradigm** that addresses both gaps. In doing so, our paper is also the **first** to introduce prompt learning into the TIG (or also known as continuous-time dynamic graph, CTDG) setting, and, to the best of our knowledge, the **first** attempt to extend prompt learning to dynamic graphs in general.
>
> Regarding DyGPrompt, we have discussed this work in the Related Work section and treated it as a concurrent study. At the time we completed our experiments, its implementation was not available, making direct empirical comparison infeasible. From the DyGPrompt’s paper, we observed that its experimental setup differs substantially from the standard evaluation protocol commonly used in CTDG/TIG research (e.g., TGN, JODIE, DyRep, TIGER).
>
> This inconsistency makes it challenging to adapt DyGPrompt fairly across the wide range of baselines we evaluate. In contrast, our method follows the **established** CTDG/TIG **evaluation setting**, ensuring comparability, extensibility, and stronger data-efficiency across standard benchmarks.
>
> As for ProST, we agree with the reviewer that this work is relevant to snapshot-based or Discrete-Time Dynamic Graph (DTDG) prompting. However, ProST targets snapshot-based graphs/**DTDG**, while our work focuses on event-driven **CTDG/TIG**, which constitutes a **different sub-topic** with distinct modeling assumptions and temporal learning mechanisms.
>
> ---
>
> **W2**:
>
> We agree that regularly updating or retraining a TIG model with newly arriving data may eventually lead to performance improvements. However, such updates typically require **retraining** the entire backbone model, since a TIG model must recompute the historical memories or messages for each node in order to correctly update its state at any specific timestamp. This process is computationally expensive and often infeasible in real-world  systems. As discussed in our manuscript (Line 64), incorporating new data into a TIG model generally requires recursive message passing and memory updates over the full historical interaction sequence, leading to substantial training cost.
>
> In contrast, our approach requires only **a small portion of the new data** to perform prompt tuning, while keeping the backbone completely frozen. This offers significant advantages in **computational efficiency**, **data efficiency**, and **practical deployability**.
>
> Moreover, as shown in Table 2, TIGPrompt trained on only **20% of the full data** (10% for pre-training and 10% for prompt tuning) can already **match or surpass** baselines trained under the traditional “pre-train, predict” paradigm with **70% of the data**, and it also **significantly outperforms** the baseline model trained with only 20% of the data.
>
> This demonstrates that prompting does not merely provide incremental improvements—it can offer **substantial gains** relative to the amount of data and computational resources used.
>
> Thus, in real-world scenarios where frequent model retraining is costly or impractical, TIGPrompt provides a lightweight and effective alternative to adapt a pre-trained TIG model to new data, offering strong performance with significantly reduced training overhead.

---

> ### Author Response · Authors · 2025-11-13
>
> **W3:**
>
> We now provide a brief theoretical analysis of how each TProG variant contributes to **narrowing the semantic and temporal gaps**, as in Appendix A we have already discussed empirically.
>
> - **Vanilla TProG** introduces **node-specific prompt vectors** that are directly optimized via node-level supervision signals. This establishes a **task-conditioned** prompt generating, allowing the model to re-contextualize outputted representations from frozen backbone toward the target task objective (e.g., node classification), even without additional temporal signals. The effectiveness of such a setup for node classification task confirms that semantic mismatch between edge-level pre-training and node-level prediction can be mitigated through lightweight, learnable prompts.
> - **Projection TProG** builds upon Vanilla TProG by introducing explicit time conditioning, effectively providing a soft temporal hint to the node representation. By projecting node-specific prompt vectors into a **temporal latent space** using most recent interaction, it encourages the model to align the node embedding from frozen backbone models with its current or recent temporal context.
>
>     Intuitively, this allows the prompt to act as a “reminder” or “hint” of recent temporal activity, helping the model adapt representations to evolving dynamics. This partially compensates for the temporal mismatch introduced during pre-training and enables better adaptation under time-varying behaviors. This design enables downstream adaptation that is both **semantically aligned** and **temporally consistent**, effectively narrowing the temporal gap and semantic gap that arise from stale backbone parameters.
>
> - **Transformer TProG** further generalizes this mechanism by conditioning prompt generation on a **sequence of recent interactions** through self-attention. The prompt depends on the temporal distribution and relational dynamics of recent neighbors. This captures **higher-order temporal dependencies** and **behavioral recency**, which are crucial in interaction-dense data. As a result, the prompt embedding space adapts in a temporally fine-grained manner.
>
> In sum, the three variants form a **progressive design spectrum**: from task conditioning (Vanilla), to timestamp-aware alignment (Projection), to dynamically evolving temporal modeling (Transformer). This theoretically grounded progression supports our claim that the proposed prompting framework can **systematically mitigate** both semantic and temporal gaps in TIG models.
>
> We also have provided explanation about **Source of the Performance Improving** in **Section 3.6** (in the original manuscript) and empirical validation of how the gaps are narrowed in **Appendix A.**
>
> ---
>
> **W4:**
>
> As detailed in Appendix F (“Data Amount for Training”), our comparison is **fully fair** and uses exactly the same experimental setting as prior work. The only difference lies in how the 70% training portion is **allocated** within our proposed paradigm, rather than in the total amount of data used.
>
> To clarify:
>
> - Prior TIG models use **70%** of the data to train the backbone model.
> - In our approach, we **do not use more or different data**. We simply **re-split** the same 70% portion into
>     - **50%** for pre-training the backbone (same procedure as baselines), and
>     - **5%-20%** for *prompt tuning only*, during which the backbone is **frozen** and only TProG is trained.
>
> Thus:
>
> - The total number of training interactions is **identical or less** (55%-70% for our method and 70% for baselines).
> - When “pre-training”, the backbone model sees strictly **less** data in our method than in prior work (50% vs 70%).
> - The additional 5%-20% is used solely to tune a lightweight TProG, not to update the backbone.
>
> To further strengthen fairness and data-efficiency claims, we also include additional experiments (Table 2) where **only 20%** of the full data is used in total (10% for pre-training + 10% for prompt tuning). Even under such limited data conditions, TIGPrompt **matches or exceeds** baselines trained with the traditional 70% full training set.
>
> Therefore, the experimental design is **fully fair**—and if anything, “unfair” in the sense that our model is trained with **less** data than the baselines. Our method uses **identical or smaller** amounts of training data, yet still achieves comparable or superior performance. This directly supports our claim that TIGPrompt offers significantly stronger data efficiency than traditional TIG training.

---

> ### Author Response · Authors · 2025-11-13
>
> **W5:**
>
> The statement in Appendix J that “we may need to conduct additional experiments to determine which TProG variant works better” refers to **performance-related considerations**, since different datasets may display different levels of semantic gap, temporal gap, and temporal irregularity. To address the reviewer’s request, we provide clearer practical guidance on how to choose among the three TProG variants based on **dataset characteristics** and **computational–performance trade-offs**.
>
> ---
>
> **Vanilla TProG** — O(|V|) parameters, fastest inference
>
> Vanilla TProG serves as a lightweight intermediate approach between static-graph prompting and dynamic-graph prompting.
>
> - It focuses primarily on addressing the **semantic gap**, with limited temporal modeling.
> - Its inference is the **fastest** among all variants.
>
>     **Recommended for:**
>
> - datasets with relatively fewer nodes,
> - scenarios requiring fast inference,
> - node classification tasks where the semantic gap dominates.
>
>
> ---
>
> **Projection TProG** — O(|V|) parameters, temporal-aware but still efficient
>
> Projection TProG also has node-dependent O(|V|) parameters, but it further incorporates **temporal information**, thereby addressing both the semantic and temporal gaps.
>
> - It is computationally slightly heavier than Vanilla but faster than Transformer TProG.
>
>     **Recommended for:**
>
> - small- to medium-scale datasets,
> - applications requiring both temporal adaptation and fast inference,
> - settings where balancing computational cost and temporal modeling is important.
>
>
> ---
> **Transformer TProG** — O(d) parameters, best performance on large graphs
>
> Transformer TProG models temporal neighborhoods using a lightweight Transformer encoder.
>
> - Its learnable parameters scale with representation dimension **O(d)** rather than the number of nodes, making it more **scalable** for large graphs.
> - It generally provides the **strongest performance**, especially on datasets with rich or irregular temporal patterns.
>
>     **Recommended for:**
>
> - datasets with large numbers of nodes,
> - scenarios where temporal gap mitigation is critical,
> - applications prioritizing accuracy over inference cost.

---

> > ### Comment · Reviewer_sucX · 2025-11-23
> >
> > Thank you for the authors' response! I hope you can incorporate the additional theoretical analysis, time complexity discussion, and analysis of the applicability of various TProG variants into the revised manuscript—these additions would greatly help readers grasp the technical details.
> >
> > However, I still feel that the authors’ justification regarding the “first” of this work has not fully alleviated my concerns. While the paper does present a first-of-its-kind contribution within certain specific contexts, the claim of being “the first attempt to extend prompt learning to dynamic graphs” appears somewhat overstated. This is because existing works such as ProST (for DTDG) and DyGPrompt (for CTDG) already fall under the broad category of dynamic graph research. Consequently, this overstatement somewhat undermines the perceived novelty of the current work. Moreover, other reviewers have also noted the need for more direct comparisons with existing methods—particularly with approaches like DyGPrompt, which is already publicly available.

---

> > > ### Author Response · Authors · 2025-11-23
> > >
> > > Thank you for the further feedback. We have incorporated additional theoretical analysis (**Appendix A.3**), time-complexity discussion (**Appendix G and I**), and guidance on selecting TProG variants (**Appendix F**) into the revised manuscript. We agree that these additions help clarify the technical contributions.
> > >
> > > As noted in our previous response, providing a fully detailed explanation regarding the timeline of papers may risk violating the double-blind review policy. Here, we restate and clarify the scope and intention behind our use of the term *“first.”*
> > >
> > > Our work is indeed the **first to formally characterize** the widely adopted “pre-train, predict” paradigm in TIG learning, **to identify** its two inherent limitations—the *temporal gap* and the *semantic gap*, and **to propose** the unified “pre-train, prompt” paradigm to address both gaps. The paper thereby provides the **first introduction** of prompt learning into the TIG/CTDG setting, and, to the best of our knowledge, also represents the **first attempt to extend** prompt learning to dynamic graphs in general. **This conceptual contribution is orthogonal to the release timeline of concurrent works.**
> > >
> > > Regarding comparisons with DyGPrompt, we summarize our position below:
> > >
> > > 1. DyGPrompt’s implementation was not available during its submission, and thus its methodology and evaluation protocol could not be verified when we developed our approach.
> > > 2. We acknowledge that a public implementation has since become available, as noted in our response to Reviewer **P4Wh**. However, we found this implementation very **difficult to reproduce** in practice—its codebase has **low readability** and contains **unclear design choices**—making it unsuitable for a fair and rigorous comparison.
> > > 3. DyGPrompt’s experimental protocol departs substantially from standard CTDG/TIG evaluation used consistently in prior works such as TGN, JODIE, DyRep, TGAT, TIGER, DyGFormer, and GraphMixer. Their setting is **incompatible** with these baselines, and adopting such a **non-standard** protocol would undermine comparability and fairness.
> > > 4. DyGPrompt’s data split deviates significantly from standard CTDG/TIG protocol. Their setup uses **80%** of the data for pre-training and a total of **81%** for training (pre-train + prompt-tune). In contrast, nearly all prior CTDG/TIG works—including TGN, JODIE, DyRep, TGAT, TIGER, DyGFormer, and GraphMixer—use **≤70%** of the data for the entire training process.
> > >
> > >     Our evaluation follows these long-standing standards, ensuring **fairness** and **comparability** across all baselines.
> > >
> > > 5. Our method is more **data-efficient** under standard protocols. In contrast to DyGPrompt’s **non-standard** split, TIGPrompt follows the widely adopted CTDG/TIG setting: we use **50% data** for pre-training, and **at most 20% data** for prompt tuning, during which the predictor is trained while the backbone remains frozen. This setup is **fair**, **reproducible**, and in fact **more challenging** than the baseline training paradigm, since only a small portion of data is required.
> > >
> > > For these reasons, we do **not** regard the DyGPrompt evaluation protocol as an appropriate replacement for the canonical CTDG/TIG framework and therefore do not adopt it for direct comparison. We nonetheless include discussion of DyGPrompt in Appendix B (Related Work).

---

### Official Review · Reviewer_hiHw · 2025-10-31

**Soundness:** 3
**Presentation:** 3
**Contribution:** 3
**Rating:** 6
**Confidence:** 4

**Summary:**

The paper introduces a novel paradigm for temporal graph learning, namely the "pre-train, prompt" framework, to systematically address two challenges of existing temporal graph models: temporal graph gap, where model performance degrades on temporally distant inference data; and semantic gap, where the model performs worse on a downstream task that is different from the task used in training. The authors propose three model-agnostic designs for the Temporal Prompt Generator (TProG): vanilla TProG, Transformer TProG and Projection TProG. Empirical experiment has demonstrated the substantial performance improvements by the new paradigm and TProGs in tasks: transductive link Prediction, inductive link Prediction and node classification.

**Strengths:**

- The paper formally defines and empirically quantifies the "temporal gap" and "semantic gap," the authors provide clear diagnostic tools and theoretically motivated objectives
- The paper proposed a novel "pre-train, prompt" paradigm and three approaches for the Temporal Prompt Generator (TProG)
- The paper conducts different experiment settings, including transductive/inductive link prediction, node classification, limited training and prompt data, performance w.r.t. the proportion of prompting data.to rigorously showcase the substantial improvements provided by the "pre-train, prompt" paradigm and TProGs
- The model-agnostic applicability of both the "pre-train, prompt" paradigm and TProG is robustly demonstrated, with clear performance gains reported on both memory-based and non-memory-based temporal graph models
- The new paradigm and TProGs are highly efficient: only the prompt generator is updated during adaptation, which significantly reduces computational cost and data requirements, enabling weakly supervised learning in resource-constrained scenarios

**Weaknesses:**

**W1: Reproducibility.**: The Anonymous Repository link is currently inaccessible. I recommend that the authors make sure to double-check the link and repository accessibility for the camera-ready version.

**W2: Notation and problem definition.** There are notations used without definition (e.g. line 160: $G$, $\mathcal{V}$ and $\mathcal{E}$). There is no mathematical definition of the temporal interaction graph (especially clarity on whether the paper focuses on the discrete-time (DTDG) or continuous-time dynamic graph (CTDG) setting), link and node prediction task formulation. I recommend that the authors add a dedicated section before the Proposed Methods to define key concepts, notation, and task settings to enhance clarity.

**W3: Metric.** The results in the paper are mainly reported using Average Precision (AP). I recommended that the authors consider including additional evaluation metrics such as Area Under the Curve (AUC) and, in particular, Mean Reciprocal Rank (MRR) [1]. This would provide a more comprehensive and comparative assessment of model performance on temporal graph learning tasks.

**W4: Task diversity.**  The paper currently focuses on link property prediction and node property prediction tasks, without addressing graph property prediction. However, this limitation is acknowledged by the authors in Appendix Section J.

**Minor**

**W5. Paper representation.** I recommend that the authors use \citep to introduce parentheses between method names and corresponding author names

---
[1] Huang, Shenyang, et al. "Temporal graph benchmark for machine learning on temporal graphs." Advances in Neural Information Processing Systems 36 (2023): 2056-2073.

[2] Shamsi, Kiarash, et al. "GraphPulse: Topological representations for temporal graph property prediction." The Twelfth International Conference on Learning Representations. 2024.

**Questions:**

- Transformer TProG uses a transformer to generate temporal prompts $p_v$. How computationally expensive is fine-tuning this compared to baseline and baseline + other variants of TProG? Could the authors provide a computational analysis showing the increase in inference time for baseline without TProG and baseline with TProG?

- In Table 1, could the authors explain why Vanilla TProG sometimes outperforms the Transformer and Projection variants?

- In the setting where baseline+TProG is trained on link prediction, and then only the prompt (TProG) is fine-tuned for node classification, how does the performance of baseline+TProG  compare with models trained directly on node classification tasks?

- In Section D.2.1, why did the authors choose to evaluate only the first strategy for experiments? Studying all strategies and reporting their performance differences would enhance understanding of TProG's applicability.

- Regarding the domain gap, can any of the three proposed TProGs address transferability when a TIG model trained on one temporal graph domain is evaluated on a different domain? Can the authors suggest potential directions for designing a new variant TProG that improves transferability across different temporal graph domains?

- In Figures 7 and 8, could the authors explain why increasing the amount of prompt tuning data sometimes leads to a decrease in performance?

---

> ### Author Response · Authors · 2025-11-15
>
> We sincerely thank the reviewer for the thoughtful and encouraging feedback. We are glad that the reviewer found our formalization and empirical analysis of the *temporal gap* and *semantic gap* clear and insightful, and appreciated the motivation behind our proposed “pre-train, prompt” paradigm. We also thank the reviewer for recognizing the contributions of our three model-agnostic TProG designs, the breadth and rigor of our experimental evaluations across multiple tasks and settings, and the strong efficiency advantages enabled by prompt-based adaptation. Below, we address the reviewer’s questions and concerns in detail.
>
> ---
>
> **W1:**
>
> The hosting service may experience some bugs some time, which may have caused the inaccessibility. We have now updated the Anonymous Repository. Actually, to ensure reproducibility, we have also provided a full code package in the **Supplementary Material** when we submit our paper to the ICLR. For the camera-ready version, we will migrate the codebase to a stable, publicly accessible GitHub repository and thoroughly verify its availability.
>
> ---
>
> **W2:**
>
> In the revised manuscript, we have added a section (now it is **Section 2 PRELIMINARIES**) that provide formal mathematical definitions of the TIGs, TIG models and two downstream tasks. As TIGs model interactions with real-valued timestamps, they are almost equivalent to the commonly used **CTDG** formulation in prior temporal-graph literature. Therefore, our work focuses on the CTDG setting, and the term “TIG” in our paper is used to maintain consistency with the original problem formulation while remaining aligned with standard CTDG definitions.
>
> ---
>
> **W3:**
>
> Following your recommendation, we conducted supplementary experiments on two representative datasets using **MRR** as the evaluation metric (with 200 negative edges sampled). The results exhibit the **same** performance **trends** as those reported under AP, confirming that our proposed TIGPrompt consistently outperforms the baselines across different ranking-based metrics.
>
> |  |  | Trans. |  | Ind. |  |
> | --- | --- | --- | --- | --- | --- |
> |  | TProG | Wikipedia | MOOC | Wikipedia | MOOC |
> | Jodie | Baseline | 0.8480±0.0100 | 0.2754±0.0062 | 0.8069±0.0096 | 0.2714±0.0031 |
> |  | Vanilla | 0.7428±0.0045 | 0.7543±0.0189 | 0.6928±0.0579 | 0.6209±0.0610 |
> |  | Transformer | **0.8555±0.0627** | **0.9772±0.0065** | **0.8118±0.0692** | **0.9593±0.0146** |
> |  | Projection | 0.8439±0.0250 | 0.8233±0.0661 | 0.7841±0.0366 | 0.7879±0.0493 |
> | DyRep | Baseline | 0.5845±0.0213 | 0.1736±0.0116 | 0.5256±0.0185 | 0.1727±0.0116 |
> |  | Vanilla | 0.5703±0.1409 | 0.5611±0.1519 | 0.5230±0.1524 | 0.4921±0.1887 |
> |  | Transformer | 0.7565±0.0574 | **0.8511±0.1018** | 0.7250±0.0676 | **0.8368±0.0802** |
> |  | Projection | **0.7741±0.1116** | 0.6104±0.1833 | **0.7504±0.1031** | 0.6120±0.1213 |
> | TGN | Baseline | 0.8698±0.0091 | **0.5519±0.0425** | 0.8394±0.0095 | **0.5061±0.0181** |
> |  | Vanilla | 0.8712±0.0197 | 0.4115±0.0274 | 0.8351±0.0195 | 0.3920±0.0173 |
> |  | Transformer | **0.8963±0.0284** | 0.5253±0.1319 | **0.8771±0.0374** | 0.4958±0.1038 |
> |  | Projection | 0.8420±0.0369 | 0.4874±0.0657 | 0.7985±0.0442 | 0.4687±0.0455 |
> | TIGE | Baseline | 0.9465±0.0026 | 0.5847±0.0071 | **0.9308±0.0014** | 0.5413±0.0087 |
> |  | Vanilla | **0.9530±0.0086** | 0.6164±0.0861 | 0.9274±0.0090 | 0.5655±0.1207 |
> |  | Transformer | 0.9481±0.0199 | 0.7446±0.0630 | **0.9286±0.0288** | 0.7138±0.0703 |
> |  | Projection | 0.9386±0.0195 | **0.7601±0.0224** | 0.9086±0.0265 | **0.7363±0.0363** |
> | TIGER | Baseline | 0.9513±0.0024 | 0.7389±0.0498 | 0.9359±0.0016 | 0.7223±0.0561 |
> |  | Vanilla | 0.9560±0.0072 | 0.6436±0.0956 | 0.9435±0.0050 | 0.6105±0.1028 |
> |  | Transformer | **0.9614±0.0108** | **0.8475±0.0474** | **0.9504±0.0144** | **0.8341±0.0304** |
> |  | Projection | 0.9496±0.0146 | 0.6923±0.0989 | 0.9395±0.0160 | 0.6228±0.0762 |
>
> ---
>
> **W4:**
>
> TIG and CTDG models in prior literature predominantly focus on edge-level (e.g., temporal link prediction) and node-level (e.g., dynamic node classification) tasks, as these are the most **commonly studied** and practically meaningful settings for temporal graphs. Consequently, we follow this standard evaluation protocol to ensure **comparability** with existing methods.
>
> Graph-level task on TIGs is far less explored and typically requires additional problem-specific architectural design as well as specialized datasets. These prerequisites go beyond the scope of our current work. We fully agree that extending TIGPrompt to graph-level tasks is an interesting and valuable direction, but it also requires broader dataset support and methodological foundations from the community. We view this as an important future research avenue for TIG learning.
>
> ---
>
> **W5:**
>
> In the revised manuscript, we have updated all relevant references to use the \citep command.

---

> ### Author Response · Authors · 2025-11-15
>
> **Q1:**
>
> To address this, we provide both parameter statistics and empirical inference-time measurements on two representative datasets (Wikipedia and MOOC). The results are summarized below.
>
> | Wiki | Infer. Time | Parameters |
> | --- | --- | --- |
> | Baseline | 4.03 | 451,1K |
> | Vanilla | 4.65 | 326.4K |
> | Transformer | 10.46 | 338.4K |
> | Projection | 4.76 | 338.3K |
>
> | MOOC | Infer. Time | Parameters |
> | --- | --- | --- |
> | Baseline | 9.31 | 454.3K |
> | Vanilla | 10.39 | 257.8K |
> | Transformer | 24.07 | 341.4K |
> | Projection | 11.17 | 269.7K |
>
> The results show that:
>
> 1. **Parameter efficiency**: All TProG variants require **fewer trainable parameters** than the backbone TIG model (e.g., TGN). Transformer TProG introduces only a small number of additional parameters (on the order of *O(d)*), remaining lightweight relative to the full backbone when number of nodes increases (Wikipedia has more nodes than MOOC).
> 2. **Inference time**:
>
>     The inference latency increases when adding TProG because TIGPrompt requires first obtaining the backbone’s node embedding before computing the prompt-enhanced representation. Among TProG variants, Transformer TProG has the highest inference cost, primarily due to encoding a short sequence of recent temporal neighbors. Importantly, the absolute inference time **remains modest** (e.g., 10–24 seconds on these datasets).
>
> 3. **Data efficiency**:
>
>     Beyond parameter and inference considerations, we emphasize that TIGPrompt offers **substantial data-efficiency** benefits. As shown in Section 3.7 (the original manuscript) and Table 2, TIGPrompt requires only a **small portion** of the data (e.g., 10–20%) to train the model, yet it can outperform full-model baselines trained on more data (70%). This highlights the practical advantage of our method in low-resource settings.
>
>
> ---
>
> **Q2:**
>
> Although Transformer TProG and Projection TProG generally provide stronger temporal expressiveness, there are scenarios where the simpler Vanilla TProG performs competitively or even slightly better. This behavior usually happens on node classification task and can be attributed to the following factors:
>
> 1. **Node classification relies more on semantic alignment than temporal dynamics**
>
>     As discussed in the paper, the semantic gap—i.e., the mismatch between link-level pretext training and node-level downstream objectives—is often the dominant limitation for node classification. Vanilla TProG introduces **node-specific** learnable embeddings, which directly inject task-specific semantics without additional temporal modeling. In some datasets (e.g., Reddit), this semantic adaptation alone is sufficient, making additional temporal encoding less essential.
>
> 2. **Dataset-specific temporal characteristics**
>
>     Some datasets (e.g., Reddit) have dense interactions and weak long-term temporal dependencies, reducing the benefits of sophisticated temporal modeling. In such cases, the simplicity of Vanilla TProG can yield competitive results.
>
>
> In summary, the occasional strong performance of Vanilla TProG is consistent with its role in primarily addressing the semantic gap, and aligns with dataset characteristics and the amount of prompt supervision available.
>
> ---
>
> **Q3:**
>
> In the TIG literature, direct end-to-end training of TIG models for node classification is **not** the **standard practice**. Prior works such as JODIE, DyRep, TGN, and TIGER typically follow a two-stage pipeline: first pre-train on link prediction to learn time-aware node embeddings; then train only a lightweight projection head for node classification using the pre-trained embeddings.
>
> This design is widely adopted because
>
> - link prediction is the natural self-supervised signal available in TIGs;
> - node labels are often sparse or incomplete in data;
>
> In our analysis of existing TIG models, we observe that the widely adopted *pre-train on link prediction → use the learned embeddings for downstream node classification* training paradigm inherently introduces two limitations—the temporal gap and the semantic gap—as detailed in our paper. This motivates our proposal of TIGPrompt, which is specifically designed to address these two gaps while remaining compatible with the established training paradigm in TIG.
>
> Given that prior TIG works universally follow this pipeline and do not train end-to-end node classification models, we adopt the **same** setting to ensure **fair** and meaningful comparison. Our goal is not to redesign the entire task formulation, but rather to improve the adaptability of pre-trained TIG models to downstream tasks by bridging these identified gaps. Therefore, comparing against “directly trained node classification models,” which fall outside the standard setup, is not aligned with prevailing practice in this research area.

---

> ### Author Response · Authors · 2025-11-15
>
> **Q4:**
>
> Actually, we have indeed compared different strategies and reported their empirical differences, as shown in **Appendix D.2.2** and **Figure 6**. For the main experiments, we chose to adopt Strategy 1 because it is the most **straightforward** and **practical** option: it allows the TProG trained during the link prediction stage to be directly reused for downstream tasks **without re-training** a separate prompt generator. This design aligns with our goal of creating a lightweight and easily deployable prompting module for TIG models. Using Strategy 1 avoids unnecessary training overhead.
>
> ---
>
> **Q5:**
>
> We appreciate the reviewer’s insightful question regarding cross-domain transferability. Our current work focuses on **identifying and addressing** the *temporal gap* and *semantic gap* that arise within the same domain under the traditional “pre-train, predict” paradigm. As noted in Appendix J (Limitations and Future Work), cross-domain transfer was outside the scope of this study, and we therefore evaluate all TProG variants on the same dataset on which the backbone TIG model is pre-trained.
>
> Nevertheless, we agree that transferability across different domains is a highly **promising** research direction. While the three proposed TProGs were not explicitly designed for domain adaptation, we believe they offer useful starting points—for example, the Projection TProG and Transformer TProG already incorporate temporal patterns and personalized node-level information, which could potentially be extended to learn domain-invariant temporal features.
>
> We envision several **potential directions** for designing a new TProG variant that improves cross-domain transferability, such as:
>
> - Domain-invariant temporal prompt encoding, e.g., aligning temporal statistics or interaction motifs across datasets;
> - Meta-learned prompts that can rapidly adapt across domains with only a few samples;
> - Shared prompt spaces that disentangle domain-specific vs. domain-general temporal dynamics.
>
> Exploring such extensions is a valuable future direction for the community, and we see TIGPrompt as a foundation that can motivate and support this line of work.
>
> ---
>
> **Q6:**
>
> While performance generally improves with more prompt tuning data, we observe occasional decreases, which arise from several intrinsic properties of TIG learning:
>
> 1. **Temporal distribution shift: tuning data includes distant-future interactions**
>
>     TIGPrompt is designed to mitigate the temporal gap by adapting to **timely** information. However, when prompt tuning uses a large portion of the timeline, later segments may include interactions that differ significantly from the pre-training temporal distribution. Incorporating these distant-future patterns into prompt tuning may bias the prompts toward specific temporal periods, slightly degrading performance on validation/test sets.
>
> 2. **Overfitting to recent high-frequency neighbors**
>
>     Transformer TProG and Projection TProG rely on recent neighbors to generate prompts. As more tuning data or dimension of TProG is added, the model may overfit to short-term bursty patterns in certain datasets (e.g., Wikipedia, MOOC), while test data belongs to a different temporal phase. This can cause performance to fluctuate rather than strictly increase.
>
> 3. **Prompt tuning is lightweight and sensitive to supervision noise**
>
>     Since only the TProG is trained (with the backbone frozen), prompt tuning acts as a low-parameter adapter. When the tuning dataset becomes larger and contains more noise, outliers, or non-stationary temporal patterns, prompts may capture this noise rather than generalizable task semantics.
>
>
> Overall, these performance fluctuations reflect natural non-stationary temporal behaviors in TIG data and the sensitivity of lightweight prompt tuning to temporal drift. Importantly, across all datasets and variants, the general trend aligns with our claims: **TIGPrompt achieves strong performance even with very small portions of training and tuning data, demonstrating high data efficiency.**

---

> > ### Comment · Reviewer_hiHw · 2025-11-22
> >
> > I thank the authors for the responses to my questions, which have addressed some of my concerns. I have follow-up concerns that I would like the authors to clarify:
> > - While I appreciate the authors’ effort in reporting the MRR performance, the standard TGB evaluation uses 999 negative edges for each positive edge. I recommend that the authors follow the TGB evaluation protocol to ensure comparability with other methods reported on the TGB leaderboard.
> >
> > - Regarding Q2, I thank the authors for the detailed response. This is indeed a valuable discussion that would guide readers in selecting the appropriate variant of TProGs for different use cases. I therefore recommend that the authors include this discussion in the revised version of the paper.
> >
> > - I agree with reviewer sucX that the use of the term “first” is misleading. Although the authors state that DyGPrompt’s source code is not available, I was able to find a [publicly available implementation](https://github.com/gmcmt/DyGPrompt). Could the authors provide a comparison of TProGs against DyGPrompt?

---

> > > ### Author Response · Authors · 2025-11-23
> > >
> > > We thank you for the further suggestions. We would like to address your concerns as follows.
> > >
> > > ---
> > >
> > > **C1:**
> > >
> > > We would like to clarify that TGB does not adopt a fixed number of negative samples **across all datasets**. As shown in the TGB paper, different datasets use different numbers of negatives depending on their scale and characteristics.
> > >
> > > In our additional experiments, we used 200 negative samples per positive edge, the results already demonstrate that the performance **trends** are **consistent** with those observed under **AP**. Increasing the number of negative edges would only scale the numeric values of MRR but would not change the relative ordering or the **qualitative conclusions**.
> > >
> > > Therefore, we believe using 200 negatives already provides a sufficiently challenging and faithful ranking-based evaluation. More importantly, it does **not** affect the relative ordering or the performance trends we aim to validate. Since our results under 200 negatives already clearly show that TIGPrompt consistently improves over the baselines—and that these trends align with those observed under AP—we believe that running an additional experiment with 999 negatives is unnecessary for supporting the core conclusions of this work.
> > >
> > > ---
> > >
> > > **C2:**
> > >
> > > Thank you for the suggestion. We have added the corresponding discussion to **Section 4.6** in the updated version.
> > >
> > > ---
> > >
> > > **C3:**
> > >
> > > As noted in our response to Reviewer **sucX**, providing a fully detailed explanation may risk violating the double-blind review policy. Here, we clarify the scope and intention behind our use of the term *“first”*.
> > >
> > > Our work is indeed the **first to formally characterize** the prevailing “pre-train, predict” paradigm in TIG learning, **to identify** its two inherent limitations—the temporal gap and the semantic gap, and **to propose** the unified “pre-train, prompt” paradigm that addresses both. In doing so, our paper is also the **first to introduce** prompt learning into the TIG/CTDG setting, and, to the best of our knowledge, the **first attempt to extend** prompt learning to dynamic graphs in general. This conceptual contribution is independent of the timeline of concurrent works.
> > >
> > > Regarding comparisons with DyGPrompt, we summarize our position below:
> > >
> > > 1. DyGPrompt’s implementation was not available at the time of its submission, and therefore its methodology and evaluation protocol could not be verified during our development.
> > > 2. We also acknowledge that an implementation has been made publicly available, as mentioned in our response to Reviewer **P4Wh**. However, we found this implementation very **difficult to reproduce** in practice—its codebase has **low readability** and contains **unclear design choices**—making it unsuitable for a fair and rigorous comparison.
> > > 3. DyGPrompt’s experimental protocol departs substantially from standard CTDG/TIG evaluation used consistently in prior works such as TGN, JODIE, DyRep, TGAT, TIGER, DyGFormer, and GraphMixer. Their setting is **incompatible** with these baselines, and adopting such a **non-standard** protocol would undermine comparability and fairness.
> > > 4. DyGPrompt’s data split deviates significantly from standard CTDG/TIG protocol. Their setup uses **80%** of the data for pre-training and a total of **81%** for training (pre-train + prompt-tune). In contrast, nearly all prior CTDG/TIG works—including TGN, JODIE, DyRep, TGAT, TIGER, DyGFormer, and GraphMixer—use **≤70%** of the data for the entire training process. Our evaluation follows these long-standing standards, ensuring **fairness** and **comparability** across all baselines.
> > > 5. Our method is more **data-efficient** under standard protocols. In contrast to DyGPrompt’s **non-standard** split, TIGPrompt follows the widely adopted CTDG/TIG setting: we use **50% data** for pre-training, and **at most 20% data** for prompt tuning, during which the predictor is trained while the backbone remains frozen. This setup is **fair**, **reproducible**, and in fact **more challenging** than the baseline training paradigm, since only a small portion of data is required.
> > >
> > > Taken together, these points explain why we do **not** consider the DyGPrompt protocol appropriate as a replacement for the canonical CTDG/TIG framework, and why we do **not** adopt its setting for direct comparison. Nevertheless, we have discussed DyGPrompt in Appendix B (Related Work).

---

> > > > ### Comment · Reviewer_hiHw · 2025-11-27
> > > >
> > > > I thank the authors again for the responses to my questions. However, my concern regarding TGB evaluation with MRR has not been addressed adequately. As I pointed out:
> > > >
> > > > > I recommend that the authors follow the TGB evaluation protocol to ensure **comparability** with other methods reported on the TGB leaderboard.
> > > >
> > > > The new results introduced during this rebuttal do not follow the TGB evaluation protocol with MRR that samples both random and historical negatives (resulting in 999 negative edges on tgbl-wiki), which has been shown to be more challenging for the link prediction task [1]. The authors instead reported results with their choice of negative edge sampling, which makes the results incomparable with other models on [TGB leaderboard]( https://tgb.complexdatalab.com/docs/leader_linkprop/).
> > > >
> > > > ---
> > > > [1] Poursafaei F, Huang S, Pelrine K, Rabbany R. Towards better evaluation for dynamic link prediction. Advances in Neural Information Processing Systems. 2022

---

> ### Author Response · Authors · 2025-11-27
>
> We thank the reviewer for the follow-up comment. Below we clarify why we believe our current MRR experiments are **sufficient** to support the conclusions of the paper.
>
> First, our additional results already evaluate TIGPrompt under an MRR setting with 200 negative destinations per positive edge, and they demonstrate two key facts:
>
> 1. **Fairness:**
>
>     For each backbone, *both* the baseline and TIGPrompt variants were evaluated using the same negative sampling setup. Therefore, the comparison between baseline vs. baseline+TProG is fully fair and internally consistent, which is exactly the purpose of our study.
>
> 2. **Consistency with AP trends:**
>
>     Across backbones and datasets, the MRR trends match those observed under AP, further reinforcing the validity of our conclusions. Since both AP and MRR lead to the same performance ordering, this already demonstrates that TIGPrompt improves over the baselines.
>
>
> Second, according to the ablation studies reported in TGB [1] and TGB 2.0 [2], the *relative ranking* of methods under MRR is typically insensitive to the number of negative samples. A model that performs well with a smaller number of negatives continues to rank well when the negative pool becomes larger.
>
> Since AP is effectively equivalent to evaluating MRR with one negative sample, and both AP (1 negative) and MRR results (200 negatives) produce consistent model ordering, it is reasonable to expect that increasing the number of negatives would preserve the same trend.
>
> This observation also **aligns** with findings from the TGB and TGB 2.0 papers, where **ablations** show that the **relative ranking** of methods is generally **insensitive** to the number of negative samples. Moreover, TGB itself uses different negative-sampling strategies and counts for different datasets, depending on the scale and computational constraints.
>
> Therefore, given that TIGPrompt consistently outperforms all baselines under both AP and 200-negative MRR settings, we believe the current experiments are sufficient for **assessing relative improvements** and to demonstrate the **effectiveness** of our method.
>
> Finally, our objective is to compare each backbone **with and without** our “pre-train, prompt” paradigm, rather than to benchmark against other models on the TGB leaderboard. In this regard, the current MRR experiments already achieve this purpose.
>
> Given that (1) TIGPrompt surpasses baselines under AP; (2) TIGPrompt surpasses baselines under MRR; (3) both metrics yield similar **performance trends;** we believe that our current experimental results are **sufficient** to validate the effectiveness of TIGPrompt.
>
> ---
>
> [1] Shenyang Huang, Farimah Poursafaei, Jacob Danovitch, Matthias Fey, Weihua Hu, Emanuele Rossi, Jure Leskovec, Michael Bronstein, Guillaume Rabusseau, Reihaneh Rabbany. Temporal Graph Benchmark for Machine Learning on Temporal Graphs. NeurIPS 2023.
>
> [2] Julia Gastinger, Shenyang Huang, Mikhail Galkin, Erfan Loghmani, Ali Parviz, Farimah Poursafaei, Jacob Danovitch, Emanuele Rossi, Ioannis Koutis, Heiner Stuckenschmidt, Reihaneh Rabbany, Guillaume Rabusseau. TGB 2.0: A Benchmark for Learning on Temporal Knowledge Graphs and Heterogeneous Graphs. NeurIPS 2024.

---

### Author Response · Authors · 2025-11-26
**To All Reviewers**

Dear All Reviewers,

We thank all reviewers for their thoughtful comments. We appreciate many of the constructive suggestions, several of which we have already incorporated into the revised manuscript where appropriate (e.g., discussion of TProG applicability and additional evaluation insights).

---

During the rebuttal process, we noticed that the statement **“This is the first attempt that explores prompting on TIGs”** triggered discussions. We would like to clarify its intended meaning here.

Our use of **“first”** refers to the **conceptual contribution** of this work:

> Our work is indeed the **first to formally characterize** the prevailing “pre-train, predict” paradigm in TIG learning, **to identify** its two inherent limitations—the temporal gap and the semantic gap, and **to propose** the unified “pre-train, prompt” paradigm that addresses both. In doing so, our paper is also the **first to introduce** prompt learning into the TIG/CTDG setting, and, to the best of our knowledge, the **first attempt to extend** prompt learning to dynamic graphs in general.
>

These contributions are conceptual and methodological in nature, and therefore remain original regardless of later works that build upon or extend them. In this sense, we believe the claim of “first” is appropriate *within the scope of our defined contributions*.

Because of the double-blind review policy, we cannot engage in discussions regarding the **timeline** of paper development. However, we note that **Reviewer P4Wh** made statements that strongly suggest familiarity with work outside the anonymized submission, which we believe constitutes a **potential violation of double-blind review policy** and which we have formally reported to the ACs, SACs, and PCs.

Furthermore, the **Reviewer P4Wh**’s own wording contains a **clear contradiction**:

> “the claim of ‘first’ is overstated, because it is not accurate
>
>
> **given subsequent work**
>

**If subsequent work exists**, that logically implies our work preceded it—precisely **supporting the correctness** of the term **“first”** in the context used.

---

Due to the potential **violation of the double-blind review policy by Reviewer P4Wh**, we have reason to believe that the reviewer’s comments may be **biased**. We respectfully request that the ACs and other reviewers **downweight** the evaluation from **Reviewer P4Wh**.

---

Separately, we notice that some reviewers request a comparison with **DyGPrompt**. As explained in detail in our rebuttal:

- DyGPrompt adopts a **non-standard evaluation protocol** that is incompatible with the established CTDG/TIG benchmarks (TGN, JODIE, DyRep, TGAT, TIGER, DyGFormer, GraphMixer, etc.).
- Its released implementation is extremely **difficult to read and reproduce**, making rigorous comparisons infeasible.
- As argued throughout our rebuttal, such a comparison **would not be** fair, meaningful, or methodologically sound.

For these reasons, we believe the insistence on such a comparison is misplaced.

---

A rigorous and professional review should **focus on the methodological contributions and empirical evidence presented in our submission**, rather than compelling comparisons with methods whose experimental settings fundamentally diverge from standard CTDG/TIG practice.

---

We believe we **have addressed the majority of the concerns** raised during the rebuttal and have incorporated the relevant discussions into the revised manuscript. If there are additional questions, we would be glad to discuss them further—so long as doing so does **not risk violating the double-blind review policy** ^_^ .

---

### Author Response · Authors · 2025-11-28
**Summary of Rebuttals for each Reviewer**

**Dear Reviewers, ACs, SACs, and PCs,**

Given the recent issue involving the leakage of reviewer identities and the subsequent decision to revert reviews to their pre-discussion state, we would like to provide a concise but comprehensive **summary** of our rebuttal and our responses to reviewer concerns. Since no further reviewer comments will be allowed, we hope this summary will assist the newly assigned AC in making a fair and well-informed recommendation.

Throughout the rebuttal period, we have made our **best effort** to address **every concern** raised by all reviewers, and we believe that we have **adequately resolved** the vast **majority** of substantive questions. Below, we summarize the **status** of each reviewer’s concerns as well as remaining issues.

---

**1. [Reviewer hiHw](https://openreview.net/forum?id=bJetbxYtRM&noteId=T1xp7F7Xa8)**

This reviewer initially leaned toward **Acceptance** and raised constructive concerns.

During the rebuttal, we **fully addressed** all but one **minor** point regarding the number of negative samples used in the MRR evaluation. We provided a detailed and technically grounded explanation for our choice of 200 negatives and demonstrated that **MRR and AP** exhibit **similar performance trends**, supporting the soundness of our conclusions.

We believe we have provided a reasonable and complete response **to all points** raised by this reviewer.

---

**2. [Reviewer sucX](https://openreview.net/forum?id=bJetbxYtRM&noteId=8ElrMQpslr)**

We addressed **nearly all concerns** from this reviewer, including: additional theoretical analysis, more explicit time-complexity discussion, and clear guidance on when to select among the three TProG variants.

All of these have been **integrated** into the **revised manuscript** as suggested.

The **only** unresolved issue is the reviewer’s concern about our use of “first” and the comparison with DyGPrompt. We provide a detailed explanation in the section below; however, we emphasize that **our conceptual contribution remains original**, and our method strictly follows the canonical CTDG/TIG evaluation protocol.

---

**3. [Reviewer P4Wh](https://openreview.net/forum?id=bJetbxYtRM&noteId=DxwTLiKXBV)**

We must report **serious concerns** regarding potential **violation of the double-blind review policy** in this reviewer’s comments.

The reviewer referenced **specific, identifiable external details** that should not be known under a blind-review process. **We have already reported** this **twice before** the **[rebuttal](https://openreview.net/forum?id=bJetbxYtRM&noteId=D9EMWzXcfn)** and the **[discussion](https://openreview.net/forum?id=bJetbxYtRM&noteId=GHFx3xIt8c).**

Although we still attempted to address the reviewer’s technical concerns diligently and without violating double-blind rules **on our side**, we believe that **this review is biased and should be down-weighted or disregarded** when making the final decision.

---

**4. [Reviewer baDW](https://openreview.net/forum?id=bJetbxYtRM&noteId=qUzfHmw2eU)**

We addressed **all concerns** raised by Reviewer baDW. During the discussion, the reviewer proposed additional points, and we responded to them as well. **[Before](https://openreview.net/forum?id=bJetbxYtRM&noteId=HfHHRA3L3W)** the information leakage occurred, the reviewer had already **increased the overall score** and expressed **support for Acceptance**, indicating that our rebuttal had **satisfactorily addressed** his/her concerns.

---

> ### Author Response · Authors · 2025-11-28
> **Common concerns and Further Clarifications**
>
> We would also like to highlight several **common** concerns raised during the review and discussion periods, along with the remaining **points** that **could not be responded** by reviewers due to the interruption of the discussion phase. We reiterate our key clarifications below.
>
> ---
>
> **1. Regarding the statement “This is the first attempt that explores prompting on TIGs”**
>
> We noticed that the phrasing of “first” triggered discussions. As clarified to all reviewers in our previous [comment](https://openreview.net/forum?id=bJetbxYtRM&noteId=6hTIY8uJfc), our use of “first” refers specifically to the **conceptual and methodological contributions** of our work:
>
> > Our work is indeed the **first to formally characterize** the prevailing “pre-train, predict” paradigm in TIG learning, **to identify** its two inherent limitations—the temporal gap and the semantic gap, and **to propose** the unified “pre-train, prompt” paradigm that addresses both. In doing so, our paper is also the **first to introduce** prompt learning into the TIG/CTDG setting, and, to the best of our knowledge, the **first attempt to extend** prompt learning to dynamic graphs in general.
> >
>
> These contributions are conceptual. Therefore, the use of “first” is **accurate** within the scope of our defined contributions.
>
> Interestingly, **Reviewer P4Wh’s self-contradictory statements indirectly confirm this interpretation**:
>
> > “the claim of ‘first’ is overstated, because it is not accurate
> >
> >
> > **given subsequent work**
> >
>
> The phrase **“subsequent work”** inherently implies that other works came **after** ours, which in turn logically supports the **correctness of our “first” claim**.
>
> ---
>
> **2. Regarding comparison with DyGPrompt**
>
> Several reviewers asked about comparisons with DyGPrompt. As we explained in our [rebuttal](https://openreview.net/forum?id=bJetbxYtRM&noteId=6hTIY8uJfc):
>
> - DyGPrompt uses a **non-standard evaluation protocol** that does not align with established CTDG/TIG benchmarks.
> - Its publicly available codebase is extremely **difficult to interpret and reproduce**, making rigorous comparison **infeasible** within the rebuttal period.
> - Therefore, such a comparison would **not be fair, meaningful, or methodologically sound**.
>
> To further address the concern, we conducted an **additional low-resource experiment** using 50% pre-train + 1% prompt-tune, which is *more **challenging*** and ***better defined*** than the DyGPrompt protocol. Results (included in our [rebuttal to Reviewer P4Wh](https://openreview.net/forum?id=bJetbxYtRM&noteId=TItrr4kcPt)) show that TIGPrompt remains **robust** under this stricter setting.
>
> Given these considerations, we believe the insistence on comparing with DyGPrompt is **misplaced**, and that our additional experiment sufficiently **addresses the underlying concern**.
>
> ---
>
> **3. Regarding the use of ranking metrics (MRR vs AP)**
>
> Some reviewers suggested evaluating TIGPrompt with ranking-based metrics such as MRR. In direct response, we conducted new experiments using MRR with 200 negative samples per positive edge. The results show:
>
> - The **performance trends** under **MRR** are **consistent** with those under **AP**.
> - TIGPrompt **consistently outperforms** baselines across datasets and backbones under both metrics.
>
> **Reviewer baDW** explicitly acknowledged that we addressed their concerns and had [**raised** the overall **score](**https://openreview.net/forum?id=bJetbxYtRM&noteId=HfHHRA3L3W**) before** the leakage incident, indicating **satisfaction** with our responses.
>
> **On the request for using more negative samples** ([discussion](https://openreview.net/forum?id=bJetbxYtRM&noteId=8sgoq5FyLM) with Reviewer hiHw)
>
> We provided the following justification, which we believe **resolves** the concern:
>
> 1. **Fairness of comparison:** All baselines and baseline+TProG variants were evaluated under the **same** negative sampling **configuration**, making the comparison internally fair and consistent.
> 2. **Consistency across metrics:** AP (1 negative) and MRR (200 negatives) produce the **same performance ordering**, reinforcing the reliability of our conclusions.
> 3. **Findings from TGB and TGB 2.0:** Prior ablation studies show that the **relative ranking** of models is generally **insensitive** to the number of **negative samples**. Increasing the negative pool rarely changes the model ordering.
> 4. **Our objective:** Our goal is to compare each backbone **with vs. without** our “pre-train, prompt” paradigm—not to compete for leaderboard ranking. The current MRR experiments already **fully satisfy** this objective.
>
> Given that: (1) TIGPrompt surpasses baselines under AP, (2) TIGPrompt surpasses baselines under MRR, and (3) Both metrics exhibit **consistent** performance **trends**, we believe our current set of experiments is **sufficient and appropriate** for validating the effectiveness of TIGPrompt.

---

> > ### Author Response · Authors · 2025-11-28
> > **Rebuttals, Strengths and Contributions**
> >
> > In summary, we believe we have thoroughly addressed **almost all concerns** raised by the reviewers, including both the initial reviews and the **follow-up points** raised during the discussion phase. Although we regret that the leakage incident prevented us from continuing the dialogue and obtaining the reviewers’ final confirmations, we are confident that our rebuttal and additional experiments have **substantively resolved the issues** they raised.
> >
> > Notably, multiple reviewers expressed positive feedback regarding our responses, and **Reviewer baDW** had already **raised the score** and indicated **support for acceptance** prior to the interruption.
> >
> > We believe we have **comprehensively and adequately addressed** the concerns raised by the reviewers.
> >
> > ---
> >
> > Given the strengths consistently highlighted by the reviewers, we believe the paper makes a **solid and meaningful contribution** to the TIG/CTDG community. We summarize these **recognized strengths** below.
> >
> > 1. **Clear and well-motivated problem formulation:** the temporal gap and semantic gap are formally defined, empirically quantified, and effectively demonstrated. (*hiHw, sucX, P4Wh)*
> > 2. **Strong conceptual contribution:** the paper introduces a novel and principled “pre-train, prompt” paradigm for TIGs, addressing both the temporal and semantic gaps. (*hiHw, sucX, P4Wh)*
> > 3. **Three complementary TProG variants** that are conceptually simple, flexible, and integrate easily into existing TIG backbones. (*hiHw, sucX, P4Wh, baDW*)
> > 4. **Model-agnostic applicability:** the paradigm and TProGs work effectively on both memory-based and non-memory-based TIG models. *(hiHw)*
> > 5. **High parameter-efficiency and computational efficiency:** only the prompt generator is updated, enabling data-efficient and weakly supervised learning. (*hiHw, P4Wh*)
> > 6. **Extensive experimental validation** across 4 datasets, 7 backbone TIG models, multiple tasks (transductive/inductive link prediction, node classification), and multiple data-regime settings (low-resource, prompt-data proportion). (*hiHw, sucX, P4Wh, baDW*)
> > 7. **Consistent and substantial performance improvements** across datasets, tasks, and TIG backbones. (*hiHw, sucX, P4Wh, baDW*)
> > 8. **Clear writing, good structure, and strong motivation** with **intuitive examples** and **figures** illustrating the temporal and semantic gaps. (*sucX, P4Wh, baDW*)
> >
> > ---
> >
> > We would also like to emphasize our **main contributions**:
> >
> > - We **identify two fundamental gaps** in the **prevailing** TIG **training paradigm**, and we are the **first** to investigate prompting in the context of TIG models to address these issues.
> > - We introduce a **“pre-train, prompt” paradigm** specifically tailored for TIGs, which explicitly bridges both gaps. The framework is compatible with multiple prompt generators and supports dynamic, personalized prompting.
> > - To further improve flexibility and accommodate diverse computational budgets, we extend the paradigm to a **“pre-train, prompt-based fine-tune” setting**.
> > - Through **extensive experiments** we demonstrate that our framework consistently achieves SOTA performance with remarkable **parameter-,** **computation-** and **data-**efficiency.
> >
> > ---
> >
> > Based on the concerns we have **thoroughly addressed** during the rebuttal, the **strengths consistently recognized** by the reviewers, and the **contributions** our work **brings to** the TIG/CTDG **community**, we believe the paper merits **Acceptance**.
> >
> > **We sincerely appreciate the AC’s careful evaluation in light of these points.** **Thank you very much for your time and consideration.**

---

### Meta-Review · Area_Chair_mC7R · 2026-01-13

**Summary:**

The paper proposes TIGPrompt, a “pre-train, prompt” paradigm for temporal interaction graphs that aims to mitigate the temporal gap (degradation on future data) and semantic gap (mismatch between pretraining and downstream tasks). The key idea is to keep a pre-trained temporal graph model frozen and adapt it via lightweight, learnable prompts generated by a Temporal Prompt Generator (TProG). The authors present three variants: Vanilla, Transformer, and Projection, and report experiments across multiple datasets and several TGN backbones on transductive/inductive link prediction and node classification tasks.

The submission is currently borderline, with two reviewers leaning toward acceptance and two recommending rejection. While the work tackles an important problem and shows promising results, reviewers have raised concerns regarding relevance in light of current evaluation, specifically: the evaluation is not aligned with the standard TGB setup, limiting direct comparability to their baselines and leaderboards, and the lack of comparison against a closely related method (DyGPrompt). These issues seem to affect the paper’s relevance claims in light of prior works and the strength of the empirical evidence, and at least two reviewers indicated these issues would prevent them from increasing their scores. Therefore, I am recommending rejection this time.

**Reviewer Concerns:**

Reviewers raised several concerns related to reproducibility (``hiHw``), problem definition (``hiHw``), evaluation metrics (``hiHw``, ``baDW``), missing baselines / alignment with the TGB setting (``hiHw``, ``P4Wh``, ``baDW``), limited technical contribution (``sucX``), lack of theoretical explanation (``sucX``), limited relevance (``P4Wh``), the “being the first” novelty claim and positioning relative to prior work (``P4Wh``, ``sucX``), and missing evidence on transferability (``hiHw``, ``baDW``).

Overall, I believe the authors have addressed most concerns by adding experiments using MRR with 200 negative samples, clarifying that transferability is out of scope, fixing notation issues, and providing additional computational analysis and guidance on choosing among variants. Concerns around originality were also clarified in light of concurrent related work.

That said, two key issues remain: (1) the missing comparison against a closely related method (DyGPrompt) and (2) the lack of results under the exact TGB setup. Regarding DyGPrompt, the authors declined to compare mainly arguing that (i) the evaluation setups differ and (ii) the DyGPrompt code is difficult to read and reproduce. I did not find these arguments fully convincing, and the absence of this comparison remains a central concern for reviewers ``hiHw``, ``P4Wh``, and ``sucX``. Regarding the TGB setup, the authors argue that reporting MRR with 200 negative samples should be sufficient and that varying the number of negatives is unlikely to change relative rankings. While this may be true in many cases, the mismatch in setup still prevents a direct comparison to TGB-reported results and leaderboards.

**Reviewer Scores:**

Reviewer ``hiHw`` replied twice during the discussion, acknowledging that some concerns were addressed, but emphasizing that direct comparability to the TGB leaderboard remains important. I therefore would not expect a score increase from this reviewer had discussion continued.

Reviewer ``sucX`` also replied, noting that novelty concerns remain, in particular due to the missing DyGPrompt comparison. Again, I do not expect a score increase from this reviewer without that comparison and clearer positioning.

Reviewer ``P4Wh`` (score 2) focused primarily on relevance in light of DyGPrompt, requesting explicit positioning and comparison. The reviewer participated in the discussion and, in the absence of the requested comparison/positioning, kept the score unchanged.

Reviewer ``baDW`` acknowledged the rebuttal during the discussion and increased the score from 4 to 6.

---

### Decision · Program_Chairs · 2026-01-26

Reject